SciPost Physics                                            

# Spin chains, defects, and quantum wires
# for the quantum-double edge

Victor V. Albert[1,2], David Aasen[3,4], Wenqing Xu[1],
Wenjie Ji[5], Jason Alicea[2], John Preskill[2]

**1** Joint Center for Quantum Information and Computer Science,
National Institute of Standards and Technology and University of Maryland, College Park MD
**2** Institute for Quantum Information and Matter & Walter Burke Institute for
Theoretical Physics, California Institute of Technology, Pasadena CA
**3** Kavli Institute for Theoretical Physics, University of California, Santa Barbara CA
**4** Microsoft Quantum, Microsoft Station Q, University of California, Santa Barbara CA
**5** Department of Physics, University of California, Santa Barbara CA

## Abstract

**Non-Abelian defects that bind Majorana or parafermion zero modes are prominent in several topological quantum computation schemes. Underpinning their established understanding is the quantum Ising spin chain, which can be recast as a fermionic model or viewed as a standalone effective theory for the surface-code edge — both of which harbor non-Abelian defects. We generalize these notions by deriving an effective Ising-like spin chain describing the edge of quantum-double topological order. Relating Majorana and parafermion modes to anyonic strings, we introduce quantum-double generalizations of non-Abelian defects. We develop a way to embed finite-group valued qunits into those valued in continuous groups. Using this embedding, we provide a continuum description of the spin chain and recast its non-interacting part as a quantum wire via addition of a Wess-Zumino-Novikov-Witten term and non-Abelian bosonization.**

## Contents

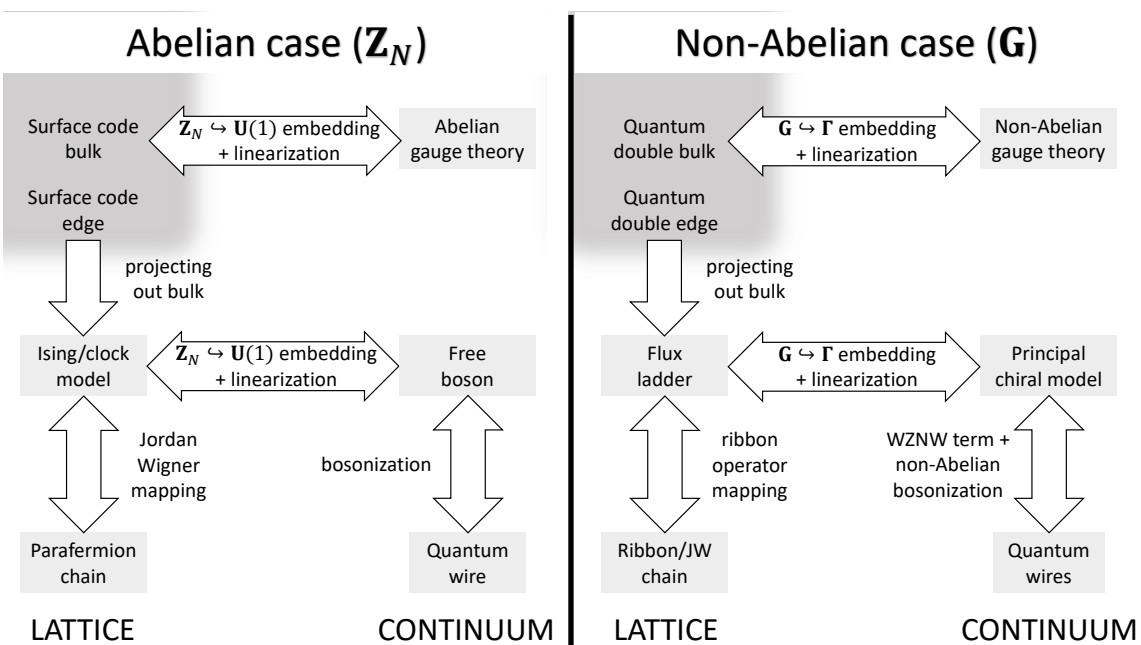

Figure 1: Left panel: Summary of connections between the $\mathbf{Z}_N$ surface code and various physical systems, enabled by the 1D quantum Ising spin chain. Right panel: Analogous summary for quantum double models associated with finite group $\mathbf{G}$, outlining the results of this work.

# 1 Introduction

Topological quantum computation [1–5], in contrast to many other available blueprints for a quantum computer, holds the promise of computation in an inherently robust fashion. While there is no intrinsic topological order in a 1D spin system [6,7], 1D subsystems (e.g., edges or line defects) of a 2D topologically ordered state can inherit some of the parent medium's topological properties — yielding phenomena not possible in strictly one dimension. In particular, the 1D subsystem endpoints can realize point-like *non-Abelian defects* that induce nontrivial effects when the medium's anyonic excitations are moved around them. Such defects, though strictly speaking confined, emulate bona-fide non-Abelian anyonic quasiparticles and provide a promising avenue for topologically protected quantum operations [8–18].

Non-Abelian defects can be rigorously identified in certain lattice models for topological phases. However, since many experimental platforms are more naturally described by a low-energy continuum framework, developing continuum versions of defects is also important. Specifically, 1D sections of various materials, modeled by novel types of *quantum wires* [19–21], admit low-energy counter-propagating excitations represented by field operators, and identifying defects in such systems hinges on the ability to construct them out of continuum degrees of freedom. The lattice and continuum versions of defects are of complementary importance: the former yields topologically protected gates for quantum information stored in an engineered quantum device (see [22] and refs. therein), while the latter yields such gates for quantum information stored in a real material [23].

## 1.1 $\mathbf{Z}_N$-type defects on the lattice and in the continuum

Paradigmatic lattice-form defects arise on the edge of the surface code — a representative of a topological phase known as $\mathbf{Z}_2$ topological order. The surface code admits gapped edges, which are modeled by instances of the quantum Ising spin chain. When treated as a standalone

model, this $\mathbf{Z}_2$-symmetric spin chain admits two phases, described by a $\mathbf{Z}_2$-ordered and a disordered spin state, respectively, with neither admitting any topological protection. However, when treated as an effective theory for the surface-code edge [8, 24–29], the spin chain's symmetry becomes a locally unbreakable constraint arising from the surface-code medium being in a ground state. This constraint yields protected subspaces as well as robust operations that can be performed using anyonic excitations or domain-wall defects between gapped edges in different phases.

The Ising spin chain is also relevant to realizing the above family of $\mathbf{Z}_2$-*type* defects in electronic media. When viewed as a fundamentally fermionic system via the Jordan-Wigner mapping [30, 31], the model's $\mathbf{Z}_2$ symmetry becomes a locally unbreakable fermion-parity constraint. The $\mathbf{Z}_2$-ordered spin state translates into a topological superconducting phase for fermions, whereas the disordered spin state translates into a trivial superconductor. Domain-wall defects separating topological and trivial phases bind unpaired *Majorana zero modes* — operators that leave the energy of the system unchanged, but cycle through states in the system's topologically protected ground-state subspace [32,33]. The electronic connection makes such defects unique, in that they can in principle occur even in strictly 1D platforms [34–38]. Majorana zero modes continue to be the centerpiece of several blueprints for scalable and protected quantum information processing (e.g., [39–42]).

$\mathbf{Z}_{N>2}$-type generalizations of the above defects on the lattice can be readily found in the $\mathbf{Z}_N$ topologically ordered phase of the qunit extension of the surface code, whose edge is described by the *clock model*, and whose domain-wall defects bind lattice *parafermion zero modes* [43–47]. The situation becomes more difficult on the electronic front: since the clock model for the $N > 2$ case is no longer readily mapped to a fermionic system, this case can only be realized on a 1D section of 2D topological order. Using the fact that edge excitations of a fractional quantum Hall medium have similar algebraic structure as that of $\mathbf{Z}_N$-type defects, several works have identified continuum-form defects on the edges of such systems [14–17].

## 1.2   G-type defects and this work

With defects associated with $\mathbf{Z}_N$-symmetric systems extensively studied both on the lattice and in the continuum, we study defects associated with **G**-symmetric systems, continuing a line of work [48, 49] aimed at identifying applications of quantum state spaces described by *non-Abelian groups* **G**. On the lattice front, the relevant family consists of the surface code's intricate "non-Abelian" generalizations — *quantum double models* [1–3, 5] — defined on lattices whose spins take values in **G** [50–55]. Such systems house excitations described by non-Abelian anyons, whose fusion can yield several outcomes (as opposed to just one in the abelian case). Information can be encoded in a robust fashion directly into superpositions of the fusion outcomes, and universal computation can be implemented with potentially less overhead than conventional schemes (e.g., without magic-state distillation) [56–60]. The price paid for these advantages is increased complexity of edge excitations and bulk defects, and there has been a comprehensive effort establishing their algebraic properties [61–65]. On a similarly abstract footing, a category-theoretic classification of quantum-double edge theories has also been established [10, 29, 64, 66–70]. Prior efforts modeled the edge while explicitly including the bulk, and what is missing is a 1D *standalone* microscopic model describing exclusively the edge, along with its 1D **G**-type defects. A continuum formulation of the quantum-double edge, essential for identifying **G**-type defects in electronic media, has also yet to be explored.

In this work, we identify effective 1D lattice- and continuum-based theories for the quantum-double edge, providing a general ansatz for their associated 1D **G**-type defects on the lattice. By viewing the aforementioned $\mathbf{Z}_N$-type models through the mathematical lens of representation theory, we map out a web of connections for quantum double models (Fig. 1, right panel) that generalizes the existing web for the $\mathbf{Z}_N$ surface code (Fig. 1, left panel). At the center of

this web is a freshly-derived effective theory for a general quantum-double edge, whose lattice version we show to be a lightly generalized version of the *flux ladder* of Munk, Rasmussen, and Burrello [71]. This Ising-like edge theory contains one-body "transverse-field" and two-body "longitudinal-field" terms similar in spirit to those of the Ising model, and its **G** symmetry can be interpreted as a locally unbreakable constraint arising from the quantum-double medium being in a ground state.

We also make some progress on the continuum front. First, we provide a systematic procedure for transforming an edge lattice model into a field theory by embedding each **G**-valued lattice site into a larger space valued in a Lie group $\mathbf{\Gamma} \supset \mathbf{G}$ and taking the continuum limit. We then make first contact with electronic systems by mapping a specific and slightly modified case of the continuum edge theory to a set of coupled quantum wires.

To contrast with previous work, we do not pursue exhaustive and abstract classification schemes, but instead focus on microscopic realizations of general defects (avoiding the language of category theory in the spirit of Ref. [5]). We lessen the complexity of 2D quantum-double defects, introducing simplified versions expressed in an effective space with one less dimension. We also provide a direct, albeit heuristic, procedure to map a lattice model to a corresponding field theory, which can be recast as a quantum wire in certain cases. In these ways, our extended web of connections helps pave the way toward eventual realization of robust defect-based gates in both engineered quantum devices and electronic materials.

## 2 Summary of results

We analyze a general quantum-double edge from a lattice perspective, developed in Secs. 3 and 4, as well as a continuum perspective, developed in Secs. 5 and 6.

**Lattice edge theory**   We derive a standalone effective theory for the edge of a quantum double model with group **G**. The only requirement we impose on the edge Hamiltonian is commutation with the bulk, encompassing special cases such as gapped edges [61–65], gapless points [69, 70], and combinations thereof [72–74]. The theory is obtained by course-graining the 2D quantum-double bulk [24, 75–78] and projecting its edge into a 1D subspace defined by the bulk being in a ground state. After reviewing the known surface code edge derivation ($\mathbf{G} = \mathbf{Z}_N$) [8, 24–29], we derive the edge theory for general groups in Sec. 3.

The edge model consists of the most general Ising-like local translationally invariant Hamiltonian acting on a chain of **G**-valued spins, with a global **G**-symmetry enforced by the bulk ground-state constraint. Following Ref. [71], we refer to the model as the *flux ladder*. For finite abelian groups, the model encompasses the quantum Ising ($\mathbf{Z}_2$) and clock ($\mathbf{Z}_N$) models [45, 47]. For continuous groups, the flux ladder generalizes the XY model [for $\mathbf{G} = \mathbf{U}(1)$, a.k.a. the $\mathbf{O}(2)$-rotor model] [79, 80], principal chiral model [for continuous groups such as $\mathbf{SU}(2)$, a.k.a. the $\mathbf{O}(4)$-rotor model or nonlinear sigma model] [81–83], and Kogut-Susskind (pure) gauge theory [84–86] on a particular geometry.

We identify quantum-double analogues of $\mathbf{Z}_N$ parafermionic modes. In abelian cases, such modes arise from the Jordan-Wigner transformation, and correspond to nonlocal anyonic strings lying on the edge of the surface code. Quantum-double generalizations of such strings are called ribbon operators. In Sec. 4, we project ribbon operators into the bulk ground-state subspace, obtaining 1D "flattened" ribbons that behave similar to ordinary ribbons in terms of carrying anyonic excitations, but, notably, fit into the 1D space of the flux ladder. We use flattened ribbons to express the flux ladder, and superimpose them in a Jordan-Wigner-like operator construction. This construction differs from the Jordan-Wigner-like basis of Ref. [71], which is based on a different notion of locality that depends nontrivially on the properties of

| | $\cos(N\widehat{\phi})$ potential (63) | $\cos(\frac{2\pi}{N}\widehat{L})$ potential (85) |
|---|---|---|
| Quantum wires | Ferromagnetic regime | Superconducting pairing regime |
| Surface-code edge | Pure fluxes condense | Pure charges condense |
| Symmetry breaking | $\mathbf{Z}_N$-broken phase | $\mathbf{Z}_N$-unbroken phase |
| Representation theory | $\mathbf{U}(1)$ group $\to \mathbf{Z}_N$ subgroup | $\mathbf{U}(1) \to \mathbf{U}(1)/\mathbf{Z}_N$ subspace |

Table 1: Cosine potentials acting on a planar $\mathbf{U}(1)$ rotor, with angular position (momentum) operator $\widehat{\phi}$ ($\widehat{L}$). The $\cos(N\widehat{\phi})$ potential is minimized at positions $\phi = \frac{2\pi}{N}k$ with $k \in \mathbf{Z}_N$. The $\cos(\frac{2\pi}{N}\widehat{L})$ potential is minimized in the subspace of states whose momentum is a multiple of $N$; this subspace is equivalent to the quotient space $\mathbf{U}(1)/\mathbf{Z}_N$. The two potentials are minimized at the same subspaces as the ferromagnetic and superconducting potentials of a quantum wire, respectively (see Sec. 6.2). Our representation-theoretic identification yields generalizations of these effects for systems with non-Abelian symmetries.

$\mathbf{G}$. Using our construction, we introduce candidate generalized zero modes for the dihedral group case $\mathbf{G} = \mathbf{D}_N$. We leave the study of stability of these zero modes — a subtle issue for any non-Majorana ($\mathbf{G} \neq \mathbf{Z}_2$) zero modes [47, 71, 87, 88] — to future work.

**Continuum description** Analogues of parafermion zero modes in the continuum have been identified [14–17, 23, 38] in variants of the free-boson field theory that describes the $\mathbf{Z}_N$ clock model [89–91]. The original clock model can then be recovered by pinning the $\mathbf{U}(1)$ degree of freedom of the free boson to values in $\mathbf{Z}_N$. Conversely, the field theory can be obtained by *embedding* each $\mathbf{Z}_N$-valued site of the lattice model into a $\mathbf{U}(1)$ degree of freedom and taking the continuum limit. In Sec. 5, we extend this procedure by embedding each $\mathbf{G}$-valued site of the flux ladder into a site taking values in a Lie group $\mathbf{\Gamma} \supset \mathbf{G}$. The resulting continuum field theory — a generalized $\mathbf{\Gamma}$ principal chiral model [81–83] — should be sufficiently versatile to admit continuum low-energy descriptions of various quantum-double edges, defects between them, and any phase transitions.

From a quantum information perspective, our embedding procedure can be thought of as the opposite of digitization. Just like a bit can be obtained by digitizing an analog voltage signal, each qunit of the surface code can be obtained from a properly discretized field. Such discretization is done not only at the spatial level, where the continuum is discretized into a lattice, but also at each spatial point, where the $\mathbf{U}(1)$-valued field is restricted to values in $\mathbf{Z}_N$ that delineate each qunit. The $\mathbf{G}$-valued qunits of quantum double models are more complex "digital" structures associated with discrete groups, and our procedure identifies continuous fields that, if appropriately discretized, naturally yield such structures.

**Quantum wire constructions** Proposals [14–17, 23] for realizing parafermions rely on the language of bosonization/fermionization [19–21, 92] to describe fractionalized edge states in terms of continuum fermionic degrees of freedom making up a quantum wire. Continuum analogues of parafermion zero modes are purported to lie at the interface between 'superconducting' and 'ferromagnetic' phases for the edge (or, more generally, at the boundary between two incompatibly gapped edge regions).

A necessary ingredient for producing continuum zero modes is the field-pinning mechanism used to generate the two competing interfacial phases. We show that the superconducting-phase pinning mechanism is equivalent to restricting the $\mathbf{U}(1)$ field of the bosonized theory to values in $\mathbf{Z}_N$, while the ferromagnetic-phase pinning mechanism is equivalent to projecting the field into the quotient space $\mathbf{U}(1)/\mathbf{Z}_N$ at each spatial point (see Table 1). This interpretation yields a natural extension to the non-Abelian case.

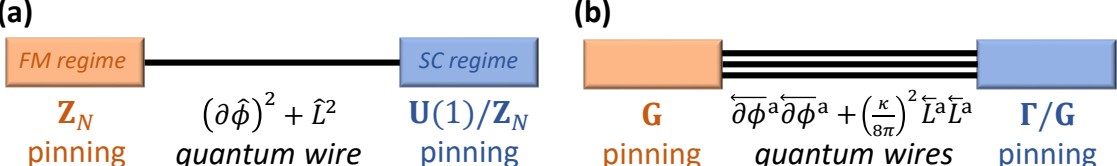

Figure 2: (a) Continuum Majorana and parafermion zero modes lie at the interface of the superconducting and ferromagnetic phases of the free boson field theory, recast as a fermionic quantum wire. The effect of the two phases amounts to pinning the fields at the ends of the quantum wire to particular values. We show that the ferromagnetic-phase pinning mechanism is equivalent to restricting the configuration space $U(1)$ of the field $\widehat{\phi}$ to the subgroup space $Z_N$, while the superconducting-phase pinning mechanism is equivalent to restricting $U(1)$ to the quotient subspace $U(1)/Z_N$. (b) In a representation-theoretic extension, we conjecture that generalized zero modes lie at interfaces between coupled quantum wires associated with a Lie group $\Gamma$, whose fields are subject to subgroup ($G$) and quotient-space ($\Gamma/G$) boundary conditions at the ends. Other boundary conditions based on gapped quantum-double edges can also be implemented.

In Sec. 6, we develop potentials pinning the $\Gamma$-valued field of the principal chiral model to either values in a subgroup $G$ or a quotient space $\Gamma/G$. We also convert a particular instance of the principal chiral model to a set of coupled quantum wires via addition of a Wess-Zumino-Novikov-Witten (WZNW) term [81,93–96] and subsequent non-Abelian bosonization [95,97–99]. Our connection of the flux ladder to the WZNW model (via the intermediate mapping to the principal chiral model) generalizes the connection of the clock model to a quatum wire (via the free boson), while the subgroup and quotient space potentials generalize, respectively, the superconducting and ferromagnetic pinning mechanisms. Continuing with this analogy, we conjecture that the WZNW theory, lying at the interface between $G$- and $\Gamma/G$-pinned regions, houses continuum topological zero-modes associated with quantum doubles (see Fig. 2).

While we leave verification of our conjecture to future work, we provide additional evidence for this idea by showing that the $G \hookrightarrow \Gamma$ embedding procedure used to obtain the theory retains topological properties in two other related examples — the ($G = Z_N$) surface-code and ($D_N$) quantum-double bulk — in Appx. D. We picked those subgroups to be explicit, and note that the embedding procedure can be applied to any $G$. Embedding those models, respectively, yields abelian [$\Gamma = U(1)$] and non-Abelian [$SO(3)$] Yang-Mills gauge theory [52,82,100,101] in the continuum limit. Both are known to admit the same low-energy topological excitations as the anyons of the original lattice models [102–106].

## 3  Effective theory for the quantum-double edge

Our effective edge theory is obtained from a 2D quantum double model on a disk via the two steps shown in Fig. 3: (1) molding the lattice into a bicycle wheel and (2) projecting the edge into the bulk ground-state subspace. In this bicycle-wheel geometry, the bulk ground-state constraint is realized on the edge as a projector onto the $+1$ eigenspace of a global symmetry. We first review the $Z_N$ derivation in a way that allows straightforward extension to general $G$.

### 3.1  Surface-code edge

Consider a qu$N$it with $N$ prime, "position" states $\{|k\rangle, k \in Z_N\}$, and corresponding generalized Pauli matrices $\widehat{X}, \widehat{Z}$. Powers of $\widehat{X}$ shift the qunit's position (modulo $N$), while $\widehat{Z}$ labels the

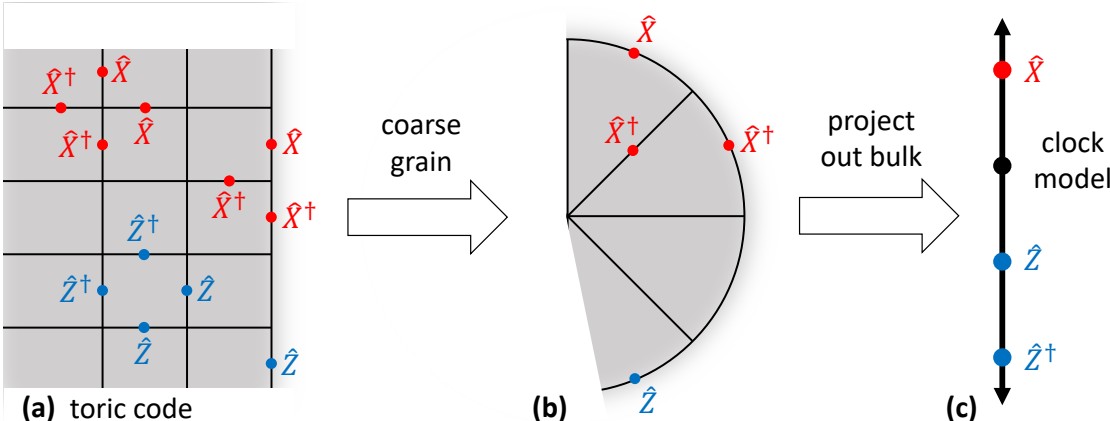

Figure 3: (a) A portion of the surface code model (3) on a square lattice with vertical edge. The $Z$-type plaquettes and $X$-type stars in the bulk and on the edge are denoted by blue and red, respectively. (b) The resulting course-grained model and its edge operators on a "bicycle-wheel" lattice. (c) The effective edge model after projection into the bulk ground-state subspace. For the quantum double model (18), the setup is the same if $\widehat{Z} \to \widehat{Z}$, $\widehat{X}^\dagger \to \overleftarrow{X}$, and $\widehat{X} \to \overrightarrow{X}$.

position states: with $h, \lambda \in \mathbf{Z}_N$,

$$\widehat{X}_h|k\rangle = |k + h\rangle \qquad \text{and} \qquad \widehat{Z}_\lambda|k\rangle = e^{i\frac{2\pi}{N}\lambda k}|k\rangle. \tag{1}$$

Above, we have put the power labels $h, \lambda$ into the subscript for later convenience. These satisfy the well-known Weyl commutation relation [107],

$$\widehat{X}_h\widehat{Z}_\lambda\widehat{X}_h^\dagger = e^{-i\frac{2\pi}{N}\lambda h}\widehat{Z}_\lambda. \tag{2}$$

The $\mathbf{Z}_N$ surface code Hamiltonian on a 2D square lattice consists of two types of four-body interactions — *Z-type plaquette* terms and *X-type star* terms,

$$\widehat{H}_{\mathbf{Z}_N}^{\text{bulk}} = -\frac{1}{N}\sum_{\square}\sum_{\lambda \in \mathbf{Z}_N} \widehat{Z}_\lambda^{\,\square}\, \widehat{Z}_\lambda^{\,\square}\, \widehat{Z}_\lambda^{\dagger\,\square}\, \widehat{Z}_\lambda^{\dagger\,\square} \;-\; \frac{1}{N}\sum_{+}\sum_{h \in \mathbf{Z}_N} \widehat{X}_h^{\dagger\,+}\, \widehat{X}_h^{+}\, \widehat{X}_h^{\dagger\,+}\, \widehat{X}_h^{\dagger\,+}, \tag{3}$$

where, e.g., the site $\square$ is the site at the bottom of a plaquette $\square$, and $+$ is the site at the right of a star $+$. The prefactors $1/N$ make sure that the terms are commuting projectors. The $N = 2$ case reduces to the usual qubit surface code; $N > 2$ is a straightforward extension.

The most general translationally-invariant edge Hamiltonian,

$$\widehat{H}_{\mathbf{Z}_N}^{\text{edge}} = \sum_{\square}\sum_{\lambda \in \mathbf{Z}_N} \mathsf{C}_\lambda \widehat{Z}_\lambda^{\,\square} \;+\; \sum_{\dashv}\sum_{h \in \mathbf{Z}_N} \mathsf{f}_h \widehat{X}_h^{\dagger\,\dashv}\, \widehat{X}_h^{\dagger\,\dashv}\, \widehat{X}_h^{\dashv}, \tag{4}$$

consists of arbitrary linear combinations of one-body $Z$-type truncated plaquettes $\square$ and three-body $X$-type truncated stars $\dashv$ lying along the vertical edge depicted in Fig. 3(a). Both types of edge operators commute with the bulk stars and plaquettes, but do not commute with each other. The combined Hamiltonian, $\widehat{H}_{\mathbf{Z}_N}^{\text{bulk}} + \widehat{H}_{\mathbf{Z}_N}^{\text{edge}}$, may thus not be gapped, and one can consider gapped edges by excluding either the truncated stars or plaquettes [8]. We will not restrict ourselves to particular edges (i.e., gapped or otherwise) for now in order to extract a general edge theory, and as such will specify the parameters $\{\mathsf{C}_\lambda, \mathsf{f}_h\}$ later.

The first step involves molding the lattice from its original square form into a bicycle wheel, whose "tire" corresponds to the system's edge and whose "spokes" correspond to the bulk. Such

molding is done via a series of "local moves" [75, 76] which preserve the bulk $\mathbf{Z}_N$ topological order. A local move allows one to incorporate new vertices into the topological ground state via local unitary transformations. For example, to split a plaquette into two triangular pieces ($\square \to \boxslash$), one initializes a fresh site at the plaquette's center to $|h = 0\rangle$, and performs a series of CNOT gates,

$$\text{CNOT} = \sum_{h \in \mathbf{Z}_N} |h\rangle\langle h| \otimes \widehat{X}_h \,, \tag{5}$$

in order to extend the ground state of the system onto that site. Such gates can also add an edge to a star ($\times \to \times\!\!\!\prec$); we will describe such a move in more detail in Sec. 3.4. Both moves are reversible, and applying them backwards (e.g., $\boxslash \to \square$) disentangles unneeded edges.

Local moves can morph the square lattice into a bicycle-wheel geometry shown in Fig. 3(b). The bulk plaquettes on this geometry correspond to three-body terms $\widehat{Z}_\lambda^{\triangleleft} \equiv \widehat{Z}_\lambda \widehat{Z}_\lambda \widehat{Z}_\lambda^\dagger$, acting on two spokes and their joint section of tire. There is only one bulk star — the nonlocal operator $\widehat{X}^{\text{(global)}} \equiv \cdots \widehat{X}_h \widehat{X}_h \cdots$, consisting of a product of $\widehat{X}$ operators on all of the spokes. The edge terms are converted to three-body truncated stars $\widehat{X}^\dagger \widehat{X}^\dagger \widehat{X}$, acting on a spoke and the two sections of tire on either side of the spoke, and a one-body truncated plaquette $\widehat{Z}$ on each section of tire. The bulk and edge operators satisfy the same algebra as those on the original square lattice.

From now on, we assume that there are no bulk excitations, working in the subspace where the bulk plaquettes and star are unity for all $\lambda, h$,

$$\widehat{Z}_\lambda^{\triangleleft} \equiv \widehat{Z}_\lambda \widehat{Z}_\lambda \widehat{Z}_\lambda^\dagger = 1 \qquad \text{and} \qquad \widehat{X}_h^{\text{(global)}} \equiv \cdots \widehat{X}_h \widehat{X}_h \cdots = 1 \,. \tag{6}$$

We proceed to project the edge Hamiltonian onto the subspace defined by these constraints. First, we introduce a spoke index $\mathsf{x}$ that labels an effective spatial site within the bulk ground-state subspace on the edge. We use sans serif font for spatial indices from now on. Next, we define

$$\widehat{Z}^{(\mathsf{x})} \equiv \widehat{Z} \qquad \text{and} \qquad \widehat{X}^{(\mathsf{x})} \equiv \widehat{X}^\dagger \widehat{X}^\dagger \widehat{X} \,. \tag{7}$$

The $\widehat{Z}^{(\mathsf{x})}$ term acts on the spoke $\mathsf{x}$, while $\widehat{X}^{(\mathsf{x})}$ acts on the spoke and the two sections of tire to either side of it. The edge plaquette is equivalent to the product $\widehat{Z}^{\dagger(\mathsf{x}-1)}\widehat{Z}^{(\mathsf{x})}$ via multiplication of the latter by the bulk stabilizer $\widehat{Z}^{\triangleleft}$, constrained to be 1 by Eq. (6),

$$\widehat{Z}^{\triangleleft} = (\widehat{Z}^\dagger\widehat{Z})^{\triangleleft} \widehat{Z}^{\triangleleft} (\widehat{Z}^\dagger\widehat{Z})^{\triangleleft} = \widehat{Z}^{\dagger} \widehat{Z}^{\triangleleft} \widehat{Z} = \widehat{Z}^{\dagger} \widehat{Z} = \widehat{Z}^{\dagger(\mathsf{x}-1)}\widehat{Z}^{(\mathsf{x})} \,. \tag{8}$$

Applying the substitution (8) and the definition of $\widehat{X}^{(\mathsf{x})}$ (7) converts the general edge Hamiltonian $\widehat{H}_{\mathbf{Z}_N}^{\text{edge}}$ (4) into the clock model. We redefine

$$\widehat{H}_{\mathbf{Z}_N}^{\text{edge}} = -\sum_{\mathsf{x}} \sum_{\lambda \in \mathbf{Z}_N} \mathsf{C}_\lambda \widehat{Z}_\lambda^{\dagger(\mathsf{x})}\widehat{Z}_\lambda^{(\mathsf{x}+1)} + \sum_{h \in \mathbf{Z}_N} \mathsf{f}_h \widehat{X}_h^{(\mathsf{x})} \,. \tag{9a}$$

It will be useful for our purposes to express the clock model in another way,

$$\widehat{H}_{\mathbf{Z}_N}^{\text{edge}} = -\sum_{\mathsf{x}} \sum_{h \in \mathbf{Z}_N} \mathsf{c}_h \widehat{\Pi}_h^{(\mathsf{x},\mathsf{x}+1)} + \mathsf{f}_h \widehat{X}_h^{(\mathsf{x})} \,, \tag{9b}$$

where the two-body term has been converted into a sum of projections,

$$\widehat{\Pi}_h^{(\mathsf{x},\mathsf{y})} = \frac{1}{N} \sum_{\lambda \in \mathbf{Z}_N} e^{-i\frac{2\pi}{N}h\lambda} \widehat{Z}_\lambda^{\dagger(\mathsf{x})}\widehat{Z}_\lambda^{(\mathsf{y})} \,, \tag{10}$$

onto the subspace spanned by $|k_x, k_y\rangle$ with $k_y - k_x = h$ modulo $N$. The coefficients can likewise be interconverted via the Fourier transform

$$c_h = \sum_{\lambda \in \mathrm{irr}(\mathbf{Z}_N)} C_\lambda e^{i \frac{2\pi}{N} \lambda h} \qquad \leftrightarrow \qquad C_\lambda = \frac{1}{N} \sum_{h \in \mathbf{Z}_N} c_h e^{-i \frac{2\pi}{N} \lambda h} . \tag{11}$$

The $c_h$ are real because $\widehat{\Pi}$ are Hermitian, and the Fourier transform pins down the general form of $C_\lambda$.

The model admits a $\mathbf{Z}_N$ symmetry, being invariant under $\widehat{X}_h^{(\mathrm{global})}$, which rotates each site $\mathsf{x}$ by $\widehat{X}_h$. Two important edge cases of the model's parameter space correspond to $\{C_\lambda = 1/N, f_h = 0\}$ and $\{C_\lambda = 0, f_h = 1/N\}$, respectively. The former pure-$ZZ$ case favors alignment of position states at neighboring sites, yielding $N$ symmetry-broken ground states $\cdots|h\rangle|h\rangle|h\rangle\cdots$. The latter pure-$X$ case yields a unique disordered ground state that is a tensor product of an equal superposition of position states $|\mathbf{Z}_N\rangle = \frac{1}{\sqrt{N}} \sum_{h \in \mathbf{Z}_N} |h\rangle$.

When treating the model as an edge theory of a surface-code bulk, we need to restrict ourselves to a subspace of the global spin-chain Hilbert space that corresponds to there being no bulk charges. Applying the star constraint (6), this subspace is such that

$$\widehat{X}_h^{(\mathrm{global})} = \prod_{\mathsf{x}} \widehat{X}_h^{(\mathsf{x})} = 1 \qquad \text{for all } h \in \mathbf{Z}_N . \tag{12}$$

This global symmetry is enforced by the bulk being in a topologically ordered ground state. The presence of bulk excitations corresponds to projecting into other sectors of the global space as well as different boundary conditions (see Appx. A).

Anyonic strings carrying bulk excitations that touch a gapped edge can affect the state of the edge theory. Those anyons that do not raise the energy when acting on a clock-model ground state are said to *condense*, and those that do are said to be *confined* [8,108]. For example, one type of gapped edge corresponds to the symmetry-breaking case $\{C_\lambda = 1/N, f_h = 0\}$, in which the edge ground state in the no-bulk-excitation sector is an equal superposition of all symmetry-breaking states $\cdots|h\rangle|h\rangle|h\rangle\cdots$. One end of an anyonic string carrying a charge anyon $\lambda$ is represented by $\widehat{Z}_\lambda^{(\mathsf{x})}$. Acting on the ground state, $\widehat{Z}_\lambda^{(\mathsf{x})}$ imparts relative phases between the symmetry-breaking states, but does not take the state out of the ground-state subspace. On the other hand, strings carrying flux excitations $h$, represented by a product of $\widehat{X}_h$ operators acting on a segment of the chain, convert the ground state into a superposition of domain-wall excitations. We thus conclude that, at this gapped edge, charges condense while fluxes are confined. Conversely, the disordered phase $\{C_\lambda = 0, f_h = 1/N\}$ of the clock model confines charges and condenses fluxes, and there is a phase transition between the two.

## 3.2 Group space

Like the surface code, the quantum double model for a general non-Abelian discrete group $\mathbf{G}$ is also defined on a lattice of finite-dimensional qunits. However, its constituent operators are best described not by the conventional Pauli matrices, but by their extensions associated with $\mathbf{G}$. Here we develop these operators and use them to write down the model.

Given any finite group $\mathbf{G}$ of size $|\mathbf{G}|$, the group elements $g \in \mathbf{G}$ can serve as labels for the basis states of a $|\mathbf{G}|$-dimensional qunit, $\{|g\rangle, g \in \mathbf{G}\}$. The "$X$-type" operators associated with this space will be labeled by group elements, and will form "regular" representations of the group acting on itself via permutation [109]. Since $\mathbf{G}$ is generally non-Abelian, such matrices have two ways to act — by multiplication from the left or right. For any $g, h \in \mathbf{G}$,

$$\overrightarrow{X}_h |g\rangle = |hg\rangle \qquad \text{and} \qquad \overleftarrow{X}_h |g\rangle = |gh^{-1}\rangle . \tag{13}$$

Being a representation, each set of multipliers satisfies the group's multiplication properties, $\overleftarrow{X}_h \overleftarrow{X}_k = \overleftarrow{X}_{hk}$ for all $h, k \in \mathbf{G}$ (and same for $\overrightarrow{X}$). The "left" and "right" regular representation operators commute, $\overleftarrow{X}_h \overrightarrow{X}_k = \overrightarrow{X}_k \overleftarrow{X}_h$.

The "$Z$-type" matrices that we use are diagonal in the $\{|g\rangle\}$ basis. To construct them, we use $\mathcal{Z}_\lambda(g)$ — the *irreducible representation* (or *irrep*) $\lambda$ of the group, evaluated at group elements $g$. This $\mathcal{Z}_\lambda$ reduces to a root of unity scalar for $\mathbf{G} = \mathbf{Z}_N$, but can be a matrix for non-Abelian $\mathbf{G}$. These also satisfy the group multiplication properties, i.e., $\mathcal{Z}(g).\mathcal{Z}(h) = \mathcal{Z}(gh)$, where we denote the internal matrix product by ".". Their matrix elements $[\mathcal{Z}_\lambda(g)]_{mn}$ can be used to construct $Z$-type qunit operators $\sum_{g \in \mathbf{G}} [\mathcal{Z}_\lambda(g)]_{mn} |g\rangle\langle g|$. We combine such qunit operators into the group-valued $Z$-type operator

$$\widehat{\mathcal{Z}}_\lambda = \sum_{g \in \mathbf{G}} \mathcal{Z}_\lambda(g) \otimes |g\rangle\langle g|, \qquad \text{acting as} \qquad \widehat{\mathcal{Z}}_\lambda |g\rangle = \mathcal{Z}_\lambda(g) \otimes |g\rangle, \qquad (14)$$

where the first factor in the tensor product acts on the *internal space* of the irrep $\lambda$, and the second on the qunit space. We abuse notation and omit "$\otimes$" from now on. However, the reader should keep in mind that one has to take matrix elements of $\widehat{\mathcal{Z}}_\lambda$ within the internal space to obtain a bona-fide operator acting exclusively on the qunit space.

Grouping $Z$-operators into irrep matrices acting on an internal space allows us to omit the matrix-element indices $m, n$ associated with the irrep. The resulting composite operators are natural analogues of group-valued fields considered later in the paper, and provide a concise way to generalize relations satisfied by the Pauli operators $\widehat{X}, \widehat{Z}$. For example, one can write a Weyl-type relation [cf. (2)] that holds for all $\mathbf{G}$:

$$\overrightarrow{X}_g \widehat{\mathcal{Z}}_\lambda \overrightarrow{X}_g^\dagger = \sum_{h \in \mathbf{G}} \mathcal{Z}_\lambda(g^{-1}h) \otimes |h\rangle\langle h| = \mathcal{Z}_\lambda(g^{-1}).\widehat{\mathcal{Z}}_\lambda \tag{15a}$$

$$\overleftarrow{X}_g \widehat{\mathcal{Z}}_\lambda \overleftarrow{X}_g^\dagger = \widehat{\mathcal{Z}}_\lambda.\mathcal{Z}_\lambda(g). \tag{15b}$$

**Dihedral case**    Let us introduce the above objects for a concrete group. The dihedral group $\mathbf{G} = \mathbf{D}_N$ for $N > 2$ consists of $|\mathbf{D}_N| = 2N$ elements. Each element can be expressed as a product of powers of two generators (see Appx. B), a "rotation" $r$ and "reflection" $p$. These satisfy the group's defining relations, $r^N = p^2 = 1$ and $prp = r^{-1}$.

The two sets of dihedral operators from Eq. (13), $\{\overleftarrow{X}\}$ and $\{\overrightarrow{X}\}$, consist of $2N$-dimensional permutation matrices. Each set forms a representation of the group. We express these and the $Z$-type operators discussed below in terms of qunit and qubit Pauli operators in Appx. B.

To introduce the dihedral $Z$-operators, we first enumerate the irreps $\lambda \in \mathrm{irr}(\mathbf{D}_N)$ and their corresponding matrices $\mathcal{Z}_\lambda$. The one-dimensional irreps include the trivial irrep $\lambda = \mathbf{1}$ $[\mathcal{Z}_\mathbf{1}(r) = \mathcal{Z}_\mathbf{1}(p) = 1]$, the sign irrep $\mathbf{1}'$ $[\mathcal{Z}_{\mathbf{1}'}(r) = 1$ and $\mathcal{Z}_{\mathbf{1}'}(p) = -1]$, and, for $N$ even, two more irreps in which $r \to -1$ and $p \to \pm 1$. Since $\mathbf{D}_N$ is non-Abelian, it also admits two-dimensional irreps $\lambda = \mathbf{2}_n$, with $n \in \{1, 2, \cdots, \lfloor (N-1)/2 \rfloor\}$. While the basis used for these irreps does not affect our results, we pick a basis in which $\mathcal{Z}(r)$ is diagonal for demonstration purposes:

$$\mathcal{Z}_{\mathbf{2}_n}(r) \equiv \begin{bmatrix} e^{i\frac{2\pi}{N}n} & 0 \\ 0 & e^{-i\frac{2\pi}{N}n} \end{bmatrix} \qquad \text{and} \qquad \mathcal{Z}_{\mathbf{2}_n}(p) \equiv \begin{bmatrix} 0 & 1 \\ 1 & 0 \end{bmatrix}. \tag{16}$$

From now on, we denote matrices acting on the internal space with square brackets "$[\cdots]$". When acting on a basis state $|g\rangle$, each dihedral $Z$-operator (14) yields the matrix representation of the group element $g$. For the case of the generating element $g = r$,

$$\widehat{\mathcal{Z}}_{\mathbf{2}_n} |r\rangle = \mathcal{Z}_{\mathbf{2}_n}(r) |r\rangle = \begin{bmatrix} e^{i\frac{2\pi}{N}n} & 0 \\ 0 & e^{-i\frac{2\pi}{N}n} \end{bmatrix} |r\rangle. \tag{17}$$

Tracing out the irrep space yields an operator on the group space, $\mathrm{TR}[\widehat{\mathcal{Z}}_{\mathbf{2}_n}]|r\rangle = 2\cos(\frac{2\pi}{N}n)|r\rangle$.

### 3.3 Quantum double model

As with the surface code, the quantum-double model Hamiltonian on a 2D square lattice can be expressed in terms of four-body $Z$-type plaquettes and $X$-type stars,

$$\widehat{H}_{\mathbf{G}}^{\text{bulk}} = -\frac{1}{|\mathbf{G}|}\sum_{\square}\sum_{\lambda\in\text{irr}(\mathbf{G})} d_\lambda \text{TR}\big[\widehat{\mathcal{Z}}_\lambda^{\,\lrcorner}.\widehat{\mathcal{Z}}_\lambda^{\,\square}.\widehat{\mathcal{Z}}_\lambda^{\dagger\,\square}.\widehat{\mathcal{Z}}_\lambda^{\dagger\,\square}\big] - \frac{1}{|\mathbf{G}|}\sum_{+}\sum_{g\in\mathbf{G}} \overrightarrow{X}_g^{\,\dagger}\,\overrightarrow{X}_g^{\,\dagger}\,\overleftarrow{X}_g^{\,\dagger}\,\overleftarrow{X}_g^{\,\dagger}\,, \tag{18}$$

where $d_\lambda$ is the dimension of irrep $\lambda$. The entire setup is the same as that in Fig. 3(b) with $\widehat{Z}\to\widehat{\mathcal{Z}}$, $\widehat{X}^\dagger\to\overleftarrow{X}$, $\widehat{X}\to\overrightarrow{X}$, and addition of the trace "TR" over the internal irrep space. All plaquettes are diagonal in the group basis, and hence commute with each other. The stars commute with each other since $\overleftarrow{X}$ and $\overrightarrow{X}$ commute. Using the Weyl-type relations (15), one can verify that plaquettes and stars commute with each other. For example, for $\widehat{\mathcal{Z}}^\square \equiv \widehat{\mathcal{Z}}_\lambda^{\,\lrcorner}.\widehat{\mathcal{Z}}_\lambda^{\,\square}.\widehat{\mathcal{Z}}_\lambda^{\dagger\,\square}.\widehat{\mathcal{Z}}_\lambda^{\dagger\,\square}$ and $\widehat{X}_g^+ \equiv \overrightarrow{X}_g^{\,\dagger}\,\overrightarrow{X}_g^{\,\dagger}\,\overleftarrow{X}_g^{\,\dagger}\,\overleftarrow{X}_g^{\,\dagger}$ intersecting at $\lrcorner$ and $\square$ ,

$$\widehat{X}_g^+ \text{TR}\big[\widehat{\mathcal{Z}}^\square\big](\widehat{X}_g^\dagger)^+ = \text{TR}\big[\mathcal{Z}(g^{-1}).\widehat{\mathcal{Z}}^{\,\lrcorner}.\widehat{\mathcal{Z}}^{\,\square}.\widehat{\mathcal{Z}}^{\dagger\,\square}.\widehat{\mathcal{Z}}^{\dagger\,\square}.\mathcal{Z}(g)\big] = \text{TR}\big[\widehat{\mathcal{Z}}^\square\big]. \tag{19}$$

Similar to the surface-code edge Hamiltonian $\widehat{H}_{\mathbf{Z}_N}^{\text{edge}}$ (4), the most general quantum-double edge Hamiltonian $\widehat{H}_{\mathbf{G}}^{\text{edge}}$ on the square lattice of Fig. 3 consists of three-body truncated stars as well as one-body truncated plaquettes.

The plaquette terms are "flux" constraints on the group states $|g^\square\rangle = |g^{\lrcorner}, g^{\square}, g^{\square}, g^{\square}\rangle$, projecting onto the subspace where the product $\nabla g^\square \equiv g^{\lrcorner}\, g^{\square}\, (g^{-1})^{\square}\, (g^{-1})^{\square} = 1$. We have written them in terms of $Z$-type operators [101], but they can be converted into their original form [1] using

$$\sum_{\lambda\in\text{irr}(\mathbf{G})}\frac{d_\lambda}{|\mathbf{G}|}\text{TR}\big[\widehat{\mathcal{Z}}_\lambda^\square\big] = \sum_{\lambda\in\text{irr}(\mathbf{G})}\frac{d_\lambda}{|\mathbf{G}|}\sum_{g^\square\in\mathbf{G}^{\times 4}}\text{TR}\big[\mathcal{Z}_\lambda\big(\nabla g^\square\big)\big]\,|g^\square\rangle\langle g^\square| \tag{20a}$$

$$= \sum_{g^\square\in\mathbf{G}^{\times 4}}\delta_{\nabla g^\square,1}^{\mathbf{G}}\,|g^\square\rangle\langle g^\square| = \sum_{\nabla g^\square=1}|g^\square\rangle\langle g^\square|\,. \tag{20b}$$

The Kronecker $\delta$-function on the group follows from the group orthogonality relations [109], valid for any elements $g, h \in \mathbf{G}$,

$$\delta_{g,h}^{\mathbf{G}} = \sum_{\lambda\in\text{irr}(\mathbf{G})}\frac{d_\lambda}{|\mathbf{G}|}\text{TR}\big[\mathcal{Z}_\lambda(g^{-1}).\mathcal{Z}_\lambda(h)\big] = \begin{cases}1 & g = h \\ 0 & \text{otherwise}\end{cases}. \tag{21}$$

For $\mathbf{G} = \mathbf{Z}_2$ with addition as the operation, we have $\nabla h^\square = h^{\lrcorner} + h^{\square} - h^{\square} - h^{\square}$, and Eq. (20) reduces to $\frac{1}{2}\big(1 \pm \widehat{\sigma}_z^\square\big) = \sum_{\nabla h^\square=0}|h^\square\rangle\langle h^\square|$.

### 3.4 General edges: the flux ladder

Molding a quantum-double model lattice [24,77,78] while preserving the $\mathbf{G}$ topological order proceeds via similar local moves as the surface code case, but with subtleties stemming from the non-Abelian nature of the group. For example, there are now two types of conditional-rotation (CROT) gates,

$$\overrightarrow{\text{CROT}} = \sum_{g\in\mathbf{G}}|g\rangle\langle g|\otimes \overrightarrow{X}_g \qquad \text{and} \qquad \overleftarrow{\text{CROT}} = \sum_{g\in\mathbf{G}}|g\rangle\langle g|\otimes \overleftarrow{X}_g\,, \tag{22}$$

and such gates have to be applied in a certain order because the target operations no longer all commute. Nevertheless, $\widehat{\mathcal{Z}}$ operators (14) prove to be a useful analogue of $Z$-type stabilizers

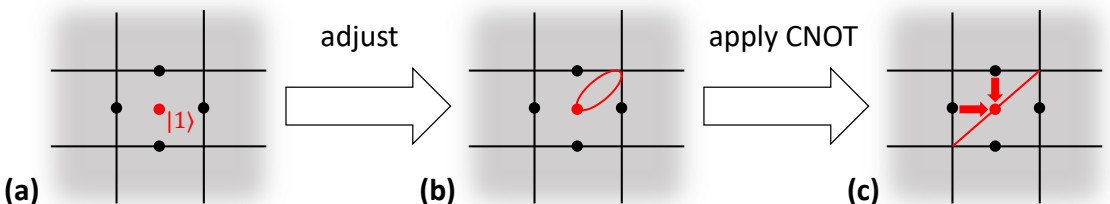

Figure 4: Sketch of the steps required for the procedure $\square \to \boxslash$, which splits a square plaquette into two triangular plaquettes. (a) Initial square quantum-double lattice, with a fresh decoupled site $\boxdot$ inside the central plaquette initialized in $|h = 1\rangle$. (b) Intermediate configuration depicting "adjusted" plaquettes and stars. (c) Final configuration after application of two $\overrightarrow{\mathrm{CROT}}$ gates, depicted by the red arrows.

from conventional error correction, allowing for straightforward extraction of the flux ladder from the quantum-double edge.

As a demonstration, we elaborate on the step $\square \to \boxslash$, which splits the central plaquette in Fig. 4(a) into two triangular plaquettes; vertex addition $\times \to \bowtie$ can be done in similar fashion. First, introduce a site $\boxdot$ at the plaquette's center, initializing its state to $|g = 1\rangle$ and adding the one-body plaquette term $\mathrm{TR}[\widehat{\mathcal{Z}}^{\boxdot}]$ to the Hamiltonian. Next, this non-Abelian case requires "adjusting" some stars and plaquettes in a way that doesn't change the bulk ground state, but that prepares the Hamiltonian for the $\overrightarrow{\mathrm{CROT}}$ gates. Noting that $|1\rangle = \overrightarrow{X}_g \overleftarrow{X}_g |1\rangle$, the upper-right star in Fig. 4(a) is adjusted by multiplying each of its four-body terms by $(\overrightarrow{X}_g \overleftarrow{X}_g)^{\boxdot}$. As for the plaquettes, the constraint $\mathrm{TR}[\widehat{\mathcal{Z}}_\lambda^{\boxdot}]|1\rangle = d_\lambda |1\rangle$ and group orthogonality relations (21) imply that $\widehat{\mathcal{Z}}_\lambda^{\boxdot} = 1_\lambda$ for any irrep $\lambda$. The central plaquette is then extended to the five-body term

$$\mathrm{TR}\big[\widehat{\mathcal{Z}}_\lambda^{\Box}.\widehat{\mathcal{Z}}_\lambda^{\Box}.1_\lambda.\widehat{\mathcal{Z}}_\lambda^{\dagger\Box}.\widehat{\mathcal{Z}}_\lambda^{\dagger\Box}\big] = \mathrm{TR}\big[\widehat{\mathcal{Z}}_\lambda^{\Box}.\widehat{\mathcal{Z}}_\lambda^{\Box}.\widehat{\mathcal{Z}}_\lambda^{\dagger\Box}.\widehat{\mathcal{Z}}_\lambda^{\dagger\Box}.\widehat{\mathcal{Z}}_\lambda^{\dagger\Box}\big]. \tag{23}$$

This yields the intermediate configuration depicted in Fig. 4(b).

Finally, one applies the sequence $\overrightarrow{\mathrm{CROT}}^{\Box} \overrightarrow{\mathrm{CROT}}^{\Box}$, with $\boxdot$ being the target in both gates. The five-body term (23) becomes the lower-triangular plaquette $\mathrm{TR}[\widehat{\mathcal{Z}}^{\Box}.\widehat{\mathcal{Z}}^{\Box}.\widehat{\mathcal{Z}}^{\dagger\Box}]$; the single-site term $\mathrm{TR}[\widehat{\mathcal{Z}}^{\boxdot}]$ becomes the upper-triangular plaquette $\mathrm{TR}[\widehat{\mathcal{Z}}^{\boxdot}.\widehat{\mathcal{Z}}^{\dagger\Box}.\widehat{\mathcal{Z}}^{\dagger\Box}]$. The star terms adjust accordingly, yielding the final configuration in Fig. 4(c).

After molding, we are once again left with a bicycle-wheel bulk Hamiltonian in Fig. 3(b), consisting of multiple three-body plaquettes and one nonlocal star. The zero-flux ground-state constraint is

$$\widehat{\mathcal{Z}}_\lambda^{\triangleleft} \equiv \widehat{\mathcal{Z}}_\lambda^{\triangleleft}.\widehat{\mathcal{Z}}_\lambda^{\triangleleft}.\widehat{\mathcal{Z}}_\lambda^{\dagger\triangleleft} = 1_\lambda, \tag{24}$$

valid for all irreps $\lambda$ with corresponding identities $1_\lambda$. The edge terms are three-body truncated stars $\overleftarrow{X}^{\rightarrow\bullet} \overleftarrow{X}^{\rightarrow\bullet} \overrightarrow{X}^{\rightarrow\bullet}$ and a one-body truncated plaquette term $\widehat{\mathcal{Z}}^{\triangleleft}$ on each section of tire.

We now project the edge Hamiltonian onto the bulk ground-state subspace. First, define the following terms,

$$\overleftarrow{X}^{(\times)} \equiv \overleftarrow{X}^{\rightarrow\bullet} \overleftarrow{X}^{\rightarrow\bullet} \overrightarrow{X}^{\rightarrow\bullet} \qquad \text{and} \qquad \widehat{\mathcal{Z}}^{(\times)} \equiv \widehat{\mathcal{Z}}^{\rightarrow\bullet}, \tag{25}$$

where $\times$ once again labels an effective site within the bulk ground-state subspace. Using the plaquette constraint (24),

$$\widehat{\mathcal{Z}}^{\triangleleft} = (\widehat{\mathcal{Z}}^{\dagger}.\widehat{\mathcal{Z}})^{\triangleleft}.\widehat{\mathcal{Z}}^{\triangleleft}.(\widehat{\mathcal{Z}}^{\dagger}.\widehat{\mathcal{Z}})^{\triangleleft} = \widehat{\mathcal{Z}}^{\dagger\triangleleft}.\widehat{\mathcal{Z}}^{\triangleleft}.\widehat{\mathcal{Z}}^{\triangleleft} = \widehat{\mathcal{Z}}^{\dagger\triangleleft}.\widehat{\mathcal{Z}}^{\triangleleft} = \widehat{\mathcal{Z}}^{\dagger(\times-1)}.\widehat{\mathcal{Z}}^{(\times)}. \tag{26}$$

This yields the ingredients for the effective edge theory.

The above manipulations yield the most general translationally-invariant edge model within the bulk ground-state subspace of **G** quantum-double topological order,

$$
\widehat{H}_{\mathbf{G}}^{\text{edge}} = -\sum_{\times} \sum_{\lambda \in \text{irr}(\mathbf{G})} \text{Tr}\!\left[ C_\lambda . \widehat{\mathcal{Z}}_\lambda^{\dagger(\times)} . \widehat{\mathcal{Z}}_\lambda^{(\times+1)} \right] + \sum_{g \in \mathbf{G}} f_g \overleftarrow{X}_g^{(\times)} . \tag{27a}
$$

This corresponds to a slightly generalized version of the *flux ladder* [71], and we stick with that name from now on. In this notation, the two-body and one-body terms closely resemble their clock-model counterparts. As such, the two-body $Z$-type term tries to align neighboring sites in some fashion that depends on the parameter matrices C, while the one-body $X$-type term tries to stabilize some quantum superposition of states $|g\rangle$ at each site that depends on the parameters f. We elaborate on notable special cases in the next subsections, and collect all cases we have identified for $\mathbf{G} = \mathbf{D}_N$ in Appx. B.

The parameters f satisfy $f_{g^{-1}} = f_g^\star$ in order for $\widehat{H}$ to be Hermitian. To extract a general expression for the matrices $C_\lambda$, it is useful to re-write the Hamiltonian as

$$
\widehat{H}_{\mathbf{G}}^{\text{edge}} = -\sum_{\times} \sum_{g \in \mathbf{G}} c_g \overleftarrow{\Pi}_g^{(\times,\times+1)} + f_g \overleftarrow{X}_g^{(\times)} , \tag{27b}
$$

where the $c_g$ are manifestly real because the two-body projection,

$$
\overleftarrow{\Pi}_g = \sum_{h \in \mathbf{G}} |h, hg\rangle\langle h, hg| , \tag{28}
$$

is Hermitian. This $\overleftarrow{\Pi}_g$ projects onto the subspace spanned by $|k, h\rangle$ such that $k^{-1}h = g$. Using group orthogonality (21), these projections can be used interchangeably with the $ZZ$-type operators,

$$
\overleftarrow{\Pi}_g^{(\times,\times+1)} = \sum_{\lambda \in \text{irr}(\mathbf{G})} \frac{d_\lambda}{|\mathbf{G}|} \text{Tr}\!\left[ \mathcal{Z}_\lambda(g^{-1}) . \widehat{\mathcal{Z}}_\lambda^{\dagger(\times)} . \widehat{\mathcal{Z}}_\lambda^{(\times+1)} \right] \tag{29a}
$$

$$
\widehat{\mathcal{Z}}_\lambda^{\dagger(\times)} . \widehat{\mathcal{Z}}_\lambda^{(\times+1)} = \sum_{g \in \mathbf{G}} \mathcal{Z}_\lambda(g) \overleftarrow{\Pi}_g^{(\times,\times+1)} . \tag{29b}
$$

There is likewise a "Fourier transform" for the coefficients:

$$
c_g = \sum_{\lambda \in \text{irr}(\mathbf{G})} \text{Tr}\!\left[ C_\lambda . \mathcal{Z}_\lambda(g) \right] \qquad \leftrightarrow \qquad C_\lambda = \frac{d_\lambda}{|\mathbf{G}|} \sum_{g \in \mathbf{G}} c_g \mathcal{Z}_\lambda(g^{-1}) . \tag{30}
$$

This pins down the general form of $C_\lambda$ — linear combinations of group elements $g$, evaluated at irrep $\lambda$ and multiplied by real coefficients that depend only on $g$. The C and f parameter sets each contain $|\mathbf{G}| - 1$ independent nontrivial degrees of freedom.

The flux-ladder model (27) has a global **G**-symmetry under

$$
\overrightarrow{X}_g^{(\text{global})} \equiv \cdots \overrightarrow{X}_g^{\overleftarrow{\phantom{.}}} \; \overrightarrow{X}_g^{\overleftarrow{\phantom{.}}} \cdots = \prod_{\times} \overrightarrow{X}_g^{(\times)} . \tag{31}
$$

All $\overleftarrow{X}$-operators in the one-body term commute with $\overrightarrow{X}_g^{(\text{global})}$, and the two-body term [recalling the Weyl-type relations (15)] transforms as

$$
\overrightarrow{X}_g^{(\text{global})} \widehat{\mathcal{Z}}^{\dagger(\times)} . \widehat{\mathcal{Z}}^{(\times+1)} \overrightarrow{X}_g^{\dagger(\text{global})} = \widehat{\mathcal{Z}}^{\dagger(\times)} . \mathcal{Z}(g^{-1}) . \mathcal{Z}(g) . \widehat{\mathcal{Z}}^{(\times+1)} = \widehat{\mathcal{Z}}^{\dagger(\times)} . \widehat{\mathcal{Z}}^{(\times+1)} , \tag{32}
$$

showing that the symmetry holds for arbitrary $\{C_\lambda, f_g\}$. Because this symmetry stems from a non-local excitation constraint on the quantum-double bulk, only symmetric operators can be considered as *physical* in this context. The bulk ground-state constraint translates to a restriction of the flux ladder to the collective subspace where $\overrightarrow{X}_g^{(\text{global})} = 1$ for all $h$. We consider more general bulk-excitation conditions in Appx. A.

## 3.5 Gapped edges

Quantum double models admit gapped edges, which are classified by a subgroup $\mathbf{K} \subseteq \mathbf{G}$ and its representations, which can be linear or projective [62]. In a projective representation, the group's multiplication table is decorated with phases in a way that is consistent with associativity. We consider only linear representations of $\mathbf{K}$ in this work, but an extension to the projective case should be straightforward.

A gapped edge described by $\mathbf{K}$ corresponds to the following instance of the flux ladder, written in terms of either $ZZ$-type operators or projections using Eqs. (21) and (29),

$$\widehat{H}_{\mathbf{K}\subseteq\mathbf{G}}^{\text{edge}} = -\sum_{\times} \sum_{\lambda\in\text{irr}(\mathbf{G})} \frac{d_\lambda}{|\mathbf{G}|/|\mathbf{K}|} \text{TR}\big[\Pi_{\mathbf{1}}^{\mathbf{K}}.\widehat{\mathcal{Z}}_\lambda^{\dagger(\times)}.\widehat{\mathcal{Z}}_\lambda^{(\times+1)}\big] + \frac{1}{|\mathbf{K}|}\sum_{k\in\mathbf{K}}\overleftarrow{X}_k^{(\times)} \tag{33a}$$

$$= -\sum_{\times} \frac{1}{|\mathbf{K}|}\sum_{k\in\mathbf{K}}\overleftarrow{\Pi}_k^{(\times,\times+1)} + \frac{1}{|\mathbf{K}|}\sum_{k\in\mathbf{K}}\overleftarrow{X}_k^{(\times)} , \tag{33b}$$

where $\Pi_{\mathbf{1}}^{\mathbf{K}} = \frac{1}{|\mathbf{K}|}\sum_{k\in\mathbf{K}}\mathcal{Z}_\lambda(k)$ (67), and $|\mathbf{K}|$ is the number of elements in $\mathbf{K}$.

Apart from the $\overrightarrow{X}_g^{(\text{global})}$-symmetry inherited from the general flux ladder (27), these particular cases are also symmetric under $\overleftarrow{X}_b^{(\text{global})}$ for any $b \in \mathbf{N}(\mathbf{K})$ — the *normalizer* of $\mathbf{K}$ in $\mathbf{G}$. The normalizer consists of any elements that permute elements of $\mathbf{K}$ under conjugation, and can include elements outside of $\mathbf{K}$. Using the definition of $\overleftarrow{\Pi}$ (28) and changing the summation index,

$$\overleftarrow{X}_b^{(\text{global})}\left(\sum_{k\in\mathbf{K}}\overleftarrow{\Pi}_k^{(\times,\times+1)}\right)\overleftarrow{X}_b^{\dagger(\text{global})} = \sum_{k\in\mathbf{K}}\overleftarrow{\Pi}_{bkb^{-1}}^{(\times,\times+1)} = \sum_{k\in\mathbf{K}}\overleftarrow{\Pi}_k^{(\times,\times+1)} . \tag{34}$$

Similar logic implies that the one-body sum is likewise invariant, and that the two-body sum commutes with each one-body term. In fact, this Hamiltonian can be made into a commuting projector model.

When treated as a standalone spin chain, the above Hamiltonian is a paradigmatic example of symmetry breaking of $\mathbf{G}$ into $\mathbf{K}$ — a generalization of similar models for abelian cases [110–112]. Two notable extreme cases are

$$\widehat{H}_{\langle 1\rangle\subseteq\mathbf{G}}^{\text{edge}} = -\sum_{\times} \sum_{\lambda\in\text{irr}(\mathbf{G})} \frac{d_\lambda}{|\mathbf{G}|} \text{TR}\big[\widehat{\mathcal{Z}}_\lambda^{\dagger(\times)}.\widehat{\mathcal{Z}}_\lambda^{(\times+1)}\big] = -\sum_{\times} \frac{1}{|\mathbf{G}|}\overleftarrow{\Pi}_1^{(\times,\times+1)} \tag{35a}$$

$$\widehat{H}_{\mathbf{G}\subseteq\mathbf{G}}^{\text{edge}} = -\sum_{\times} \frac{1}{|\mathbf{G}|}\sum_{g\in\mathbf{G}}\overleftarrow{X}_g^{(\times)} , \tag{35b}$$

up to identity terms. The case $\mathbf{K} = \langle 1\rangle$ yields a $\mathbf{G}$-ordered phase: the one-body terms become trivial, and the $|\mathbf{G}|$ ground states are tensor products $\cdots|g\rangle|g\rangle|g\rangle\cdots$ of the same group element $g$. Order parameters distinguishing the $|\mathbf{G}|$ symmetry-broken states are functions of $\widehat{\mathcal{Z}}$ (e.g., $\text{TR}[\widehat{\mathcal{Z}}_\lambda]$), generalizing the famous $\widehat{\sigma}_z$ order parameter for $\mathbf{G} = \mathbf{Z}_2$. The case $\mathbf{K} = \mathbf{G}$ yields a disordered phase: the two-body $ZZ$-type term becomes trivial, and there is a unique product ground state with $|\mathbf{G}\rangle \equiv \frac{1}{|\mathbf{G}|}\sum_{g\in\mathbf{G}}|g\rangle$ at each site. In the inbetween cases, the two-body term is minimized at those states $|g,h\rangle$ that satisfy $g^{-1}h \in \mathbf{K}$. The one-body term [see Eq. (89)] projects onto $\mathbf{G}/\mathbf{K}$ — the space of coset or *quotient states*

$$|a\mathbf{K}\rangle = \frac{1}{|\mathbf{K}|}\sum_{k\in\mathbf{K}}|ak\rangle . \tag{36}$$

Both terms are jointly minimized by product states $\cdots|a\mathbf{K}\rangle|a\mathbf{K}\rangle|a\mathbf{K}\rangle\cdots$, indexed by coset representatives $a \in \mathbf{G}/\mathbf{K}$. There are $|\mathbf{G}/\mathbf{K}| \equiv |\mathbf{G}|/|\mathbf{K}|$ ground states in total.

When treated as an edge of a quantum double model with no bulk excitations, the physical eigenstates of the edge theory are those for which $\overrightarrow{X}_h^{(\text{global})} = 1$ for all $h \in \mathbf{G}$ (31). The unique

ground state in this sector is the equal superposition of all ground states,

$$|\text{gnd}_{\mathbf{K}}\rangle = \frac{1}{\sqrt{|\mathbf{G}/\mathbf{K}|}} \sum_{a\in\mathbf{G}/\mathbf{K}} \cdots|a\mathbf{K}\rangle|a\mathbf{K}\rangle|a\mathbf{K}\rangle\cdots \propto \sum_{\{g_{\mathsf{x}},\, g_{\mathsf{x}}^{-1}g_{\mathsf{x}+1}\in\mathbf{K}\}} \cdots|g_{\mathsf{x}-1}\rangle|g_{\mathsf{x}}\rangle|g_{\mathsf{x}+1}\rangle\cdots. \tag{37}$$

The presence of nontrivial bulk excitations can modify this ground state, with an anyonic excitation either mapping the state (37) to a ground state in this or another symmetry sector (for which it is said that the anyon *condenses* at the edge), or mapping the state to an excited state (the anyon becomes *confined*) [8]. How the anyons condense depends on the anyon type as well as type of gapped edge [10, 62–64]. While the general case is quite involved, the extreme cases $\mathbf{K} \in \{\langle 1\rangle, \mathbf{G}\}$ serve as illuminating examples. The relevant excitation operators are

$$\overrightarrow{\Gamma}_{1,\lambda}^{(2\mathsf{x})} = \widehat{Z}_{\lambda}^{(\mathsf{x})} \qquad \text{and} \qquad \overrightarrow{\Gamma}_{g,\mathbf{1}}^{(2\mathsf{x})} = \cdots\overrightarrow{X}_{g}^{(\mathsf{x}-2)}\overrightarrow{X}_{g}^{(\mathsf{x}-1)}\overrightarrow{X}_{g}^{(\mathsf{x})}, \tag{38}$$

defined here using a convention we use later on. The first houses a charge excitation, while the second excites a flux at the edge. In the $\mathbf{K} = \langle 1\rangle$ case, all charges condense on the edge: $\text{Tr}\{\overrightarrow{\Gamma}_{1,\lambda}^{(\mathsf{x})}\}|\text{gnd}_{\langle 1\rangle}\rangle$ remains in the ground-state subspace, while $\overrightarrow{\Gamma}_{g,\mathbf{1}}^{(\mathsf{x})}|\text{gnd}_{\langle 1\rangle}\rangle$ is a linear superposition of domain-wall excitations. In the $\mathbf{K} = \mathbf{G}$ case, all fluxes condense: the separable state $|\text{gnd}_{\mathbf{G}}\rangle$ is invariant under $\overrightarrow{\Gamma}_{g,\mathbf{1}}$, while $\overrightarrow{\Gamma}_{1,\lambda}$ creates local excitations. In general, $\overrightarrow{\Gamma}_{1,\lambda}$ act as "order" operators creating local excitations, while $\overrightarrow{\Gamma}_{g,\mathbf{1}}$ act as "disorder" operators creating domain walls [113–115].

Since the excitation operators (38) do not commute with the global symmetry $\overrightarrow{X}^{(\text{global})}$, they can be interpreted as anyonic strings that straddle the bulk and edge. We will see them once more as special cases of our 1D flattened ribbon operators in the next section. They also modify the boundary conditions and symmetry sector of the model, which are explained in Appx. A.

## 3.6   Self-dual edge

Another notable instance of the flux ladder is the linear combination of $\mathbf{K} = \langle 1\rangle$ and $\mathbf{K} = \mathbf{G}$ gapped-edge Hamiltonians (35),

$$\widehat{H}_{\mathbf{G}}^{\text{sd}} = \widehat{H}_{\langle 1\rangle\subseteq\mathbf{G}}^{\text{edge}} + \widehat{H}_{\mathbf{G}\subseteq\mathbf{G}}^{\text{edge}} = -\frac{1}{|\mathbf{G}|}\sum_{\mathsf{x}} \overleftarrow{\Pi}_{1}^{(\mathsf{x},\mathsf{x}+1)} + |\mathbf{G}\rangle\langle\mathbf{G}|^{(\mathsf{x})}. \tag{39}$$

The two-body term becomes a projection onto the symmetric subspace, comprised of the states $|g, g\rangle$. At the same time, the competing one-body term, with coefficients $f_g = \frac{1}{|\mathbf{G}|}$, becomes a projection onto the state $|\mathbf{G}\rangle$ (36) — an equal superposition of all $|g\rangle$ at each site. The global symmetry of this case is the entire permutation group of $|\mathbf{G}|$ objects, represented by tensor products of permutation matrices.

The Hamiltonian (39) represents a "self-dual" point in the flux ladder's parameter space, as the two types of gapped edges that comprise it are related via Morita equivalence [10, 116]. The Hamiltonian also corresponds to the self-dual point of the $\mathbf{Z}_{N=|\mathbf{G}|}$ clock model [43, 91, 114, 115, 117–119] because it can equivalently be expressed in terms of Pauli $\widehat{X}, \widehat{Z}$ matrices associated with $|\mathbf{G}|$-dimensional qunits. We relate this point to a fermionizable point in the parameter space of the flux ladder's continuum version in Sec. 6.

## 4   Ribbon and Jordan-Wigner operators

A key feature of an anyonic string in the $\mathbf{Z}_N$ surface code is that it violates plaquette and/or star constraints only at its ends, while commuting with all Hamiltonian terms supported on

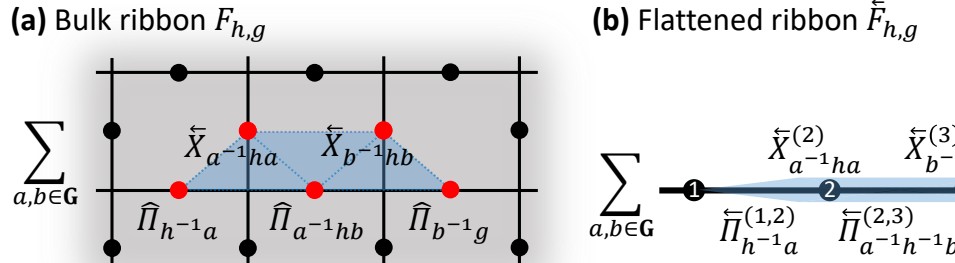

Figure 5: (a) Sketch of a ribbon operator in the bulk of a quantum double model [1], which acts on sites (red) of a ladder-like region of the lattice outlined by linked triangles (blue). Here, $\widehat{\Pi}_g = |g\rangle\langle g|$. (b) A flattened ribbon (43), acting on a 1D chain and obtained by taking an ordinary ribbon and applying the bulk ground-state constraints from Sec. 3.4.

its body. In this way, the body can be moved around (as long as the ends are fixed) without changing the system's energy. Their 1D analogues on the surface-code edge are tensor-product "strings" of $\widehat{X}$ operators acting on the clock model (9), similarly carrying energy only at their ends. The symmetry $\widehat{X}^{(\mathrm{global})}$ (12) is the special case of a string occupying the entire chain.

Quantum-double analogues of anyonic strings — ribbon operators [1] — maintain the key property of carrying energy only at their ends, but can no longer be expressed as a tensor product of single-site terms. Projecting them onto the effective bulk ground-state space yields *flattened ribbons*, one of the main results of this work that we present in Sec. 4.1. In Sec. 4.2, we superimpose flattened ribbons to obtain Jordan-Wigner (JW) operators $\overleftarrow{\Lambda}$ associated with defects of quantum-double models. We simplify the form of the JW operators by introducing their "decoupled" versions $\overrightarrow{\Gamma}$ in Sec. 4.3, and use them to render the flux ladder (27).

## 4.1 Flattened ribbons

Flattened ribbons are obtained by projecting ordinary ribbons onto the space of the effective flux-ladder edge theory. We omit such a derivation here for simplicity and instead elaborate on the flattened ribbons' properties. We will need the formulas

$$\overleftarrow{X}^{(\mathsf{x})}_k \overleftarrow{\Pi}^{(\mathsf{x},\mathsf{y})}_h \overleftarrow{X}^{\dagger(\mathsf{x})}_k = \overleftarrow{\Pi}^{(\mathsf{x},\mathsf{y})}_{kh} \qquad \text{and} \qquad \overleftarrow{X}^{(\mathsf{y})}_k \overleftarrow{\Pi}^{(\mathsf{x},\mathsf{y})}_h \overleftarrow{X}^{\dagger(\mathsf{y})}_k = \overleftarrow{\Pi}^{(\mathsf{x},\mathsf{y})}_{hk^{-1}}, \qquad (40)$$

as well as the "multiplication" and "addition" rules for $\overleftarrow{\Pi}$,

$$\overleftarrow{\Pi}^{(\mathsf{x},\mathsf{y})}_a \overleftarrow{\Pi}^{(\mathsf{x},\mathsf{y})}_b = \delta^{\mathbf{G}}_{a,b} \overleftarrow{\Pi}^{(\mathsf{x},\mathsf{y})}_a \qquad \text{and} \qquad \sum_{b \in \mathbf{G}} \overleftarrow{\Pi}^{(\mathsf{x},\mathsf{z})}_{ab} \overleftarrow{\Pi}^{(\mathsf{z},\mathsf{y})}_{b^{-1}c} = \overleftarrow{\Pi}^{(\mathsf{x},\mathsf{y})}_{ac}. \qquad (41)$$

A flattened ribbon has support on a chain of sites from $\mathsf{x}$ to $\mathsf{x}+\mathsf{y}$, with its two *ends* being $\mathsf{x}$ and $\mathsf{x}+\mathsf{y}$ and the *body* consisting of all sites inbetween. A ribbon $\overleftarrow{F}$ with the smallest nontrivial body lies on three sites,

$$\overleftarrow{F}^{[1,3]}_{h,g} = \sum_{a \in \mathbf{G}} \overleftarrow{\Pi}^{(1,2)}_{h^{-1}a} \overleftarrow{X}^{(2)}_{a^{-1}ha} \overleftarrow{\Pi}^{(2,3)}_{a^{-1}g}, \qquad (42)$$

where $h, g \in \mathbf{G}$. A nontrivial property of a ribbon is that it commutes with any $\overleftarrow{X}$ and $\overleftarrow{\Pi}$ supported on sites within its body. The body of the above ribbon lies on site 2, large enough to support an $\overleftarrow{X}$. Using Eqs. (40) and $\overleftarrow{X}_k \overleftarrow{X}_g = \overleftarrow{X}_{kgk^{-1}} \overleftarrow{X}_k$, conjugation of $\overleftarrow{F}$ by $\overleftarrow{X}^{(2)}_k$ is equivalent to letting $a \to ak$ in the sum, which merely permutes its terms.

Longer ribbons follow a similar pattern of $\overleftarrow{X}$'s intertwined by $\overleftarrow{\Pi}$'s,

$$\overleftarrow{F}^{[\mathsf{x},\mathsf{x}+\mathsf{y}]}_{h,g} = \sum_{\{a_\iota\} \in \mathbf{G}^{\times(\mathsf{y}-1)}} \overleftarrow{\Pi}^{(\mathsf{x},\mathsf{x}+1)}_{h^{-1}a_1} \overleftarrow{X}^{(\mathsf{x}+1)}_{a_1^{-1}ha_1} \overleftarrow{\Pi}^{(\mathsf{x}+1,\mathsf{x}+2)}_{a_1^{-1}h^{-1}a_2} \overleftarrow{X}^{(\mathsf{x}+2)}_{a_2^{-1}ha_2} \cdots \overleftarrow{X}^{(\mathsf{x}+\mathsf{y}-1)}_{a_{\mathsf{y}-1}^{-1}ha_{\mathsf{y}-1}} \overleftarrow{\Pi}^{(\mathsf{x}+\mathsf{y}-1,\mathsf{x}+\mathsf{y})}_{a_{\mathsf{y}-1}^{-1}g}. \qquad (43)$$

Ribbons commute with $\overrightarrow{X}^{(\text{global})}$ (31) since both $\overleftarrow{X}$ and $\overleftarrow{\Pi}$ do so. For the $\mathbf{G} = \mathbf{Z}_N$ case, sequential use of the addition formula (41) collapses the intermediate $\widehat{\Pi}$'s and yields the string $\widehat{X}_h^{(\text{x}+1)}\widehat{X}_h^{(\text{x}+2)}\cdots\widehat{X}_h^{(\text{x}+y-1)}\widehat{\Pi}_g^{(\text{x},\text{x}+y)}$.

Now we show that flattened ribbons commute with any $\overleftarrow{\Pi}$'s within their body. Letting x = 0, consider permuting $\overleftarrow{\Pi}_k$ on sites 1 and 2 through the ribbon. This projection interacts only with three operators ($\overleftarrow{X}^{(1)}$, $\overleftarrow{\Pi}^{(1,2)}$, and $\overleftarrow{X}^{(2)}$) in each term in the ribbon's sum. Permuting those operators through with the help of Eqs. (40) yields

$$\left(\overleftarrow{X}^{(1)}_{a_1^{-1}ha_1}\overleftarrow{\Pi}^{(1,2)}_{a_1^{-1}h^{-1}a_2}\overleftarrow{X}^{(2)}_{a_2^{-1}ha_2}\right)\overleftarrow{\Pi}^{(1,2)}_k = \overleftarrow{\Pi}^{(1,2)}_{a_1^{-1}ha_1ka_2^{-1}h^{-1}a_2}\left(\overleftarrow{\Pi}^{(1,2)}_{a_1^{-1}a_2}\overleftarrow{X}^{(1)}_{a_1^{-1}ha_1}\overleftarrow{X}^{(2)}_{a_2^{-1}ha_2}\right). \quad (44)$$

Using the multiplication rule (41) for the two $\overleftarrow{\Pi}$'s yields zero unless $a_1^{-1}ha_1ka_2^{-1}h^{-1}a_2 = a_1^{-1}a_2$, which simplifies to $k = a_1^{-1}a_2$. Thus, we can substitute $a_1^{-1}ha_1ka_2^{-1}h^{-1}a_2 \to k$, completing the proof.

The above ribbons have a $\overleftarrow{\Pi}$ on each end, and ribbons ending with $\overleftarrow{X}$ are defined as

$$\overleftarrow{F}^{[\![\text{x},\text{x}+y]\!]}_{h,g} = \overleftarrow{X}^{(\text{x})}_h\overleftarrow{F}^{[\text{x},\text{x}+y]}_{h,g} \qquad \text{and} \qquad \overleftarrow{F}^{[\text{x},\text{x}+y]\!]}_{h,g} = \overleftarrow{F}^{[\text{x},\text{x}+y]}_{h,g}\overleftarrow{X}^{(\text{x}+y)}_{g^{-1}hg}. \quad (45)$$

In general, flattened ribbons can be recursively constructed from the minimal ribbons

$$\overleftarrow{F}^{[\text{x},\text{x}+1]}_{h,g} = \overleftarrow{\Pi}^{(\text{x},\text{x}+1)}_g \qquad \text{and} \qquad \overleftarrow{F}^{[\![\text{x},\text{x}]\!]}_{h,g} = \overleftarrow{X}_h\delta_{hg,1}, \quad (46)$$

along with the "gluing" relation, for x $\leq$ z $\leq$ x + y,

$$\overleftarrow{F}^{[\text{x},\text{x}+y]}_{h,g} = \sum_{a\in\mathbf{G}}\overleftarrow{F}^{[\text{x},\text{z}]}_{h,h^{-1}a}\overleftarrow{F}^{[\![\text{z},\text{x}+y]}_{a^{-1}ha,a^{-1}g} = \sum_{a\in\mathbf{G}}\overleftarrow{F}^{[\text{x},\text{z}]}_{h,h^{-1}a}\overleftarrow{F}^{[\text{z},\text{x}+y]}_{a^{-1}ha,a^{-1}g}. \quad (47)$$

Ribbons with nontrivial bodies (y $\geq$ 2) satisfy

$$\overleftarrow{F}^{[\text{x},\text{x}+y]}_{h_1,g_1}\overleftarrow{F}^{[\text{x},\text{x}+y]}_{h_2,g_2} = \delta^{\mathbf{G}}_{g_2,g_1}\overleftarrow{F}^{[\text{x},\text{x}+y]}_{h_2h_1,g_1} \qquad \text{and} \qquad \overleftarrow{F}^{\dagger[\text{x},\text{x}+y]}_{h,g} = \overleftarrow{F}^{[\text{x},\text{x}+y]}_{h^{-1},g}. \quad (48)$$

They form a basis for all $\overrightarrow{X}^{(\text{global})}$-symmetric operators, and the flux ladder (27) can be expressed using the minimal ribbons (46).

Flattened ribbons $\overleftarrow{F}^{[\![\text{x},\text{y}]\!]}$ satisfy the same commutation relations with Hamiltonian operators at the ribbons' ends as ordinary ribbons do with plaquette and star operators of the bulk quantum double { [62], Eqs. (9-12)}, namely,

$$\begin{aligned}\overleftarrow{X}^{(\text{x})}_k\overleftarrow{F}^{[\![\text{x},\text{y}]\!]}_{h,g} &= \overleftarrow{F}^{[\![\text{x},\text{y}]\!]}_{khk^{-1},kg}\overleftarrow{X}^{(\text{x})}_k & \overleftarrow{X}^{(\text{y})}_k\overleftarrow{F}^{[\![\text{x},\text{y}]\!]}_{h,g} &= \overleftarrow{F}^{[\![\text{x},\text{y}]\!]}_{h,gk^{-1}}\overleftarrow{X}^{(\text{y})}_k \\ \overleftarrow{\Pi}^{(\text{x}-1,\text{x})}_k\overleftarrow{F}^{[\![\text{x},\text{y}]\!]}_{h,g} &= \overleftarrow{F}^{[\![\text{x},\text{y}]\!]}_{h,g}\overleftarrow{\Pi}^{(\text{x}-1,\text{x})}_{kh} & \overleftarrow{\Pi}^{(\text{y},\text{y}+1)}_k\overleftarrow{F}^{[\![\text{x},\text{y}]\!]}_{h,g} &= \overleftarrow{F}^{[\![\text{x},\text{y}]\!]}_{h,g}\overleftarrow{\Pi}^{(\text{y},\text{y}+1)}_{g^{-1}h^{-1}gk}\end{aligned}. \quad (49)$$

This means that flattened ribbons carry pairs of anyonic excitations at their ends, forming a basis for the algebra of anyonic excitations that can be used to determine which bulk anyons condense on particular gapped edges. A given ribbon can carry multiple anyons, but particular superpositions of ribbons carrying a desired anyon can be constructed via a basis change [ [61], Eq. (B66); see also [64], Thm. 2.6]. Examples of ribbons carrying charge and flux anyons are shown in Table 2. An alternative basis expressing explicit edge excitations can also be constructed { [63], Sec. 2.3.3}. Flattened ribbons cannot change the bulk anyonic data, however, since they commute with the global symmetry $\overrightarrow{X}^{(\text{global})}$ (31). They thus represent strictly edge excitations that do no change the symmetry sector of the model.

| Excitations on edge | Expression | Ribbon |
|---|---|---|
| charges $\lambda$ | $\mathrm{Tr}\big[\widehat{\mathcal{Z}}_\lambda^{\dagger(\mathsf{x})}.\widehat{\mathcal{Z}}_\lambda^{(\mathsf{y})}\big]$ | $\sum\limits_{g\in\mathbf{G}}\mathrm{Tr}\big[\mathcal{Z}_\lambda\left(g\right)\big]\overset{\leftarrow}{F}_{1,g}^{[\mathsf{x},\mathsf{y}]}$ |
| fluxes $\mathrm{cls}(g)$ | $\sum\limits_{\{a_\iota\}\in\mathbf{G}}\overset{\leftarrow}{X}_g^{(\mathsf{x})}\overset{\leftarrow}{\Pi}_{g^{-1}a_1}^{(\mathsf{x},\mathsf{x}+1)}\overset{\leftarrow}{X}_{a_1^{-1}ga_2}^{(\mathsf{x}+1)}\cdots\overset{\leftarrow}{X}_g^{(\mathsf{y})}$ | $\overset{\leftarrow}{F}_{g,1}^{[\![\mathsf{x},\mathsf{y}]\!]}$ |

| Excitations from bulk | Expression | JW operator |
|---|---|---|
| charge $\lambda$ | $\mathrm{Tr}\big[\widehat{\mathcal{Z}}_\lambda^{(\mathsf{x})}\big]$ | $\mathrm{Tr}\big[\overset{\rightarrow}{\varGamma}_{1,\lambda}^{(2\mathsf{x})}\big]$ |
| flux $\mathrm{cls}(g)$ | $\cdots\overset{\rightarrow}{X}_g^{(\mathsf{x}-2)}\overset{\rightarrow}{X}_g^{(\mathsf{x}-1)}\overset{\rightarrow}{X}_g^{(\mathsf{x})}$ | $\overset{\rightarrow}{\varGamma}_{g,\mathbf{1}}^{(2\mathsf{x})}$ |

Table 2: Anyonic excitations on the edge of a quantum double can come from ribbon operators that are confined to the edge ("Excitations on edge" in the above table), or those that come from the bulk and touch the edge with only one end ("Excitations from bulk"). The former corresponds to flattened ribbons $\overset{\leftarrow}{F}$ (43), which cause anyonic excitations at each of their two ends. Rows two and three of the above table contains examples of such ribbons. Examples of the latter, shown in the last two rows, can be represented by the decoupled Jordan-Wigner operators $\overset{\rightarrow}{\varGamma}$ (56).

## 4.2 Jordan-Wigner operators

Having covered how to construct anyonic excitations out of flattened ribbons, here we convert our ribbons into those that touch the edge at only one end. This generalizes the Jordan-Wigner mapping to general groups, yielding a way to construct point defects between flux ladders in different phases.

Recall that $\mathbf{Z}_N$ Jordan-Wigner (JW) operators [43,44,47] are of the form $\widehat{X}\widehat{X}\cdots\widehat{Z}$, consisting of a *tail* of $\widehat{X}$ operators labeled by a group element $h$, and a $\widehat{Z}$-operator *head* labeled by an irrep $\lambda$. "Two-headed" versions of such operators (beginning and ending with a $\widehat{Z}$ operator) can be obtained by properly superimposing $\mathbf{Z}_N$ ribbons,

$$\sum_{b\in\mathbf{Z}_N}e^{i\frac{2\pi}{N}\lambda b}\overset{\leftarrow}{F}_{h,b}^{[\mathsf{x},\mathsf{x}+\mathsf{y}]}=\widehat{Z}_\lambda^{\dagger(\mathsf{x})}\widehat{X}_h^{(\mathsf{x}+1)}\widehat{X}_h^{(\mathsf{x}+2)}\cdots\widehat{X}_h^{(\mathsf{x}+\mathsf{y}-1)}\widehat{Z}_\lambda^{(\mathsf{x}+\mathsf{y})}. \qquad (50)$$

These create an anyonic string carrying one charge and one flux excitation at each end. In order to obtain a bona-fide JW string, we remove the effect of the leftmost $\widehat{Z}_\lambda^\dagger$ by placing it on a site far away from the chain. For a chain with sites $\mathsf{x}=1$ through $\mathsf{L}$ and open boundary conditions, the $\widehat{Z}_\lambda^\dagger$ can be placed at a site symbolically denoted as $\mathsf{x}=0$.

Extending to the non-Abelian case via Eq. (29b), we introduce JW-operators

$$\overset{\leftarrow}{\varLambda}_{g,\lambda}^{(2\mathsf{x}-1)}=\sum_{b\in\mathbf{G}}\mathcal{Z}_\lambda\left(b\right)\overset{\leftarrow}{F}_{g,b}^{[0,\mathsf{x}]} \qquad\text{and}\qquad \overset{\leftarrow}{\varLambda}_{g,\lambda}^{(2\mathsf{x})}=\sum_{b\in\mathbf{G}}\mathcal{Z}_\lambda\left(b\right)\overset{\leftarrow}{F}_{g,b}^{[0,\mathsf{x}]}, \qquad (51)$$

where the two types correspond to terminations $\cdots\widehat{X}_g^{(\mathsf{x}-1)}\widehat{Z}_\lambda^{(\mathsf{x})}$ and $\cdots\widehat{X}_g^{(\mathsf{x})}\widehat{Z}_\lambda^{(\mathsf{x})}$ for the $\mathbf{Z}_N$ case, respectively. Acting on both the group and internal spaces, they satisfy

$$\overset{\leftarrow}{\varLambda}_{g,\tau}^{(m)}\overset{\leftarrow}{\varLambda}_{h,\lambda}^{(n)}=\mathcal{Z}_\lambda\left(g\right).\overset{\leftarrow}{\varLambda}_{g^{-1}hg,\lambda}^{(n)}\overset{\leftarrow}{\varLambda}_{g,\tau}^{(m)} \qquad\text{and}\qquad \overset{\leftarrow}{\varLambda}_{g,\lambda}^{\dagger(n)}.\overset{\leftarrow}{\varLambda}_{g,\lambda}^{(n)}=1_\lambda\otimes 1, \qquad (52)$$

where $m>n$, $1_\lambda$ is the identity on the internal irrep space $\lambda$ [see Eq. (14)], and $1$ the identity on the group space. The $\mathcal{Z}_\lambda(k)$ matrix generalizes the root-of-unity phase present in the $\mathbf{Z}_N$ (parafermionic) case, while the conjugation $g^{-1}hg$ is a side-effect of the non-Abelian case. Just like ribbons, single JW operators generally carry multiple anyonic excitations at their ends. A notable exception is $\overset{\leftarrow}{\varLambda}_{1,\lambda}$, which carries pure charge excitations $\lambda$.

JW operators can also be *fused* via tensor products [109] in the irrep space, which reduce to ordinary multiplication in the abelian case. The tensor product of two irreps $\tau, \lambda$ can be decomposed into a direct sum over some irreps, $\mathcal{Z}_\tau \otimes \mathcal{Z}_\lambda = \bigoplus_\mu \mathcal{Z}_\mu$. A given irrep can be present more than once in the sum. This yields the fusion rules

$$\overleftarrow{\Lambda}^{(2x-1)}_{h,\tau} \otimes \overleftarrow{\Lambda}^{(2x-1)}_{g,\lambda} = \bigoplus_\mu \overleftarrow{\Lambda}^{(2x-1)}_{gh,\mu} \ . \tag{53}$$

JW operators do not commute with the global symmetry $\overrightarrow{X}^{(\text{global})}$, which means they violate the bulk ground-state constraint (31) and can be used to change the bulk anyonic data. These operators thus correspond to flattened ribbons whose end at $x = 0$ dips into and causes an excitation in the bulk. Treating the bulk as very far away, we define *local* operators to be symmetric products of a few nearby operators. For example, a single $\overleftarrow{\Lambda}^{(n)}$ is not local since it is coupled to the bulk site $x = 0$, while the product $\overleftarrow{\Lambda}^{\dagger(2x)}_{g,\lambda} . \overleftarrow{\Lambda}^{(2x+1)}_{g,\lambda} = \widehat{\mathcal{Z}}^{\dagger(x)}_\lambda . \widehat{\mathcal{Z}}^{(x+1)}_\lambda$ is local for any $g, \lambda$. This product can be used to express the most general two-body term of the flux ladder (27). Surprisingly, the other product of JW operators is *not* local. After using Eqs. (45) and (48), the resulting combination remains supported on the bulk site,

$$\text{TR}\big[\overleftarrow{\Lambda}^{\dagger(2x-1)}_{g,\lambda} . \overleftarrow{\Lambda}^{(2x)}_{g,\lambda}\big] = \sum_{b,c \in \mathbf{G}} \text{TR}\big[\mathcal{Z}_\lambda(b^{-1}c)\big] \overleftarrow{F}^{[0,x]}_{g^{-1},b} \overleftarrow{F}^{[0,x]}_{g,c} \overleftarrow{X}^{(x)}_{c^{-1}gc} = d_\lambda \sum_{b \in \mathbf{G}} \overleftarrow{\Pi}^{(0,x)}_b \overleftarrow{X}^{(x)}_{b^{-1}gb}. \tag{54}$$

Since the above is not equal to a $\overleftarrow{X}^{(x)}_g$, a workaround is required in order to render the single-body term in the flux ladder with products of two JW operators.

## 4.3 Decoupled Jordan-Wigner operators

One way to circumvent the above obstruction and construct JW operators that can be used to express the flux ladder is to fix the outside $x = 0$ site to be in the identity state $|g = 1\rangle$. This collapses the flattened ribbons' alternating products of $\overleftarrow{\Pi}$ and $\overleftarrow{X}$ into a tensor product of $\overrightarrow{X}$ using

$$\sum_{a \in \mathbf{G}} |a\rangle\langle a| \overleftarrow{X}_{a^{-1}g^{-1}a} = \overrightarrow{X}_g \ . \tag{55}$$

We define "decoupled" versions of the JW operators, which (analogous to the abelian case) consist of a product of $\overrightarrow{X}$'s for a tail and a $\widehat{\mathcal{Z}}$ head,

$$\overrightarrow{\Gamma}^{(2x-1)}_{g,\lambda} \equiv \langle 1|^{(0)} \overleftarrow{\Lambda}^{(2x-1)}_{g^{-1},\lambda} |1\rangle^{(0)} = \overrightarrow{X}^{(1)}_g \overrightarrow{X}^{(2)}_g \cdots \overrightarrow{X}^{(x-1)}_g \widehat{\mathcal{Z}}^{(x)}_\lambda \ , \tag{56}$$

and similarly, $\overrightarrow{\Gamma}^{(2x)}_{g,\lambda} \equiv \overrightarrow{X}^{(1)}_g \overrightarrow{X}^{(2)}_g \cdots \overrightarrow{X}^{(x)}_g \widehat{\mathcal{Z}}^{(x)}_\lambda$. These are precisely combinations of the "order-disorder" bulk excitation operators encountered in our analysis of gapped edges [see Eq. (38) and Table 2]. Equation (56) shows how such simplified bulk excitations can be obtained from superpositions of flattened ribbons that straddle the bulk and edge of the system.

Simple nearest-neighbor combinations of decoupled JW operators readily express a subset of all possible flux ladder Hamiltonians. One combination renders a single $\overrightarrow{X}$, while the other product renders the two-body $\widehat{\mathcal{Z}}^\dagger . \widehat{\mathcal{Z}}$ term. Combining these yields

$$\widehat{H}^{\text{JW}}_{\mathbf{G}} = -\sum_x \sum_{\lambda \in \text{irr}(\mathbf{G})} \text{TR}\big[C_\lambda . \widehat{\mathcal{Z}}^{\dagger(x)}_\lambda . \widehat{\mathcal{Z}}^{(x+1)}_\lambda\big] + \sum_{g \in \mathbf{G}} f_g \overrightarrow{X}^{(x)}_g \tag{57a}$$

$$= -\sum_x \sum_{\lambda \in \text{irr}(\mathbf{G})} \text{TR}\big[C_\lambda . \overrightarrow{\Gamma}^{\dagger(2x)}_{a,\lambda} . \overrightarrow{\Gamma}^{(2x+1)}_{a,\lambda}\big] + \sum_{g \in \mathbf{G}} f_g \frac{1}{d_\sigma} \text{TR}\big[\overrightarrow{\Gamma}^{\dagger(2x-1)}_{g,\sigma} . \overrightarrow{\Gamma}^{(2x)}_{g,\sigma}\big] \ , \tag{57b}$$

valid for any group element $a$ and irrep $\sigma$. The difference between this Hamiltonian and the general flux ladder (27a) is the direction of the arrow above the one-body terms. In order for

the sum of such terms to commute with $\overrightarrow{X}_k^{(\text{global})}$, the coefficients have to be invariant under conjugation, $\mathsf{f}_{k^{-1}gk} = \mathsf{f}_g$. That way, the sum is invariant after a change of variables,

$$\overrightarrow{X}_k \left( \sum_{g \in \mathbf{G}} \mathsf{f}_g \overrightarrow{X}_g \right) \overrightarrow{X}_k = \sum_{g \in \mathbf{G}} \mathsf{f}_g \overrightarrow{X}_{kgk^{-1}} = \sum_{g \in \mathbf{G}} \mathsf{f}_{k^{-1}gk} \overrightarrow{X}_g = \sum_{g \in \mathbf{G}} \mathsf{f}_g \overrightarrow{X}_g . \tag{58}$$

The most general form of invariant coefficients is $\mathsf{f}_g = \sum_{\lambda \in \text{irr}(\mathbf{G})} \mathsf{f}_\lambda \text{TR}[\mathcal{Z}_\lambda(g)]$, with coefficients $\{\mathsf{f}_\lambda\}$ defined such that the flux ladder is Hermitian. This Hamiltonian is, perhaps coincidentally, identical to that considered in Ref. [71], and our decoupled JW construction offers a quantum-double inspired alternative to their dyonic-mode construction.

General $\overleftarrow{X}_g$ operators can be expressed using products of JW operators with mismatched "heads". Using a trick similar to Eq. (55) to express $\overleftarrow{X}$ in terms of $\overrightarrow{X}$ and writing $|h\rangle\langle h|$ as a sum of $Z$-type operators using orthogonality relations (21) yields

$$\overleftarrow{X}_g^{(\times)} = \sum_{h \in \mathbf{G}} \overrightarrow{X}_{h^{-1}g^{-1}h}^{(\times)} \sum_{\lambda \in \text{irr}(\mathbf{G})} \frac{d_\lambda}{|\mathbf{G}|} \text{TR}\left[ \mathcal{Z}_\lambda(h) . \widehat{\mathcal{Z}}_\lambda^{(\times)} \right] \tag{59a}$$

$$= \sum_{h \in \mathbf{G}} \sum_{\lambda \in \text{irr}(\mathbf{G})} \frac{d_\lambda}{|\mathbf{G}|} \text{TR}\left[ \mathcal{Z}_\lambda(h) . \overrightarrow{\Gamma}_{h^{-1}g^{-1}h,\mathbf{1}}^{\dagger(2\times-1)} . \overrightarrow{\Gamma}_{h^{-1}g^{-1}h,\lambda}^{(2\times)} \right] . \tag{59b}$$

While this means the general flux ladder can be expressed in terms of decoupled JW bilinears, we stick with the simpler case of Eq. (57) for the example below.

**Dihedral case**   Using decoupled JW operators (56) leaves freedom to choose the types "heads" $\sigma$ and "tails" $a$ for constructing the respective one- and two- body terms in $H_\mathbf{G}^{\text{JW}}$ (57). Similar to the $\mathbf{Z}_N$ case, we do not need all possible combinations to express the Hamiltonian for $\mathbf{G} = \mathbf{D}_N$. We can develop a "generating set" of $\{g, \lambda\}$, appealing to the correspondence between irreps and conjugacy classes [109]. As an example for odd $N$, consider, respectively, the parafermion-like and Majorana-like operators

$$\overrightarrow{\eta}_\tau \equiv \overrightarrow{\Gamma}_{g=r^\tau, \lambda=\mathbf{2}_\tau} \qquad \text{and} \qquad \overrightarrow{\gamma} \equiv \overrightarrow{\Gamma}_{g=p, \lambda=\mathbf{1}'} . \tag{60}$$

There are $(N+1)/2$ parafermion-like operators since that is the number of 2D irreps labeled by $\tau \in \{1, 2, \cdots, (N-1)/2\}$. Each irrep is paired with the $\tau^{\text{th}}$ power of the rotation $r$. The remaining one-dimensional irrep is paired with the reflection $p$ to make the Majorana operator. The even $N$ case presents two more Majorana-like pairs labeled by $\{r^{N/2}, \mathbf{1}_+\}$ and $\{pr, \mathbf{1}_-\}$. Since their irreps are one-dimensional, all Majorana-like operators act only on the group space and square to the identity. Applying Eq. (52), the two respective sets of operators mimic parafermionic and Majorana commutation relations; for $\mathsf{n} < \mathsf{m}$,

$$\overrightarrow{\eta}_\sigma^{(\mathsf{m})} \overrightarrow{\eta}_\tau^{(\mathsf{n})} = \mathcal{Z}_{\mathbf{2}_\tau}(r) . \overrightarrow{\eta}_\tau^{(\mathsf{n})} \overrightarrow{\eta}_\sigma^{(\mathsf{m})} \qquad \text{and} \qquad \overrightarrow{\gamma}^{(\mathsf{m})} \overrightarrow{\gamma}^{(\mathsf{n})} = -\overrightarrow{\gamma}^{(\mathsf{n})} \overrightarrow{\gamma}^{(\mathsf{m})} , \tag{61}$$

where $\mathcal{Z}_{\mathbf{2}_\tau}(r)$ (16) is a diagonal matrix with root-of-unity entries. However, $\overrightarrow{\eta}$ does not commute with $\overrightarrow{\gamma}$, and their various combinations render the terms in $\widehat{H}_{\mathbf{D}_N}^{\text{JW}}$ (57).

Armed with this construction, we can postulate the existence of JW operators that commute with the Hamiltonian and that are stable to small perturbations. Such operators, if they exist, would correspond to extensions of $\mathbf{Z}_2$ Majorana and $\mathbf{Z}_N$ parafermion zero modes to $\mathbf{D}_N$. Such extensions could lead to novel topologically protected quantum computation schemes based on non-Abelian groups.

Consider the limiting case $\mathsf{f} = 0$ of the flux ladder. The operators $\overrightarrow{\gamma}^{(1)}$, $\overrightarrow{\gamma}^{(2L)}$, $\overrightarrow{\eta}^{(1)}$, and $\overrightarrow{\eta}^{(2L)}$ are not present in and commute with $\widehat{H}_{\mathbf{D}_N}^{\text{JW}}$, corresponding to trivial examples of zero modes. Robustness under $\mathsf{f} \ll 1$ perturbations would make them nontrivial, but such stability is

not guaranteed already in the abelian $\mathbf{Z}_{N>2}$ case. There is numerical evidence that parafermion zero modes are stable around special points of the clock model, where the spectrum of the two-body term happens to be symmetric around zero [47, 87, 88]. In Appx. B, we identify a point in the $\mathbf{D}_3$ flux ladder's parameter space with that feature, pointing to a special point at which these modes might be stable. Stability of such modes could also be related to stability of the dyonic modes of Ref. [71], and it may be possible to construct weaker versions of zero modes that are robust and that commute with a projected version of the flux ladder.

We conclude this section with a summary. We construct strictly 1D (i.e., "flattened") versions of quantum-double ribbon operators that carry anyonic excitations on the flux ladder — an effective theory for the quantum-double edge (see Fig. 5). We then construct superpositions of flattened ribbons that extend the Jordan-Wigner mapping to non-Abelian groups. In the quantum double framework, this provides a way to construct anyonic zero mode operators carrying anyons that condense on a point defect between two different boundaries, but not the boundaries themselves. Assuming the flux ladder is treated as an edge theory of a quantum double topological phase, some JW operators may bind generalized zero modes that are robust to noise and that generalize Majorana and parafermion zero modes. We leave determination of such robustness to future work.

## 5   Continuum description of the flux ladder

The rest of the paper is devoted to realizing instances of the general flux ladder with electrons near the Fermi level. The motivation is to foster realization of quantum-double non-Abelian defects beyond Majorana and parafermion zero modes in real materials. The procedure consists of two parts, (1) developing a continuum version of the flux ladder model, and (2) recasting an instance of the resulting continuum field theory in terms of fermionic operators. We develop the part (1) here, and part (2) in the next section.

The general flux ladder acts on sites that are valued in a finite group $\mathbf{G}$, an inherently discrete object. In order to "smooth out" this group space, we *embed* the flux ladder into another one acting on sites that are valued in a connected Lie group $\mathbf{\Gamma} \supset \mathbf{G}$. Such an embedding works equally well for the group-valued "position" and irrep-valued "momentum" spaces of the sites. Group elements $g \in \mathbf{G}$, labeling $X$-type operators, naturally correspond to some chosen elements of $\mathbf{\Gamma}$. Any irrep $\lambda \in \mathrm{irr}(\mathbf{G})$, labeling $Z$-type operators, can be obtained from an irrep $\ell$ of $\mathbf{\Gamma}$ whenever the corresponding irrep matrix $U_\ell$ contains (i.e., *branches to*) a copy of $\lambda$ when evaluated at elements of $\mathbf{G}$. Operators we need for our procedure are summarized in Table 3 for the known $\mathbf{Z}_N \hookrightarrow \mathbf{U}(1)$ embedding and a dihedral version introduced here.

This $\mathbf{G} \hookrightarrow \mathbf{\Gamma}$ embedding yields expressions of all terms of the original flux ladder in terms of exponentials of the Lie algebra of $\mathbf{\Gamma}$, allowing us to perform smooth manipulations that were not possible with the original discrete model. The field theory — a generalized $\mathbf{\Gamma}$ principal chiral model — is then obtained by *linearization* [19], i.e., expansion of all exponentials in the continuum limit.

### 5.1   Clock model to free boson

A conventional heuristic to construct a field theory from a lattice model is to identify the relevant fields and write down the simplest terms that respect the lattice model's symmetry. For the clock model (9), the relevant field is a $\mathbf{Z}_N$-valued order parameter distinguishing between these the $\mathbf{Z}_N$ symmetry-breaking phase in the f $\to 0$ limit and the disordered phase in the C $\to 0$ limit. A minimal field theory describing said order parameter is that of a free boson, supplemented by the leading-order $\mathbf{Z}_N$-symmetric cosine potential ( [89], Sec. 5.5; [90], pg. 186; [91]).

| $\mathbf{G} \hookrightarrow \mathbf{\Gamma}$ | $\mathbf{Z}_N \hookrightarrow \mathbf{U}(1)$ | $\mathbf{D}_N \hookrightarrow \mathbf{SO}(3)$ |
|:---:|:---:|:---:|
| $Z$-type operators | $\widehat{Z}_\lambda \hookrightarrow e^{i\lambda\widehat{\phi}}$ | $\widehat{Z}_\lambda \hookrightarrow \widehat{U}_{\ell_\lambda} = e^{-i\widehat{\phi}^{\mathrm{a}} L^{\mathrm{a}}_{\ell_\lambda}}$ |
| $X$-type operators | $\widehat{X}_h \hookrightarrow e^{-i\frac{2\pi}{N}h\widehat{L}}$ | $\overrightarrow{X}_g \hookrightarrow \overrightarrow{X}_{R=g} = e^{-i\phi^{\mathrm{a}}_R \overrightarrow{L}^{\mathrm{a}}}$ $\overleftarrow{X}_g \hookrightarrow \overleftarrow{X}_{R=g} = e^{-i\phi^{\mathrm{a}}_R \overleftarrow{L}^{\mathrm{a}}}$ |
| $\widehat{V}^{\mathrm{smooth}}_{\mathbf{G}}$ potential | $\cos(N\widehat{\phi})$ | $\mathrm{TR}\big[\Pi^{\mathbf{D}_N}_{\mathbf{1}}.\widehat{U}_{\ell=N}\big]$ |
| $\widehat{V}^{\mathrm{smooth}}_{\mathbf{\Gamma/G}}$ potential | $\cos\big(\frac{2\pi}{N}\widehat{L}\big)$ | $\frac{1}{2}\cos\big(\frac{2\pi}{N}\overleftarrow{L}^{\mathrm{z}}\big) + \cos\big(\pi\overleftarrow{L}^{\mathrm{x}}\big)$ |

Table 3: A continuum description of the flux ladder model for a discrete group $\mathbf{G}$ can be obtained by embedding each $\mathbf{G}$-valued site into a site valued in a connected Lie group $\mathbf{\Gamma} \supset \mathbf{G}$. An often-used version of this procedure embeds a $\mathbf{Z}_N$-valued qunit into a $\mathbf{U}(1)$ rotor, and we list the relevant operators of that construction in the second column. The $\mathbf{Z}_N$-potential is minimized at the rotor's $N$ equidistant angular positions forming the $\mathbf{Z}_N$ qunit, while the $\mathbf{U}(1)/\mathbf{Z}_N$-potential is minimized at angular momenta that are multiples of $N$. In Secs. 5 and 6, we develop a non-Abelian generalization of this procedure. The corresponding operators for $\mathbf{G}$ being the dihedral group and $\mathbf{\Gamma}$ being the group of proper 3D rotations are listed in the third column.

Defining a $\mathbf{U}(1)$-valued field $e^{i\widehat{\phi}(\mathsf{x})}$ with corresponding current $\partial\widehat{\phi}(\mathsf{x}) = -ie^{-i\widehat{\phi}(\mathsf{x})}\partial_{\mathsf{x}}e^{i\widehat{\phi}(\mathsf{x})}$ and conjugate angular momentum $\widehat{L}(\mathsf{x})$, the Hamiltonian is

$$\widehat{H}^{\mathrm{edge}}_{\mathbf{Z}_N} \to \int \mathrm{d}\mathsf{x}\ \mathsf{C}\,\partial\widehat{\phi}(\mathsf{x})\,\partial\widehat{\phi}(\mathsf{x}) + \mathsf{f}\,\widehat{L}(\mathsf{x})\,\widehat{L}(\mathsf{x}) - \mathsf{M}\cos N\widehat{\phi}(\mathsf{x})\ . \tag{62}$$

The parameters $\mathsf{C}, \mathsf{f}, \mathsf{M}$ are real, and the field is pinned to values in $\mathbf{Z}_N$ in the "semiclassical" limit $\mathsf{M} \to \infty$. This limit induces a variant of the Higgs mechanism (e.g., [108,120] or [121], Sec. IV.A), under which the local $\mathbf{U}(1)$ Hilbert space "breaks" into a $\mathbf{Z}_N$ subspace. A further discretization of the spatial index $\mathsf{x}$ recovers the clock-model Hamiltonian [14].

The *reverse* of the above local symmetry-breaking procedure is effectively an embedding of each $\mathbf{Z}_N$-valued qunit of the clock model into the Hilbert space $\{|\phi\rangle, \phi \in [0, 2\pi) = \mathbf{U}(1)\}$ of a planar rotor, with the potential

$$\widehat{V}^{\mathrm{smooth}}_{\mathbf{Z}_N} = \cos(N\widehat{\phi}) \tag{63}$$

remembering the labels of the original qunit. In other words, one can think of the clock model (9) as a rotor $XY$-model with a strong pinning potential. A subsequent continuum limit and expansion of all operators maps the clock model to a pinned-boson field theory.

## 5.2 Embedding non-Abelian group spaces

The $\mathbf{Z}_N \hookrightarrow \mathbf{U}(1)$ local Hilbert space embedding and subsequent continuum expansion can be extended to the flux ladder using facts about representations of groups [109] and Lie algebras [122]. We describe how to embed the local $\mathbf{G}$-labeled Hilbert space into that of a continuous group $\mathbf{\Gamma}$ that contains $\mathbf{G}$. We restrict $\mathbf{\Gamma}$ to be compact and connected so that there is a discrete set of irreps and so that every group element can be expressed using the Lie algebra of $\mathbf{\Gamma}$. Such a $\mathbf{\Gamma}$ exists[1] for any $\mathbf{G}$, and here we further restrict to two particular families for simplicity —

---

[1] Embedding $\mathbf{G}$ into a compact connected Lie group $\mathbf{\Gamma}$ can be done, e.g., via a permutation group $\mathbf{S}_M$. By Cayley's theorem, there exists an $M$ such that $\mathbf{S}_M$ contains $\mathbf{G}$. Matrices forming the permutation representation of $\mathbf{S}_M$ form a subgroup of $\mathbf{O}(M)$. Such matrices have determinant $\pm 1$, and one can increase their dimension by one and pad the new diagonal entry with $\pm 1$ such that the expanded matrices have determinant one. These expanded matrices then form a subgroup of $\mathbf{\Gamma} = \mathbf{SO}(M + 1)$. For specific groups, one should consider a more efficient embedding; e.g., the quaternions can be embedded directly into $\mathbf{SU}(2)$.

the special orthogonal $[\mathbf{\Gamma} = \mathbf{SO}(M)$ for some $M]$ and special unitary $[\mathbf{\Gamma} = \mathbf{SU}(M)]$ groups. The general $\mathbf{G} \hookrightarrow \mathbf{\Gamma}$ procedure is sketched below. Those wishing for more explicitness may enjoy a parallel description of the $\mathbf{D}_N \hookrightarrow \mathbf{SO}(3)$ example in Appx. C (see also Table 3).

The $\mathbf{\Gamma}$ group space can be defined by the group basis $|R\rangle$ with $R \in \mathbf{\Gamma}$ [49], and the difference from the finite group case is that the canonical basis states are not normalizable: $\langle R|S\rangle = \delta_{RS}^{\mathbf{\Gamma}}$, with the Dirac-like $\delta$ being infinite when $R = S$ and zero otherwise. This space admits $X$-type operators $\{\overleftarrow{X}_R, \overrightarrow{X}_R\}$ similar to those of $\mathbf{G}$. Embedding the latter operators into the space of the former merely corresponds to picking a particular subgroup $\mathbf{G} \subset \mathbf{\Gamma}$, where for any $g \in \mathbf{G}$, there is a corresponding $R(g) \in \mathbf{\Gamma}$. We will abuse notation and use $R$ to refer to elements of $\mathbf{G}$ from now on.

The larger group space also comes with group-valued $Z$-type operators

$$\widehat{U}_\ell = \int_\mathbf{\Gamma} dR \, U_\ell(R) \, |R\rangle \langle R| \,, \tag{64}$$

where $\ell$ labels irreps of $\mathbf{\Gamma}$, $U_\ell(R)$ is an $\ell$-irrep matrix evaluated at $R$, and $dR$ is the Haar measure on $\mathbf{\Gamma}$ satisfying $\int dR \equiv |\mathbf{\Gamma}|$. Naturally, $\{\overrightarrow{X}, \overleftarrow{X}, \widehat{U}\}$ satisfy the Weyl-type relations (15) with $\widehat{Z} \to \widehat{U}$.

When restricted to values in $\mathbf{G}$, the $U_\ell$ matrices form a representation of $\mathbf{G}$, which can be decomposed (i.e., block-diagonalized) into irreps $\lambda$ of $\mathbf{G}$ as

$$U_\ell(R) = \bigoplus_{\lambda, \ell \downarrow \lambda} Z_\lambda(R) \tag{65}$$

for any $R \in \mathbf{G}$. The irrep $\ell$ is said to *branch* into a particular $\lambda$, $\ell \downarrow \lambda$, when $\lambda$ is present in the above decomposition. Conversely, $\widehat{Z}_\lambda$ operators can be embedded into any $\widehat{U}_\ell$,

$$\widehat{Z}_\lambda \hookrightarrow \widehat{U}_\ell \,, \qquad \text{whenever} \qquad \ell \downarrow \lambda \,. \tag{66}$$

There can be infinitely many $\ell$ which branch into a given $\lambda$, making the embedding procedure not unique. For simplicity, one can pick an $\ell$ with smallest dimension.

A given $U_\ell$ can branch to more than one $\lambda$, and projections onto the $\mathbf{G}$-irreps can be used to select the desired operator in the decomposition. The projection operator

$$\Pi_\lambda^\mathbf{G} = \frac{d_\lambda}{|\mathbf{G}|} \sum_{R \in \mathbf{G}} \text{Tr}\left[Z_\lambda\left(R^{-1}\right)\right] U_\ell(R) \tag{67}$$

acts on the $\ell$-irrep space, projecting onto all copies of irrep $\lambda$ of the finite group. For example, when $\ell$ branches to one copy,

$$\Pi_\lambda^\mathbf{G} . U_\ell(R) = Z_\lambda(R) \oplus 0 \,, \tag{68}$$

selecting the corresponding $Z$-operator. The nonnegative integer $\text{Tr}[\Pi_\lambda^\mathbf{G}]/d_\lambda$ counts the number of copies, where $d_\lambda$ is the dimension of the irrep. When there are multiple copies, we can substitute $\Pi_\lambda$ with a projection onto the copy we desire.

Projections can be used to construct a minimal pinning potential,

$$\widehat{V}_\mathbf{G}^{\text{smooth}} = \frac{1}{2} \text{Tr}[\Pi_\mathbf{1}^\mathbf{G} . \widehat{U}_{\underline{\ell}}] + \text{h.c.} \,, \tag{69}$$

where $\Pi$ projects onto a particular copy of the trivial irrep $\lambda = \mathbf{1}$ of $\mathbf{G}$. When $\text{Tr}[\Pi_\mathbf{1}^\mathbf{G} . \widehat{U}_{\underline{\ell}}]$ is evaluated at $R \in \mathbf{G}$, we have

$$\text{Tr}[\Pi_\mathbf{1}^\mathbf{G} . U_{\underline{\ell}}(R)] |R\rangle = \text{Tr}[\Pi_\mathbf{1}^\mathbf{G} . Z_\mathbf{1}(R)] |R\rangle = \text{Tr}[\Pi_\mathbf{1}^\mathbf{G}] |R\rangle = |R\rangle \,. \tag{70}$$

Since $|\text{Tr}[\Pi . U]| \leq \text{Tr}[\Pi]$ for any unitary $U$, the potential is maximized at all elements of $\mathbf{G}$. In order for the potential to be maximized *exclusively* at those elements, we require that

| Dihedral flux ladder | $C_{ab}$ | $f_{ab}$ |
|---|---|---|
| General edge (27) | $\displaystyle\sum_{\lambda \in \mathrm{irr}(\mathbf{D}_N)} \frac{1}{2}\mathrm{TR}\big[L_{\ell_\lambda}^{\mathtt{a}}.C_{\ell_\lambda}^{\mathrm{edge}}.L_{\ell_\lambda}^{\mathtt{b}}\big]$ | $\displaystyle\sum_{g \in \mathbf{D}_N} \frac{1}{2}f_g^{\mathrm{edge}}\phi_g^{\mathtt{a}}\phi_g^{\mathtt{b}}$ |
| Gapped edge ($\mathbf{K} \subseteq \mathbf{D}_N$) (33) | $\displaystyle\sum_{\lambda \in \mathrm{irr}(\mathbf{D}_N)} \frac{d_\lambda}{4N/|\mathbf{K}|}\mathrm{TR}\big[L_{\ell_\lambda}^{\mathtt{a}}.\Pi_{\mathbf{1}}^{\mathbf{K}}.L_{\ell_\lambda}^{\mathtt{b}}\big]$ | $\displaystyle\sum_{k \in \mathbf{K}} \frac{1}{2|\mathbf{K}|}\phi_k^{\mathtt{a}}\phi_k^{\mathtt{b}}$ |
| Self-dual edge (fermionizable) (39) | $\left(\frac{\kappa}{8\pi}\right)^2 \delta_{\mathtt{ab}}$ | $\delta_{\mathtt{ab}}$ |

Table 4: Parameter matrices for the quadratic terms of the $\mathbf{\Gamma} = \mathbf{SO}(3)$ principal chiral model (78), worked out in Appx. C. The *General edge* case is a continuum version of a general $\mathbf{G} = \mathbf{D}_N$ flux ladder with parameters $\{C^{\mathrm{edge}}, f^{\mathrm{edge}}\}$. The *Gapped edge* case is a continuum version of the gapped-edge lattice models from Sec. 3.5. The *Self-dual edge* case is a specific case expressible in terms of fermions via addition of a WZNW term at level $\kappa$ (see Sec. 6).

$\underline{\ell}$ branches into the trivial irrep selected by $\Pi$ without going through the trivial irrep of any intermediate group $\mathbf{G}' \supset \mathbf{G}$. In other words, if there exists a $\mathbf{G}' \supset \mathbf{G}$ and $\tau \in \mathrm{irr}(\mathbf{G}')$ such that $\underline{\ell} \downarrow \tau \downarrow \mathbf{1}$, then $\underline{\ell}$ and $\Pi$ have to be such that $\tau \neq \mathbf{1}$. Without this requirement, the potential would also be maximized at elements of $\mathbf{G}'$. For example, in the case $\mathbf{U}(1) \supset \mathbf{Z}_{2N} \supset \mathbf{Z}_N$, the potential $\cos(2N\widehat{\phi})$, corresponding to the $\mathbf{U}(1)$-irrep $\underline{\ell} = 2N$, is maximized not only at points in $\mathbf{Z}_N$, but also at those in $\mathbf{Z}_{2N}$.

To express $\widehat{V}_{\mathbf{G}}^{\mathrm{sharp}}$ — the explicit projection onto the group subspace — in terms of $\widehat{U}$-operators, we need *all* $\ell$ for which $U_\ell$ admits at least one trivial irrep of $\mathbf{G}$. Compact groups $\mathbf{\Gamma}$ admit an analogue of the finite-group orthogonality relations (21), with the Kronecker-$\delta$ replaced by a Dirac-$\delta$. Using those yields the projection

$$\widehat{V}_{\mathbf{G}}^{\mathrm{sharp}} = \sum_{R \in \mathbf{G}} |R\rangle \langle R| = \sum_{R \in \mathbf{G}} \sum_{\ell \in \mathrm{irr}(\mathbf{\Gamma})} \frac{d_\ell}{|\mathbf{\Gamma}|}\mathrm{TR}\left[U_\ell(R^{-1}).\widehat{U}_\ell\right] = \sum_{\ell \in \mathrm{irr}(\mathbf{\Gamma})} \frac{d_\ell}{|\mathbf{\Gamma}|/|\mathbf{G}|}\mathrm{TR}\left[\Pi_{\mathbf{1}}^{\mathbf{G}}.\widehat{U}_\ell\right], \qquad (71)$$

where $d_\ell$ is the dimension of irrep $\ell$. For the abelian $\mathbf{Z}_N \hookrightarrow \mathbf{U}(1)$ embedding,

$$\widehat{V}_{\mathbf{Z}_N}^{\mathrm{sharp}} = \sum_{h \in \mathbf{Z}_N} \left|\phi = \tfrac{2\pi}{N}h\right\rangle \left\langle \phi = \tfrac{2\pi}{N}h\right| = \sum_{h \in \mathbf{Z}_N} \frac{1}{2\pi}\sum_{\ell \in \mathbf{Z}} e^{i\left(\widehat{\phi} - \frac{2\pi}{N}\ell h\right)} = \frac{1}{2\pi/N}\sum_{\ell \in \mathbf{Z}} e^{iN\ell\widehat{\phi}}. \qquad (72)$$

## 5.3   Flux ladder to principal chiral model

To complete the mapping to a field theory, we express operators $\{\overrightarrow{X}, \overleftarrow{X}, \widehat{U}\}$ as exponentials of the Lie algebra of $\mathbf{\Gamma}$ and expand while taking the continuum limit. The $X$-type operators are expressed in terms of corresponding *momentum currents* $\overrightarrow{L}, \overleftarrow{L}$, e.g.,

$$\overleftarrow{X}_R^{(\mathsf{x})} = e^{-i\phi_R^{\mathtt{a}}(\overleftarrow{L}^{\mathtt{a}})^{(\mathsf{x})}} \approx 1 - i\phi_R^{\mathtt{a}}\overleftarrow{L}^{\mathtt{a}}(\mathsf{x}) + \cdots, \qquad (73)$$

where $\phi_R^{\mathtt{a}}$ are the coordinates of $R$. A similar expansion can be done for $\overrightarrow{X}$. The typewriter-font letters index the generators of the Lie algebra of $\mathbf{\Gamma}$, and all repeated indices are summed. The currents satisfy the commutation relations

$$\left[\overrightarrow{L}^{\mathtt{a}}, \overrightarrow{L}^{\mathtt{b}}\right] = if_{\mathtt{abc}}\overrightarrow{L}^{\mathtt{c}}; \qquad \left[\overleftarrow{L}^{\mathtt{a}}, \overleftarrow{L}^{\mathtt{b}}\right] = if_{\mathtt{abc}}\overleftarrow{L}^{\mathtt{c}}; \qquad \left[\overrightarrow{L}^{\mathtt{a}}, \overleftarrow{L}^{\mathtt{b}}\right] = 0, \qquad (74)$$

where $f_{\mathtt{abc}}$ are the algebra's structure constants.

The group-valued fields $\widehat{U}_\ell$ are expressed as exponentials of irreps of $\boldsymbol{\Gamma}$'s (compact) Lie algebra, whose generators $L_\ell^{\mathsf{a}}$ satisfy

$$[L_\ell^{\mathsf{a}}, L_\ell^{\mathsf{b}}] = i f_{\mathsf{abc}} L_\ell^{\mathsf{c}} \qquad \text{and} \qquad \mathrm{TR}[L_\ell^{\mathsf{a}}.L_\ell^{\mathsf{b}}] = \delta_{\mathsf{ab}} c_\ell \,. \tag{75}$$

The last equation indicates that we have picked generators that are orthogonal under the trace, with the normalization $c_\ell$ known as the Dynkin index. The two-body $ZZ$-type operator can then be embedded into a corresponding product of $\widehat{U}_\ell$'s and expanded as

$$\widehat{\mathcal{Z}}^{\dagger(\mathsf{x})}.\widehat{\mathcal{Z}}^{(\mathsf{x}+1)} \hookrightarrow \widehat{U}^{\dagger(\mathsf{x})}.\widehat{U}^{(\mathsf{x}+1)} \approx 1 + \widehat{U}^{\dagger}.\partial\widehat{U}(\mathsf{x}) + \cdots , \tag{76}$$

where $\partial \equiv \partial_{\mathsf{x}}$. The expansion is expressed in terms of the *position current*

$$\overleftarrow{\partial\phi}_\ell(\mathsf{x}) = -i\widehat{U}_\ell^{\dagger}.\partial\widehat{U}_\ell(\mathsf{x}) = \overleftarrow{\partial\phi}^{\mathsf{a}}(\mathsf{x}) L_\ell^{\mathsf{a}} \qquad \text{and} \qquad \overleftarrow{\partial\phi}^{\mathsf{a}}(\mathsf{x}) = \tfrac{1}{c_\ell}\mathrm{TR}[\overleftarrow{\partial\phi}_\ell(\mathsf{x}).L_\ell^{\mathsf{a}}]. \tag{77}$$

This current acts on both the internal and $\boldsymbol{\Gamma}$ spaces, but its components $\overleftarrow{\partial\phi}^{\mathsf{a}}$ act only on the latter. Note that the orthogonality of the generators and division by the Dynkin index has removed the dependence of the components on the particular irrep $\ell$.

The above expansions are performed under the assumption that the terms vary over space smoothly and slowly. The values of the position current are assumed to change infinitesimally as $\mathsf{x}$ is varied. To expand the momentum current, we assume that the Hamiltonian can only impart an infinitesimal change in the group-valued position at any $\mathsf{x}$. Combined with a pinning potential, the quadratic terms yield a generalized principal chiral model associated with $\boldsymbol{\Gamma}$,

$$\boxed{\widehat{H}_{\mathbf{G}}^{\mathrm{edge}} \to \int \mathrm{d}\mathsf{x}\, \mathsf{C}_{\mathsf{ab}} \overleftarrow{\partial\phi}^{\mathsf{a}}(\mathsf{x}) \overleftarrow{\partial\phi}^{\mathsf{b}}(\mathsf{x}) + \mathsf{f}_{\mathsf{ab}} \overleftarrow{L}^{\mathsf{a}}(\mathsf{x}) \overleftarrow{L}^{\mathsf{b}}(\mathsf{x}) - \tfrac{1}{2}\mathsf{M}\big(\mathrm{TR}\big[\Pi_{\mathbf{1}}^{\mathbf{G}}.\widehat{U}_{\underline{\ell}}(\mathsf{x})\big] + \mathrm{h.c.}\big)} . \tag{78}$$

In the $\mathsf{M} \to \infty$ limit, the pinning potential (69) pins the $\boldsymbol{\Gamma}$-valued Hilbert space to the $\mathbf{G}$-valued subspace at each spatial point $\mathsf{x}$.

The quadratic terms in the above field theory are symmetric under

$$\overrightarrow{X}_R^{(\mathrm{global})} \equiv e^{-i\phi_R^{\mathsf{a}}(\overrightarrow{L}^{\mathsf{a}})^{(\mathrm{global})}} , \qquad \text{generated by} \qquad \overrightarrow{L}^{(\mathrm{global})} \equiv \int \mathrm{d}\mathsf{x}\, \overrightarrow{L}(\mathsf{x}) \tag{79}$$

for any $R \in \boldsymbol{\Gamma}$ and any values of the parameter matrices C and f. The pinning potential is only invariant under $R \in \mathbf{G}$, so the full theory has only $\mathbf{G}$-symmetry. Bulk anyonic data, associated with treating the flux ladder as an edge theory for a $\mathbf{G}$ quantum double, also carry over to the continuum. The bulk ground-state subspace is fixed by $\overrightarrow{X}_R^{(\mathrm{global})} = 1$ for all $R \in \mathbf{G}$ (31); more general bulk constraints are discussed in Appx. A.

The other global operations $\overleftarrow{X}_R^{(\mathrm{global})}$ are not guaranteed to be symmetries of the theory. Both currents transform under these operations in the *adjoint representation*,

$$\overleftarrow{X}_R^{\dagger(\mathrm{global})}\overleftarrow{\partial\phi}^{\mathsf{a}}\overleftarrow{X}_R^{(\mathrm{global})} = R_{\mathsf{ab}}\overleftarrow{\partial\phi}^{\mathsf{b}} \qquad \text{and} \qquad \overleftarrow{X}_R^{\dagger(\mathrm{global})}\overleftarrow{L}^{\mathsf{a}}\overleftarrow{X}_R^{(\mathrm{global})} = R_{\mathsf{ab}}\overleftarrow{L}^{\mathsf{b}} , \tag{80}$$

which quantifies how the group rotates elements of its own Lie algebra. The $R_{\mathsf{ab}}$ is a real-valued matrix element of an orthogonal rotation, defined as

$$R_{\mathsf{ab}} = \tfrac{1}{c_\ell}\mathrm{TR}\big[U_\ell^{\dagger}.L_\ell^{\mathsf{a}}.U_\ell.L_\ell^{\mathsf{b}}\big]. \tag{81}$$

The definition is independent of which irreps $\ell$ one picks, as long as it is nontrivial.

Besides yielding a heuristic field theory, the mapping can be made explicit, with parameters of the flux ladder directly absorbed into the matrices $\mathsf{C}_{\mathsf{ab}}$ and $\mathsf{f}_{\mathsf{ab}}$. We have performed the explicit procedure for the case $\mathbf{D}_N \hookrightarrow \mathbf{SO}(3)$ in Appx. C, and the resulting parameters are shown in Table 4 for the three cases studied in Sec. 3. While we should not assign too much importance to potentially numerological values of $\{\mathsf{C}_{\mathsf{ab}}, \mathsf{f}_{\mathsf{ab}}\}$, the procedure nevertheless helps us identify general regions in the large parameter space of the field theory that may be related to particular edges of quantum double models.

# 6 Quantum wire constructions

In the previous section, we developed a continuum version of the flux ladder, an effective lattice model for the quantum-double edge. In this section, we recast a particular instance of the version in terms of fermionic fields. This result makes first contact between the quantum double edge and quantum wires, with the long-term goal of realizing continuum analogues of quantum-double non-Abelian defects (beyond Majorana and parafermion zero modes) in electronic systems.

Continuum analogues of zero modes for the abelian $\mathbf{Z}_N$ case have been identified at interfaces between superconducting and ferromagnetic phases of the edge of a fractional-quantum-Hall medium [14–17]. The interface is modeled by a segment of the free-boson field theory (62), recast as a quantum wire with the help of bosonization [19–21, 92, 95, 97–99]. The influence of the two phases corresponds to different boundary conditions on the boson's fields at each end of the segment. The quantum-wire field theory is then solved (see, e.g., [16]), and continuum zero modes are identified within the ground-state subspace.

Our strategy is to generalize the above ingredients, namely, the segment of field theory and its boundary conditions, by viewing the original construction through the lens of representation theory. While we set the stage for generalizing results of the abelian case, we stop short of solving the resulting theory and constructing zero modes, leaving such investigations to future work.

## 6.1 The free boson as a quantum wire

Both the lattice and continuum versions of the clock model can be recast in terms of quantum-wire degrees of freedom, one of which is nonlocal relative to the spin-chain geometry [43, 91, 114, 115, 117–119]. The continuum version utilizes $\widehat{\phi}$, corresponding to the order parameter $\widehat{Z}$ on the lattice, as well as the collective string excitation

$$\widehat{\theta}(x) = \int^x dy \, \widehat{L}(y) \,, \tag{82}$$

which corresponds to the "disorder operator" $\cdots \widehat{X}^{(x-1)} \widehat{X}^{(x)}$ on the lattice (see Sec. 3). The relative nonlocality between the two "dual" fields is manifest in their commutation relations. Using the canonical commutator between $e^{i\widehat{\phi}}$ and $\widehat{L}$,

$$[e^{i\widehat{\phi}(x)}, \widehat{L}(y)] = -e^{i\widehat{\phi}(x)} \delta(x-y) \tag{83}$$

and letting $\Theta$ be the Heaviside step function yields $[e^{i\widehat{\phi}(x)}, \widehat{\theta}(y)] = -e^{i\widehat{\phi}(x)} \Theta(y-x)$. When written in terms of $\{\widehat{\phi}, \widehat{\theta}\}$, the kinetic energy term in the free-boson field theory becomes $\widehat{L}^2 \to (\partial \widehat{\theta})^2$.

"Quasiparticle" combinations $\exp[i\pi(\widehat{\phi} \pm \widehat{\theta})]$ of the dual variables for the $\mathbf{Z}_2$ case represent left- and right-moving Dirac fermionic fields. These are used to *fermionize* the free boson, i.e., express the quadratic terms of the field theory in terms of the quantum-wire degrees of freedom. The $\mathbf{Z}_{N>2}$ case is recast in similar fashion via the slight modification $\pi \to 2\pi/N$ in the quasiparticles' exponential. The resulting operators form an algebra that is equivalent to that of the edge excitations of the fractional quantum Hall phase. Such operators provide a quantum-wire formulation for the general $N$ case that is mathematically a close cousin of the $N = 2$ case (albeit physically being quite different due to the necessary fractional-quantum-Hall bulk).

When in proximity to a ferromagnet, quasiparticles are assumed to interact via a backscattering term $\cos(N\widehat{\phi})$, which we recognize as none other than the $\mathbf{Z}_N$ pinning potential $\widehat{V}_{\mathbf{Z}_N}^{\text{smooth}}$

(63) of the field theory. The other prominent potential in such systems is a superconducting pairing term

$$-M\widehat{V}_{\text{SC}} = -M\cos\left(\tfrac{2\pi}{N}\widehat{\theta}\right),$$ (84)

"dual" to $\cos(N\widehat{\phi})$ in that it pins the collective momentum $\widehat{\theta}$ as $M \to \infty$. Zero modes exist at interfaces between strongly-pinned ferromagnetic regions (the free boson with $-M\widehat{V}_{\mathbf{Z}_N}^{\text{smooth}}$ as $M \to \infty$) and superconducting regions (the free boson with $-M\widehat{V}_{\text{SC}}$ as $M \to \infty$) [14–17]. An interface between these two regions consists of a segment of the free boson, and the two strongly-pinned regions manifest themselves as boundary conditions on either side of the segment [see Fig. 2(a)]. The free boson is diagonalized[2], and continuum zero-mode operators are obtained via the bosonization techniques we outline below. Such zero modes should be thought of as continuum analogues of Jordan-Wigner zero modes of the lattice clock model from Sec. 4.

Generalizing the zero modes of the $\mathbf{Z}_N$ case, we would like to realize non-Abelian defects associated with quantum double models on edges of electronic topological phases described by continuum degrees of freedom. Using the continuum mapping of the previous section, we proceed to generalize the ingredients required for the $\mathbf{Z}_N$ case, namely, the two "dual" potentials and a segment of fermionizable field theory [see Fig. 2(b)]. For the ferromagnetic potential $\widehat{V}_{\mathbf{Z}_N}^{\text{smooth}}$, we substitute the **G** pinning term $\widehat{V}_{\mathbf{G}}^{\text{smooth}}$ (126). In Sec. 6.2, we show how a strong pairing potential $\widehat{V}_{\text{SC}}$ is equivalent to projecting onto the quotient subspace $\mathbf{U}(1)/\mathbf{Z}_N$, and introduce an analogous $\mathbf{\Gamma}$ potential that pins sites to $\mathbf{\Gamma}/\mathbf{G}$. After reviewing how to fermionize the free boson in Sec. 6.3, we show in Sec. 6.4 how to fermionize the quadratic part of the principal chiral model. Differing from the abelian case, such fermionization requires an additional Wess-Zumino-Novikov-Witten term [93–96]. This term is $\overrightarrow{X}^{(\text{global})}$-symmetric — satisfying the symmetry requirements for belonging to a general quantum-double edge — but we leave identification of any connections to quantum-double physics to future work.

## 6.2 Generalized pairing potentials

In the limit of infinitely strong pinning ($M \to \infty$), the effect of the superconducting pairing potential (84) is equivalent to that of the local potential

$$\widehat{V}_{\mathbf{U}(1)/\mathbf{Z}_N}^{\text{smooth}} = \cos\left(\tfrac{2\pi}{N}\widehat{L}\right).$$ (85)

To show this, consider the lattice version of $\widehat{V}_{\text{SC}}$, acting as $\cos(\tfrac{2\pi}{N}\widehat{\theta}^{(\mathsf{x})})$ on each site $\mathsf{x}$. Recalling Eq. (82), $\widehat{\theta}^{(1)} = \widehat{L}^{(1)}$ for the first site $\mathsf{x} = 1$, so $\widehat{V}_{\text{SC}}^{(1)} = \widehat{V}_{\mathbf{U}(1)/\mathbf{Z}_N}^{\text{smooth }(1)}$. When the first site is strongly pinned, its momentum $\widehat{L}^{(1)}$ is a multiple of $N$. Plugging this into the potential for the next site $\mathsf{x} = 2$,

$$\widehat{V}_{\text{SC}}^{(\mathsf{x}=2)} = \cos\left(\tfrac{2\pi}{N}\widehat{\theta}^{(2)}\right) = \cos\left(\tfrac{2\pi}{N}\widehat{L}^{(1)} + \tfrac{2\pi}{N}\widehat{L}^{(2)}\right) = \cos\left(\tfrac{2\pi}{N}\widehat{L}^{(2)}\right) = \widehat{V}_{\mathbf{U}(1)/\mathbf{Z}_N}^{\text{smooth }(2)}.$$ (86)

The same reasoning holds recursively for sites $\mathsf{x} \geq 3$.

As $M \to \infty$, $-MV_{\mathbf{U}(1)/\mathbf{Z}_N}$ is minimized when the integer-valued angular momentum $\widehat{L}$ is a multiple of $N$. In terms of rotor momentum states $|\ell\rangle = \frac{1}{\sqrt{2\pi}}\int_0^{2\pi} d\phi\, e^{i\ell\phi}|\phi\rangle$ [107], the pinned subspace is thus $\{|\ell = Ns\rangle, s \in \mathbf{Z}\}$. Performing the Fourier transform yields the quotient states (36),

$$|a\mathbf{Z}_N\rangle = \sqrt{\tfrac{N}{2\pi}}\sum_{s\in\mathbf{Z}} e^{-iNsa}|\ell = Ns\rangle = \frac{1}{\sqrt{N}}\sum_{h\in\mathbf{Z}_N}\left|\phi = a + \tfrac{2\pi}{N}h\right\rangle,$$ (87)

labeled by $a \in [0, \tfrac{2\pi}{N}) = \mathbf{U}(1)/\mathbf{Z}_N$. This position state basis reveals that the potential (85) pins the rotor to its *quotient subspace* $\mathbf{U}(1)/\mathbf{Z}_N$. A sharper potential uses the sum of projections

onto the basis states $|a\mathbf{Z}_N\rangle$ themselves,

$$\widehat{V}^{\text{sharp}}_{\mathbf{U}(1)/\mathbf{Z}_N} = \int_{\mathbf{U}(1)/\mathbf{Z}_N} \mathrm{d}a\, |a\mathbf{Z}_N\rangle\langle a\mathbf{Z}_N| = \frac{1}{N}\int_{\mathbf{U}(1)} \mathrm{d}\phi\, |\phi\mathbf{Z}_N\rangle\langle\phi\mathbf{Z}_N| \tag{88a}$$

$$= \frac{1}{N^2}\sum_{h\in\mathbf{Z}_N} e^{-i\frac{2\pi}{N}h\widehat{L}} \left(\int_{\mathbf{U}(1)} \mathrm{d}\phi\, |\phi\rangle\langle\phi|\right)\sum_{k\in\mathbf{Z}_N} e^{i\frac{2\pi}{N}k\widehat{L}} = \frac{1}{N}\sum_{h\in\mathbf{Z}_N} e^{-i\frac{2\pi}{N}h\widehat{L}}. \tag{88b}$$

This interpretation provides an avenue for straightforward extension.

The quotient subspace idea naturally generalizes to $\mathbf{G} \subset \mathbf{\Gamma}$, where the relevant subspace at each site is spanned by coset states $|a\mathbf{G}\rangle$ (36) with $a \in \mathbf{\Gamma}/\mathbf{G}$. These states are invariant under $\overleftarrow{X}_R$ for any $R \in \mathbf{G}$, as its action merely reshuffles the terms in the superposition. The sharp potential — a projection onto the quotient subspace — can be expressed as an equal superposition of all $\overleftarrow{X}_R$ [49, 123]; following a derivation similar to that of $\widehat{V}^{\text{smooth}}_{\mathbf{U}(1)/\mathbf{Z}_N}$ (88b),

$$\widehat{V}^{\text{sharp}}_{\mathbf{\Gamma}/\mathbf{G}} = \int_{\mathbf{\Gamma}/\mathbf{G}} \mathrm{d}a\, |a\mathbf{G}\rangle\langle a\mathbf{G}| = \frac{1}{|\mathbf{G}|}\sum_{R\in\mathbf{G}} \overleftarrow{X}_R. \tag{89}$$

For a smooth potential for the $\mathbf{G} = \mathbf{D}_N$ case, it suffices to use only the generating elements. Picking a dihedral subgroup according to the prescription from Appx. C.1,

$$\widehat{V}^{\text{smooth}}_{\mathbf{SO}(3)/\mathbf{G}} = \frac{1}{2}\cos\left(\frac{2\pi}{N}\overleftarrow{L}^{\mathsf{z}}\right) + \cos\left(\pi\overleftarrow{L}^{\mathsf{x}}\right). \tag{90}$$

These quotient-space potentials convert the principal chiral model associated with $\mathbf{\Gamma}$ (78) into a nonlinear sigma model on $\mathbf{\Gamma}/\mathbf{G}$. For example, projecting the $\mathbf{SO}(3)$ model (130) onto $\mathbf{SO}(3)/\mathbf{U}(1) = \mathbf{S}^2$ yields the well-known $\mathbf{S}^2$ nonlinear sigma model (a.k.a. the $\mathbf{O}(3)$-rotor model [79]).

## 6.3 Abelian bosonization

Bosonization allows one to convert between specific bosonic and fermionic field theories, stemming from the ability to express fields $\widehat{J}_{\text{KM}}(\mathsf{x})$ (a.k.a. *currents* or *vertex operators*) of a certain Lie algebra — a *Kac-Moody (KM) algebra*[2] — in terms of either bosonic or fermionic fields. Because of this, any bosonic fields satisfying the commutation relations of KM currents can also be written in terms of fermionic fields, and vice versa. This ability allows one to interpret the same Hilbert space in two remarkably different ways — either as the space of excitations of bosons or of fermions.

A level-$\kappa$ $\mathbf{U}(1)$ KM algebra corresponds to one current with commutation relation

$$\left[\widehat{J}_{\text{KM}}(\mathsf{x}),\widehat{J}_{\text{KM}}(\mathsf{y})\right] = i\frac{\kappa}{2\pi}\partial_{\mathsf{x}}\delta(\mathsf{x}-\mathsf{y}) \equiv i\frac{\kappa}{2\pi}\delta'(\mathsf{x}-\mathsf{y}), \tag{91}$$

where the integer parameter $\kappa$ is called the *level*. The $\delta$-function derivative may seem unruly to the uninitiated, but we soon show that, on the bosonic side, it arises naturally due to presence of the derivative in the current $\partial\widehat{\phi}$.

We need two copies of a $\mathbf{U}(1)$ KM algebra, $\widehat{J}^{\pm}_{\text{KM}}$ with levels $\kappa = \pm 1$, to fermionize the free boson (62). We show that superpositions of the fields $\partial\widehat{\phi}, \widehat{L}$ are exactly these currents. Recalling the canonical commutator (83) and applying $\partial$ yields

$$\left[\partial e^{i\widehat{\phi}(\mathsf{x})},\widehat{L}(\mathsf{y})\right] = \partial_{\mathsf{x}}\left[e^{i\widehat{\phi}(\mathsf{x})},\widehat{L}(\mathsf{y})\right] = -\left(\partial_{\mathsf{x}}e^{i\widehat{\phi}(\mathsf{x})}\right)\delta(\mathsf{x}-\mathsf{y}) - e^{i\widehat{\phi}(\mathsf{x})}\delta'(\mathsf{x}-\mathsf{y}). \tag{92}$$

---

[2] The field $\widehat{J}_{\text{KM}}(\mathsf{x})$ is the generating series [124] of the KM algebra. With $\mathsf{x}$ assumed to lie on a circle, the actual KM algebra generators are its integer-labeled Fourier components [98, 99, 125]; these components are the "normal modes" that diagonalize the quantum wire. Their algebra is infinite-dimensional (as opposed to the angular momentum algebra, which is three-dimensional).

Expressing $\partial\widehat{\phi} = -ie^{-i\widehat{\phi}}\partial e^{i\widehat{\phi}}$, using the commutator identity $[A, BC] = B[A, C] + [A, B]C$, and plugging in the above yields

$$\left[\partial\widehat{\phi}(x), \widehat{L}(y)\right] = -i\left[e^{-i\widehat{\phi}(x)}\partial_x e^{i\widehat{\phi}(x)}, \widehat{L}(y)\right] = i\delta'(x - y) . \tag{93}$$

The $\delta$-functions cancel, leaving only the required $\delta'$. Defining

$$\widehat{J}^{\pm}_{\text{KM}} = \frac{1}{2\sqrt{\pi}}(\partial\widehat{\phi} \pm \widehat{L}) \tag{94}$$

yields the desired relations (91) for each current. These currents can then be used to express the free boson (62).

To complete the dictionary, we express the KM currents as fermionic bilinears. Let $\widehat{\psi}_{\pm}(x)$ be left- and right-moving Dirac fermion fields, satisfying canonical anti-commutation relations $\{\widehat{\psi}_p(x), \widehat{\psi}^{\dagger}_q(y)\} = \delta_{pq}\delta(x - y)$ and $\{\widehat{\psi}_p(x), \widehat{\psi}_q(y)\} = 0$ for $p, q \in \pm$. To show that the bilinears $\widehat{\psi}^{\dagger}_{\pm}\widehat{\psi}_{\pm}$ are indeed representations of the KM currents (91), we have to take care of infinities arising from products of time-evolved fields ( [98], Sec. 2.II):

$$\widehat{\psi}_p(x, t)\,\widehat{\psi}^{\dagger}_q(y, 0) \sim \frac{1}{2\pi}\frac{\delta_{pq}}{t - i(x - y)} \qquad \text{as } y \to x \text{ and } t \to 0 . \tag{95}$$

This divergence is the reason behind the required $\delta'$ — the "chiral anomaly" [98] or "Schwinger term" [99, 126] — in the equal-time commutator of the fermion bilinears.

The bosonization dictionary allows us to fermionize $\partial\widehat{\phi}, \widehat{L}$ as well as the group field $e^{i\widehat{\phi}}$:

$$\frac{1}{2\sqrt{\pi}}\left(\partial\widehat{\phi} + \widehat{L}\right) \leftrightarrow \widehat{\psi}^{\dagger}_+\widehat{\psi}_+ \; ; \qquad \frac{1}{2\sqrt{\pi}}\left(\partial\widehat{\phi} - \widehat{L}\right) \leftrightarrow \widehat{\psi}^{\dagger}_-\widehat{\psi}_- \; ; \qquad \frac{2}{\sqrt{\pi}}e^{i\widehat{\phi}} \leftrightarrow \widehat{\psi}^{\dagger}_+\widehat{\psi}_- . \tag{96}$$

These fermionic fields can alternatively be expressed as exponentials of $\widehat{\phi}$ and the nonlocal $\widehat{\theta}$ field (82).

## 6.4 Non-Abelian bosonization

Here we apply a "non-abelian" version of the bosonization procedure from the previous section to determine when the quadratic terms in the principal chiral model (78) can be expressed in terms of fermions. This version uses two copies of a $\boldsymbol{\Gamma}$ Kac-Moody (KM) algebra at levels $\pm\kappa$, whose "left" and "right" currents respectively satisfy

$$\left[\overrightarrow{J}^{\text{a}}_{KM}(x), \overrightarrow{J}^{\text{b}}_{KM}(y)\right] = if_{\text{abc}}\overrightarrow{J}^{\text{c}}_{KM}(x)\delta(x - y) + i\frac{\kappa}{4\pi}\delta_{\text{ab}}\delta'(x - y) \tag{97a}$$

$$\left[\overleftarrow{J}^{\text{a}}_{KM}(x), \overleftarrow{J}^{\text{b}}_{KM}(y)\right] = if_{\text{abc}}\overleftarrow{J}^{\text{c}}_{KM}(x)\delta(x - y) - i\frac{\kappa}{4\pi}\delta_{\text{ab}}\delta'(x - y) , \tag{97b}$$

while commuting with each other. We let $\boldsymbol{\Gamma}$ be either the special unitary or special orthogonal group, with structure constants $f_{\text{abc}}$. As before, these currents have representations in terms of the bosonic fields of the principal chiral model as well as fermionic fields.

Mimicking Sec. 6.3, we proceed to identify the $\delta'$ term required for the KM currents. Since we are in a non-Abelian case, $\partial U.U^{\dagger} \neq U^{\dagger}.\partial U$, so there is a position current complementary to $\overleftarrow{\partial\phi}$ (77):

$$\overrightarrow{\partial\phi}_{\ell}(x) = i\partial\widehat{U}_{\ell}.\widehat{U}^{\dagger}_{\ell}(x) = \overrightarrow{\partial\phi}^{\text{a}}L^{\text{a}}_{\ell} \qquad \text{with} \qquad \overrightarrow{\partial\phi}^{\text{a}}(x) = \frac{1}{c_{\ell}}\text{Tr}\left[\overrightarrow{\partial\phi}_{\ell}(x).L^{\text{a}}_{\ell}\right]. \tag{98}$$

We summarize the algebra of formed by $\{\overrightarrow{\partial\phi}, \overleftarrow{\partial\phi}, \overrightarrow{L}, \overleftarrow{L}\}$ in Table 5, describing some relevant parts below.

| | $\overrightarrow{\partial\phi}^{\mathsf{b}}(\mathsf{y})$ | $\overleftarrow{\partial\phi}^{\mathsf{b}}(\mathsf{y})$ | $\overrightarrow{L}^{\mathsf{b}}(\mathsf{y})$ | $\overleftarrow{L}^{\mathsf{b}}(\mathsf{y})$ |
|---|---|---|---|---|
| $\overrightarrow{\partial\phi}^{\mathsf{a}}(\mathsf{x})$ | 0 | 0 | $if_{\mathsf{abc}}\overrightarrow{\partial\phi}^{\mathsf{c}}(\mathsf{x})\delta(\mathsf{x}-\mathsf{y})+i\delta_{\mathsf{ab}}\delta'(\mathsf{x}-\mathsf{y})$ | $-i\widehat{R}_{\mathsf{ab}}(\mathsf{x})\delta'(\mathsf{x}-\mathsf{y})$ |
| $\overleftarrow{\partial\phi}^{\mathsf{a}}(\mathsf{x})$ | $\star$ | 0 | $-i\widehat{R}_{\mathsf{ba}}(\mathsf{x})\delta'(\mathsf{x}-\mathsf{y})$ | $if_{\mathsf{abc}}\overleftarrow{\partial\phi}^{\mathsf{c}}(\mathsf{x})\delta(\mathsf{x}-\mathsf{y})+i\delta_{\mathsf{ab}}\delta'(\mathsf{x}-\mathsf{y})$ |
| $\overrightarrow{L}^{\mathsf{a}}(\mathsf{x})$ | $\star$ | $\star$ | $if_{\mathsf{abc}}\overrightarrow{L}^{\mathsf{c}}(\mathsf{x})\delta(\mathsf{x}-\mathsf{y})$ | 0 |
| $\overleftarrow{L}^{\mathsf{a}}(\mathsf{x})$ | $\star$ | $\star$ | $\star$ | $if_{\mathsf{abc}}\overleftarrow{L}^{\mathsf{c}}(\mathsf{x})\delta(\mathsf{x}-\mathsf{y})$ |

Table 5: Commutation relations between currents $\overrightarrow{L}, \overleftarrow{L}$ (74) and $\overrightarrow{\partial\phi}, \overleftarrow{\partial\phi}$ (77,98). Each entry lists the commutator $[A, B]$ of an operator $A$ from the first column and $B$ from the top row.

Lie-algebraic versions of the Weyl-type relations (15), obtained by expanding $\{\overrightarrow{X}, \overleftarrow{X}\}$ in terms of $\{\overrightarrow{L}, \overleftarrow{L}\}$ (74), are

$$\left[\overrightarrow{L}^{\mathsf{a}}(\mathsf{x}), \widehat{U}_{\ell}(\mathsf{y})\right] = -L_{\ell}^{\mathsf{a}}.\widehat{U}_{\ell}(\mathsf{x})\,\delta(\mathsf{x}-\mathsf{y}) \tag{99a}$$

$$\left[\overleftarrow{L}^{\mathsf{a}}(\mathsf{x}), \widehat{U}_{\ell}(\mathsf{y})\right] = +\widehat{U}_{\ell}(\mathsf{x}).L_{\ell}^{\mathsf{a}}\,\delta(\mathsf{x}-\mathsf{y})\,. \tag{99b}$$

To determine the commutator of $\overrightarrow{L}$ and $\overrightarrow{\partial\phi}$, we generalize the derivation of Eq. (92), using Hilbert-Schmidt orthogonality of $L_{\ell}^{\mathsf{a}}$ (75) and the definitions (98) along the way. The result is presented in Table 5, with the crucial $\delta'$ appearing due to the derivative in $\overrightarrow{\partial\phi}$. A similar procedure yields $[\overleftarrow{L}, \overleftarrow{\partial\phi}]$ in the table.

Rounding out Table 5, the calculation of the "cross" commutator $[\overleftarrow{L}, \overrightarrow{\partial\phi}]$ involves $\widehat{R}_{\mathsf{ab}} = (\widehat{R}_{\mathsf{ab}})^{\dagger}$, which yields the real-valued matrix element $R_{\mathsf{ab}}$ of the adjoint representation from Eq. (81) when applied to a state $|R\rangle$ for $R \in \Gamma$. This operator converts between left and right currents [51, 127],

$$\overrightarrow{L}^{\mathsf{a}} = -\widehat{R}_{\mathsf{ab}}\overleftarrow{L}^{\mathsf{b}} \qquad \text{and} \qquad \overrightarrow{\partial\phi}^{\mathsf{a}} = -\widehat{R}_{\mathsf{ab}}\overleftarrow{\partial\phi}^{\mathsf{b}}\,. \tag{100}$$

By defining momentum currents $\overrightarrow{L}_{\ell} \equiv \overrightarrow{L}^{\mathsf{a}}L_{\ell}^{\mathsf{a}}$ and $\overleftarrow{L}_{\ell} = \overleftarrow{L}^{\mathsf{a}}L_{\ell}^{\mathsf{a}}$, analogously to $\overrightarrow{\partial\phi}$ (77) and $\overrightarrow{\partial\phi}$ (98), the above conversion is equivalent to $\overrightarrow{L}_{\ell} = -\widehat{U}_{\ell}.\overleftarrow{L}_{\ell}.\widehat{U}_{\ell}^{\dagger}$ and $\overrightarrow{\partial\phi}_{\ell} = -\widehat{U}_{\ell}.\overleftarrow{\partial\phi}_{\ell}.\widehat{U}_{\ell}^{\dagger}$. This relationship between the currents makes contact with the classical version of our field theory, where left currents $\overrightarrow{L}_{\ell} \leftrightarrow i\partial_{\mathsf{t}}\widehat{U}_{\ell}.\widehat{U}_{\ell}^{\dagger}$ are the time-like versions of the space-like $\overrightarrow{\partial\phi}_{\ell} \leftrightarrow i\partial_{\mathsf{x}}\widehat{U}_{\ell}.\widehat{U}_{\ell}^{\dagger}$ (and similarly for the right currents). The Poisson brackets of all currents ( [128], Part Two, Ch. 1, Sec. 5) are identical to the commutators from Table 5. This correspondence serves as a key starting point for constructing the KM currents.

With the $\delta'$ in hand, one final ingredient is required to identify the KM currents for this case, namely, the commutators in Table 5 have to be slightly modified. On the classical Lagrangian level, this modification occurs to the Poisson brackets of the angular momenta $\{\overrightarrow{L}, \overleftarrow{L}\}$ ( [128], Part Two, Ch. 1, Sec. 5), and stems from an additional Wess-Zumino-Novikov-Witten (WZNW) term [93–96] that converts the principal chiral model into a *WZNW model*. On the quantum level, this translates to modifying the momentum commutators (74) to

$$\left[\overrightarrow{L}^{\mathsf{a}}(\mathsf{x}), \overrightarrow{L}^{\mathsf{b}}(\mathsf{y})\right] = if_{\mathsf{abc}}\left(\overrightarrow{L}^{\mathsf{c}}(\mathsf{x}) - \frac{\kappa}{8\pi}\overrightarrow{\partial\phi}^{\mathsf{c}}(\mathsf{x})\right)\delta(\mathsf{x}-\mathsf{y}) \tag{101a}$$

$$\left[\overleftarrow{L}^{\mathsf{a}}(\mathsf{x}), \overleftarrow{L}^{\mathsf{b}}(\mathsf{y})\right] = if_{\mathsf{abc}}\left(\overleftarrow{L}^{\mathsf{c}}(\mathsf{x}) + \frac{\kappa}{8\pi}\overleftarrow{\partial\phi}^{\mathsf{c}}(\mathsf{x})\right)\delta(\mathsf{x}-\mathsf{y})\,. \tag{101b}$$

All other commutators remain the same. In order to fermionize, we have to assume that the angular momenta $\{\overrightarrow{L}, \overleftarrow{L}\}$ used to construct the field theory (78) satisfy these relations. With

this assumption, the following superpositions satisfy the KM relations (97),

$$\overrightarrow{J}_{KM}^{\mathsf{a}} = \overrightarrow{L}^{\mathsf{a}} + \frac{\kappa}{8\pi} \overrightarrow{\partial \phi}^{\mathsf{a}} \qquad \text{and} \qquad \overleftarrow{J}_{KM}^{\mathsf{a}} = \overleftarrow{L}^{\mathsf{a}} - \frac{\kappa}{8\pi} \overleftarrow{\partial \phi}^{\mathsf{a}} . \qquad (102)$$

We now turn to the fermionic side. Expressing KM currents as fermion bilinears requires $M$ species [recall that $\mathbf{\Gamma}$ is either $\mathbf{SO}(M)$ or $\mathbf{SU}(M)$] of left and right fermion operators $\{\overrightarrow{\psi}_m, \overleftarrow{\psi}_m\}$. These satisfy

$$\left\{ \overrightarrow{\psi}_m(\mathsf{x}), \overrightarrow{\psi}_n^\dagger(\mathsf{y}) \right\} = \delta_{mn}\delta(\mathsf{x}-\mathsf{y}) \qquad \text{and} \qquad \left\{ \overleftarrow{\psi}_m(\mathsf{x}), \overleftarrow{\psi}_n^\dagger(\mathsf{y}) \right\} = \delta_{mn}\delta(\mathsf{x}-\mathsf{y}) \, , \quad (103)$$

with all others anticommuting with each other. Recalling from Sec. 6.3 the nuances of fermionic products, the fermionic representation is [97–99]

$$\overrightarrow{J}_{KM}^{\mathsf{a}} \leftrightarrow \sum_{m,n} \overrightarrow{\psi}_m^\dagger [L_{\ell=1}^{\mathsf{a}}]_{mn} \overrightarrow{\psi}_n \equiv \overrightarrow{\psi}^\dagger . L_1^{\mathsf{a}} . \overrightarrow{\psi} \quad ; \quad \overleftarrow{J}_{KM}^{\mathsf{a}} \leftrightarrow \overleftarrow{\psi}^\dagger . L_1^{\mathsf{a}} . \overleftarrow{\psi} \quad ; \quad [\widehat{U}_1]_{mn} \leftrightarrow \overrightarrow{\psi}_m^\dagger \overleftarrow{\psi}_n ,$$

$$(104)$$

where $\ell = 1$ is the *defining irrep* of $\mathbf{\Gamma}$ (e.g., for $\mathbf{\Gamma} = \mathbf{SO}(M)$, the $M$-dimensional irrep of orthogonal rotation matrices).

## 6.5 Generalized interfaces

The $\mathbf{\Gamma}$ principal chiral model is not fermionizable for all parameters $\{C, f\}$, unlike the free boson. We have, however, identified a point $(\frac{8\pi}{\kappa})^2 C_{\mathsf{ab}} = f_{\mathsf{ab}} = \delta_{\mathsf{ab}}$ for which the quadratic terms of the theory can be recast in terms of fermions; we describe this procedure below. We believe that this point is related to the self-dual edge case $\widehat{H}_{\mathbf{G}}^{\mathsf{sd}}$ of the flux ladder lattice model (see Sec. 3.6 and Table 4).

The principal chiral model at the critical point consists of products $(\overleftarrow{L})^2$ and $(\overrightarrow{\partial \phi})^2$, while the KM generators contain both $\{\overleftarrow{L}, \overleftarrow{\partial \phi}\}$ and $\{\overrightarrow{L}, \overrightarrow{\partial \phi}\}$. However, convertibility (100) between currents, along with Hermiticity of all operators involved, allows us to "change arrows" of any summed quadratic combination of $A, B \in \{L, J\}$,

$$\overrightarrow{A}^{\mathsf{a}} \overrightarrow{B}^{\mathsf{a}} = \widehat{R}_{\mathsf{ab}} \overleftarrow{A}^{\mathsf{b}} \widehat{R}_{\mathsf{ac}} \overleftarrow{B}^{\mathsf{c}} = \overleftarrow{A}^{\mathsf{b}} \widehat{R}_{\mathsf{ab}} \widehat{R}_{\mathsf{ac}} \overleftarrow{B}^{\mathsf{c}} = \overleftarrow{A}^{\mathsf{b}} \widehat{R^{-1}}_{\mathsf{ba}} \widehat{R}_{\mathsf{ac}} \overleftarrow{B}^{\mathsf{c}} = \overleftarrow{A}^{\mathsf{b}} \delta_{\mathsf{bc}} \overleftarrow{B}^{\mathsf{c}} = \overleftarrow{A}^{\mathsf{a}} \overleftarrow{B}^{\mathsf{a}} . \quad (105)$$

This allows us to express the continuum theory at $(\frac{8\pi}{\kappa})^2 C_{\mathsf{ab}} = f_{\mathsf{ab}} = \delta_{\mathsf{ab}}$ in terms of the KM currents,

$$\widehat{H}_{\mathbf{G}}^{\mathsf{sd}} \to \overleftarrow{L}^{\mathsf{a}} \overleftarrow{L}^{\mathsf{a}} + \left(\frac{\kappa}{8\pi}\right)^2 \overleftarrow{\partial \phi}^{\mathsf{a}} \overleftarrow{\partial \phi}^{\mathsf{a}} = \tfrac{1}{2}\left( \overrightarrow{J}_{KM}^{\mathsf{a}} \overrightarrow{J}_{KM}^{\mathsf{a}} + \overleftarrow{J}_{KM}^{\mathsf{a}} \overleftarrow{J}_{KM}^{\mathsf{a}} \right) . \quad (106)$$

This, in turn can be expressed in terms of the fermionic degrees of freedom using Eq. (104). One can also diagonalize the model by expressing the KM currents in terms of normal modes.[2]

Using Ref. [114], one can equivalently express the KM generators in terms of $\{\overleftarrow{\partial \phi}, \overrightarrow{\partial \phi}\}$ and non-Abelian generalizations of the nonlocal $\widehat{\theta}$-field from Eq. (82),

$$\overleftarrow{\theta}(\mathsf{x}) = \int^{\mathsf{x}} \mathrm{d}\mathsf{y} \, \overleftarrow{L}(\mathsf{y}) \qquad \text{and} \qquad \overrightarrow{\theta}(\mathsf{x}) = \int^{\mathsf{x}} \mathrm{d}\mathsf{y} \, \overrightarrow{L}(\mathsf{y}) . \quad (107)$$

Note that $\overrightarrow{\theta}$ is a continuum analogue of the disorder-like anyonic excitation from Eq. (56).

We conclude this section by conjecturing that there exists a combination of boundary conditions on the fields of the $\mathbf{\Gamma}$ WZNW model (106) that yields a degenerate ground state space, along with zero-mode operators that permute the different ground states. There are many combinations possible: one can pick various subgroups $\mathbf{G} \subset \mathbf{\Gamma}$ for either end of a wire of length L, and boundary conditions may be stated in terms of various field pairs ( [128], Part Two, Ch. 1, Sec. 5). For example, one can pin the $\mathbf{\Gamma}$-valued field $\widehat{U}(\mathsf{x})$ to a subgroup $\mathbf{G} \subset \mathbf{\Gamma}$ at $\mathsf{x} = 0$, while pinning the "dual" field $\overrightarrow{\theta}(\mathsf{x})$ to the quotient space $\mathbf{\Gamma}/\mathbf{G}$ on the other end $\mathsf{x} = \mathsf{L}$ of the

wire. In the abelian case $\Gamma = \mathbf{U}(1)$ and $\mathbf{G} = \mathbf{Z}_N$, one obtains the parafermionic zero modes at the interface between superconducting and ferromagnetic regions of the fermionized free boson [14–17]. When $\Gamma = \mathbf{SO}(3)$ and $\mathbf{G} = \mathbf{Z}_N$, the dual-field boundary condition corresponds to pinning the angular momentum $\ell$ to be zero modulo $N$, similar to the bulk ground-state constraint of the flux-ladder lattice model. We leave verification of this conjecture to future work.

# 7 Discussion & future directions

The 1D quantum Ising model can be interpreted in three different ways: as an ordinary spin chain, as a superconducting fermionic chain, or as a standalone effective edge theory for the surface code. This model and its continuum description thus lie at a key intersection of a wide variety of topics in the theory of topological order. Motivated by realizing more exotic topological order associated with non-Abelian finite groups $\mathbf{G}$ for the purpose of intrinsically protected quantum computation, we derive an analogue of the quantum Ising model for quantum doubles. We develop both a lattice effective theory for the quantum-double edge as well as its continuum description. While based on the mathematical framework of representation theory, our extensions nevertheless establish a starting point for potentially realizing exotic topological order in known physical systems. With our results described in detail in Sec. 2, here we focus on giving broader context and identifying future directions.

On the lattice front, we show that a slight generalization of a previously studied model called the *flux ladder* [71] describes the edge of the quantum double model, provided that the bulk has some fixed anyonic charge. To extract the flux ladder from the quantum-double edge, we recast the quantum-double Hamiltonian in terms of generalized "stabilizers" and manipulate said stabilizers in a similar fashion as is done in conventional stabilizer-based error correction. Such techniques may extend tools from stabilizer-based error correction to these and other exotic topological phases such as Hopf-algebra based quantum double models [129] and string-nets [130–132], which circumvent various stabilizer-based no-go theorems [133–135] and may offer a native universal set of protected gates.

On the continuum front, we provide a recipe for obtaining a field theory from a lattice model associated with $\mathbf{G}$ by embedding $\mathbf{G}$ into a Lie group and taking the continuum limit. Symmetries and parameters are naturally preserved during this embedding, and field theories describing low-energy excitations of previously known lattice models can be obtained from said models via this procedure. Applying this to the edge and bulk of quantum double models, we respectively obtain generalized versions of the 1D principal chiral model and 2D non-Abelian gauge theory. Since we consider general (and not necessarily gapped) edges, the parameter space of the obtained edge theory turns out to be quite large. We find a special point in the parameter space at which the edge field theory can be fermionized via non-Abelian bosonization and addition of a Wess-Zumino-Novikov-Witten (WZNW) term. This procedure provides a starting point for extending constructions of continuum parafermion modes in electronic systems to continuum versions of quantum-double defects [16].

Our work is necessarily technical due to the nuances of non-Abelian groups, calling for extensive derivation of statements that require little work in the $\mathbf{Z}_N$ case. As a result, there is much left to be done, and we outline some potentially interesting directions below.

**Stability of G-type zero modes**    While parafermionic ($\mathbf{G} = \mathbf{Z}_{N>2}$) strong zero modes are generically not robust against transverse-field perturbations, there exist fine-tuned points in the parameter space of the clock model at which strong zero modes are purported to exist [87,88]. At those special "integrable" points, the eigenspectrum of the longitudinal part of the

clock model happens to be invariant under a sign flip, $\widehat{H}_{\mathbf{Z}_N}^{\text{edge}} \to -\widehat{H}_{\mathbf{Z}_N}^{\text{edge}}$ [47]. We have identified a point in the parameter space of the dihedral ($\mathbf{G} = \mathbf{D}_3$) flux ladder with the same property (see Appx. B), and our preliminary numerical investigations hint at similar robustness of zero modes at such a point. It would be useful to perform a thorough numerical treatment and identify any connections of this and other special points to integrability.

**Continuum G-type defects** While we provide a plausible group-theoretic extension of the machinery that produces parafermionic (i.e., $\mathbf{Z}_N$-type) defects on edges of a quantum wire that models a fractional quantum-Hall medium, we do not construct the defects themselves. A path toward such a construction likely requires expressing the bosonized quantum wire in terms of normal modes[2], and imposing various boundary conditions on said modes. It may also be fruitful to express the quantum-wire fields in terms continuum non-Abelian order/disorder fields [see Eq. (107)] [114], akin to expressing parafermionic operators in terms of the well-known $\widehat{\theta}, \widehat{\phi}$ fields. On a similar note, we have shown that simplified versions of our ribbons behave similarly to the *lattice* order/disorder operators of the clock model [113–115]. Relating any discovered continuum defects to our flattened ribbons and order/disorder operators could be especially illuminating.

**Interpreting the WZNW term** Addition of this term, necessary for fermionization of the continuum edge theory, is allowed in the sense that it does not violate the bulk ground-state constraint imposed by the theory's global symmetry. Nevertheless, the term's relation to quantum-double edges and anyonic condensation remains unclear. One possible path toward clarity could be to investigate more general gapped edges constructed from projective representations [62]. This could yield connections to the WZNW term because its addition also yields a projective representation (of a group that is closely related to the Lie group defining the continuum theory [97]). Of course, the presence of projective representations in both contexts could also be coincidental.

**Symmetry-protected topological phases** We have identified a set of commuting-projector instances of the flux ladder that admit symmetry breaking of the parent group $\mathbf{G}$ into any subgroup $\mathbf{K}$ [see Eq. (33)], generalizing analogous constructions for the abelian case ($\mathbf{G} = \mathbf{Z}_N$) [110–112]. We note, however, that these may not be the only phases of the flux ladder, and that SPT order may also exist. Since projective representations are used to classify SPTs, such phases could exist in flux ladders constructed out of $X$-type operators that admit a projective representation of $\mathbf{K}$.

**Non-Abelian dualities** A category-theoretical duality known as "Morita equivalence" [10, 116] promises to generalize the well-known Kramers-Wannier duality of the clock model. This correspondence allows one to map between different Morita-equivalent gapped quantum-double edge instances of the flux ladder, such as the pure-charge and pure-flux edges comprising the model's "self-dual" instance (see Sec. 3.6). While there has been some work in formulating such duality transformations for lattice models [116], an explicit realization applicable to the flux ladder remains to be done. Doing so would provide a natural set of dualities for non-Abelian group models, which have so far proven difficult to construct [28, 53, 116, 118, 127, 136–142].

**Category-theoretic classifications [29, 62–65, 69, 70]** While we verify that the algebra of our flattened ribbons is indeed the same as that of ordinary ribbons [see Eq. (49)], elements of our ribbon basis can carry multiple edge excitations. A more complete treatment could explicitly construct the proper ribbon superpositions that carry single edge excitations, identify

their braiding relations, and investigate interplay with the gapped-edge instances of the flux ladder.

# Acknowledgments

We thank Daniel Arovas, Maissam Barkeshli, Barry Bradlyn, Michele Burrello, Aaron Chew, Iris Cong, Jan von Delft, Paul Fendley, Hrant Gharibyan, Andrey Gromov, Jonathan Gross, Bailey Gu, Alexander Jahn, Alexei Kitaev, Peter Kopietz, Gleb Kotoousov, Ashley Milsted, Olexei Motrunich, Sepehr Nezami, Kevin Slagle, Lev Spodyneiko, Michael Stone, Eugene Tang, Cenke Xu, Oleg Yevtushenko, Yi-Zhuang You, and Erez Zohar for valuable discussions. We gratefully acknowledge support from the Walter Burke Institute for Theoretical Physics at Caltech. The Institute for Quantum Information and Matter is an NSF Physics Frontiers Center. Contributions to this work by NIST, an agency of the US government, are not subject to US copyright. Any mention of commercial products does not indicate endorsement by NIST. V.V.A. thanks Olga Albert, Halina and Ryhor Kandratsenia, as well as Tatyana and Thomas Albert for providing daycare support throughout this work.

# Appendices

# A   Effect of bulk excitations on the edge theory

While we have focused on the quantum-double bulk being in a ground state when deriving our effective edge models, here we discuss how the edge lattice model is modified when nontrivial bulk excitations are present. We also conjecture how such excitations are transferred to the continuum.

## A.1   Charge excitations

For the qunit surface code, the bulk ground-state constraint projects the effective edge clock model (9) into the subspace where the star term $\widehat{X}_h^{(\text{global})}$ (12) is identity. More general bulk excitations correspond to projecting the edge model into a subspace where the star is a root of unity $e^{i\frac{2\pi}{N}\lambda}$, with $\lambda \in \{0, 1, \cdots, N-1\}$. The projections defining such subspaces are

$$\widehat{P}_\lambda = \frac{1}{N} \sum_{h \in \mathbf{Z}_N} e^{-i\frac{2\pi}{N}\lambda h} \widehat{X}_h^{(\text{global})} \,. \tag{108}$$

A pair of bulk charge excitations for quantum doubles is labeled by irreps $\lambda$ of $\mathbf{G}$ and is created by the operator $\text{TR}[\widehat{\mathcal{Z}}_\lambda]$ acting on a bulk site [2,143] (akin to Pauli $\widehat{Z}$ matrices creating charge excitations for the surface code). Charge excitations correspond to projecting the flux ladder (27) onto the subspace corresponding to the irrep $\lambda$ present in the representation of $\mathbf{G}$ formed by $\overrightarrow{X}_{h \in \mathbf{G}}^{(\text{global})}$ (31). The corresponding projection is

$$\overrightarrow{P}_\lambda = \frac{d_\lambda}{|\mathbf{G}|} \sum_{h \in \mathbf{G}} \text{TR}\left[\mathcal{Z}_\lambda\left(h^{-1}\right)\right] \overrightarrow{X}_h^{(\text{global})} \,. \tag{109}$$

For the flux ladder, excitations are carried by the order-parameter operator $\overrightarrow{\Gamma}_{1,\lambda} = \widehat{\mathcal{Z}}_\lambda$ (38).

## A.2 Flux excitations

Flux excitations correspond to plaquette violations, where $\widehat{Z}^{\square} \equiv \widehat{Z}^{\,\square}\, \widehat{Z}^{\,\square}\, \widehat{Z}^{\dagger\,\square}\, \widehat{Z}^{\dagger\,\square} = e^{i\frac{2\pi}{N}h} \neq 1$ with $h \in \{0, 1, \cdots, N-1\}$. In our bicycle-wheel model, this carries through to a violation of a bulk plaquette located on a section of tire,

$$\widehat{Z}_\lambda^{\triangleleft} \equiv \widehat{Z}_\lambda^{\blacktriangleleft}\, \widehat{Z}_\lambda^{\triangleleft}\, \widehat{Z}_\lambda^{\dagger\,\blacktriangleleft} = e^{i\frac{2\pi}{N}h}. \tag{110}$$

To obtain the effective edge clock model as in Sec. 3, we project onto the subspace where bulk plaquettes are one on all spokes except the above one. At this spoke, the formula (8) for obtaining the edge term in the clock model generalizes to

$$\widehat{Z}^{\triangleleft} = (\widehat{Z}^\dagger \widehat{Z})^{\blacktriangleleft}\, \widehat{Z}^{\triangleleft}\, (\widehat{Z}^\dagger \widehat{Z})^{\blacktriangleleft} = \widehat{Z}^{\dagger\,\blacktriangleleft}\, \widehat{Z}^{\triangleleft}\, \widehat{Z}^{\blacktriangleleft} = \widehat{Z}^{\dagger\,\blacktriangleleft}\, e^{i\frac{2\pi}{N}h}\widehat{Z}^{\blacktriangleleft} = e^{i\frac{2\pi}{N}h}\widehat{Z}^{\dagger(\mathsf{x}-1)}\widehat{Z}^{(\mathsf{x})}. \tag{111}$$

The extra phase is a *topological boundary condition* on the chain [112], and our derivation relates such conditions to the presence of bulk flux excitations.

In the continuum limit, the above condition can be rephrased as a condition on the field $\widehat{\phi}(\mathsf{x})$. Let us parameterize the continuum edge with spatial index $\mathsf{x} \in [0, \mathsf{L})$, and place the violated spoke such that $(\mathsf{x}-1, \mathsf{x}) \to (\mathsf{L}, 0)$ in the limit. The above constraint is then equivalent to $\widehat{\phi}(\mathsf{L}) = \widehat{\phi}(0) + h$.

The generalization to quantum doubles, where flux excitations are labeled by conjugacy classes of **G**, is straightforward. With $h \in \mathrm{cls}(h)$ a representative of the conjugacy class of $h$, a local bulk plaquette constraint (24) on a section of tire generalizes to

$$\mathrm{TR}\big[\widehat{\mathcal{Z}}_\lambda^{\triangleleft}\big] = \mathrm{TR}\big[\widehat{\mathcal{Z}}_\lambda^{\blacktriangleleft} . \widehat{\mathcal{Z}}_\lambda^{\triangleleft} . \widehat{\mathcal{Z}}_\lambda^{\dagger\,\blacktriangleleft}\big] = \mathrm{TR}\big[\mathcal{Z}_\lambda(h)\big]. \tag{112}$$

This constraint, when plugged into Eq. (26), yields the modified two-body term at the relevant site,

$$\widehat{\mathcal{Z}}^{\dagger(\mathsf{x}-1)}.\widehat{\mathcal{Z}}^{(\mathsf{x})} \to \widehat{\mathcal{Z}}^{\dagger(\mathsf{x}-1)}.\mathcal{Z}(h).\widehat{\mathcal{Z}}^{(\mathsf{x})}. \tag{113}$$

This modification can also be done by conjugating the Hamiltonian with the "disorder" operator $\overrightarrow{\Gamma}_{h,\mathbf{1}}^{(2\mathsf{x})}$ (38). We anticipate that, in the continuum limit of the corresponding principal chiral model (78), the above constraint is related to the *monodromy condition* [81] on the **SO**(3)-valued field $\widehat{U}(\mathsf{x})$, i.e., $\widehat{U}(\mathsf{x} = \mathsf{L}) = U(h).\widehat{U}(\mathsf{x} = 0)$.

## A.3 Combinations

General excitations in quantum doubles are labeled by $\{\Lambda, \mathrm{cls}(h)\}$, where $\Lambda \in \mathrm{irr}(\mathbf{C}(h))$, and $\mathbf{C}(h)$ is the *centralizer* — the subgroup of all elements that commute with $h$. Such a general bulk excitation translates simultaneously to a global subspace constraint and a twisted boundary condition in our effective edge model (27). The edge is projected onto the global subspace with projection

$$\overrightarrow{P}_\Lambda = \frac{1}{|\mathbf{C}(h)|} \sum_{b \in \mathbf{C}(h)} \chi_\Lambda(b)\, \overrightarrow{X}_b^{(\text{global})}, \tag{114}$$

where $|\mathbf{C}(h)|$ is the order of $\mathbf{C}(h)$, and $\chi_\Lambda$ is the character of $h$ in the irrep $\Lambda$ of the centralizer. The twisted boundary condition yields a two-body term that is modified by the representative element a la Eq. (113).

## B    Dihedral flux ladder

### B.1    Explicit construction of dihedral operators

Each element $h \in \mathbf{D}_N$ can be expressed as $h = r^n p^{\frac{1}{2}(1-\mu)}$, where $n \in \mathbf{Z}_N$, and $\mu \in \pm 1$ denotes either the absence or presence of the reflection. With $(n, \mu)$ uniquely determining each element, each site $\times$ can be interpreted as a tensor product of a qu$N$it $\{|n\rangle, n \in \mathbf{Z}_N\}$ and a qubit $\{|\pm\rangle\}$. Following Eqs. (13) and dihedral multiplication rules, the left- and right-multipliers act on this space as

$$
\begin{array}{ll}
\overrightarrow{X}_r |n, \mu\rangle = |n+1, \mu\rangle & \overleftarrow{X}_r |n, \mu\rangle = |n-\mu, \mu\rangle \\
\overrightarrow{X}_p |n, \mu\rangle = |-n, -\mu\rangle & \overleftarrow{X}_p |n, \mu\rangle = |n, -\mu\rangle
\end{array} \qquad (115)
$$

We can construct these explicitly using the qunit Pauli operator $\widehat{X}$, the qunit "reflection" operator $\widehat{S}$ (with $\widehat{S}\widehat{X}\widehat{S} = \widehat{X}^\dagger$ and $\widehat{S}\widehat{Z}\widehat{S} = \widehat{Z}^\dagger$), as well as the associated qubit operators $\widehat{\sigma}_x, \widehat{\sigma}_z$, and $\widehat{P}_\pm = |\pm\rangle\langle\pm|$ (cf. [71, 144, 145]):

$$
\begin{array}{ll}
\overrightarrow{X}_r = \widehat{X} \otimes 1 & \overleftarrow{X}_r = \widehat{X}^\dagger \otimes \widehat{P}_+ + \widehat{X} \otimes \widehat{P}_- \\
\overrightarrow{X}_p = \widehat{S} \otimes \widehat{\sigma}_x & \overleftarrow{X}_p = 1 \otimes \widehat{\sigma}_x
\end{array} \qquad (116)
$$

The diagonal operators can be constructed out of the qunit/qubit operators $\widehat{Z}, \widehat{\sigma}_z, \widehat{P}_\pm$:

$$
\begin{array}{lll}
\widehat{Z}_{\mathbf{1}} = 1 & \widehat{Z}_{\mathbf{1}^+} = \widehat{Z}^{N/2} \otimes 1 & \\
\widehat{Z}_{\mathbf{1}'} = 1 \otimes \widehat{\sigma}_z & \widehat{Z}_{\mathbf{1}^-} = \widehat{Z}^{N/2} \otimes \widehat{\sigma}_z & \widehat{Z}_{\mathbf{2}_n} = \begin{bmatrix} \widehat{Z}^n \otimes \widehat{P}_+ & \widehat{Z}^n \otimes \widehat{P}_- \\ \widehat{Z}^{n\dagger} \otimes \widehat{P}_- & \widehat{Z}^{n\dagger} \otimes \widehat{P}_+ \end{bmatrix}.
\end{array} \qquad (117)
$$

These satisfy the relation (14) when acting on the group states $|h\rangle \leftrightarrow |n, \mu\rangle$.

     Flux-ladder $ZZ$-terms corresponding to 1D-irreps $\lambda$ are simple products of qunit/qubit operators; e.g., $\widehat{Z}_{\mathbf{1}'}^{\dagger(\times)}.\widehat{Z}_{\mathbf{1}'}^{(\times+1)} = \widehat{\sigma}_z^{(\times)}\widehat{\sigma}_z^{(\times+1)}$ is an Ising interaction. The 2D-irrep term is

$$
\begin{aligned}
&\mathrm{T_R}[C_{\mathbf{2}_n}.\widehat{Z}_{\mathbf{2}_n}^{\dagger(\times)}.\widehat{Z}_{\mathbf{2}_n}^{(\times+1)}] = \\
&C_{\mathbf{2}_n}^{11}\left(\widehat{Z}_n^{\dagger(\times)}\widehat{Z}_n^{(\times+1)} \otimes \widehat{P}_+^{(\times)}\widehat{P}_+^{(\times+1)} + \widehat{Z}_n^{(\times)}\widehat{Z}_n^{\dagger(\times+1)} \otimes \widehat{P}_-^{(\times)}\widehat{P}_-^{(\times+1)}\right) + \text{h.c.} \\
&+ C_{\mathbf{2}_n}^{12}\left(\widehat{Z}_n^{\dagger(\times)}\widehat{Z}_n^{(\times+1)} \otimes \widehat{P}_-^{(\times)}\widehat{P}_+^{(\times+1)} + \widehat{Z}_n^{(\times)}\widehat{Z}_n^{\dagger(\times+1)} \otimes \widehat{P}_+^{(\times)}\widehat{P}_-^{(\times+1)}\right) + \text{h.c.},
\end{aligned} \qquad (118)
$$

where we have moved $n$ downstairs for compactness. The one-body $X$-type term is

$$
\sum_{n \in \mathbf{Z}_N} \mathsf{f}_{r^n}\left(\widehat{X}_n^{\dagger(\times)} \otimes \widehat{P}_+^{(\times)} + \widehat{X}_n^{(\times)} \otimes \widehat{P}_-^{(\times)}\right) + \text{h.c.} + \mathsf{f}_{r^n p}\left(\widehat{X}_n^{\dagger(\times)} \otimes \widehat{\sigma}_+^{(\times)} + \widehat{X}_n^{(\times)} \otimes \widehat{\sigma}_-^{(\times)}\right). \qquad (119)
$$

### B.2    Notable cases of the flux ladder

Let us briefly go through the different specific cases of the model (27) from Table 6, with exception of the *q-double* case covered in Sec. 3.5. In order to contrast with the clock model (9), we identify which strings of permutation matrices leave $\widehat{H}_{\mathbf{D}_N}^{\text{edge}}$ invariant in each case. For *general* coefficients, we believe the only such symmetry is the $\mathbf{D}_N$ symmetry already mentioned. This was verified numerically for $\mathbf{D}_3$ and $\mathbf{D}_4$ by randomly sampling coefficients C, f and explicitly checking commutation for all $(2N)!$ strings. The same holds when C is *Hermitian* and f is real.

     The *chiral* case [71] assumes a unitary C; given the general conditions, this happens when either $C_\lambda^{11} = 0$ or $C_\lambda^{12} = 0$. For the latter case, the permutation symmetry is the same as before.

| Case | Conditions | Symmetry | Remarks |
|---|---|---|---|
| General | $C_\lambda^{21} = C_\lambda^{12\star}, C_\lambda^{22} = C_\lambda^{11\star}, f_{h^{-1}} = f_h^\star$ | $\mathbf{D}_N$ | $\{\overrightarrow{X}_h^{(\text{global})}\}$ |
| Hermitian | $C_\lambda^{22} = C_\lambda^{11}, f_{h^{-1}} = f_h$ | " | " |
| Chiral [71] | $C_\lambda$ unitary, $f_h = f_{\text{cls}(h)}$ | " | " |
| Q-double/SB [62] | $C_\lambda = \frac{d_\lambda}{2N/|\mathbf{K}|}\Pi_{\mathbf{1}}^{\mathbf{K}}, f_h = \frac{1}{|\mathbf{K}|}\delta_{h\in\mathbf{K}}$ | $\mathbf{D}_N \times \mathbf{N}(\mathbf{K})$ | $\mathbf{K} \subset \mathbf{D}_N$ |
| Gauge theory [86] | $C_\lambda \propto 1_\lambda, f_h = f_{\text{cls}(h)}$ | $\mathbf{D}_N^{\times 2} \rtimes \mathbf{Z}_2$ | $\{\overleftarrow{X}_h^{(\text{global})}\}, |h\rangle \to |h^{-1}\rangle$ |
| Self-dual [28] | $C_\lambda = \frac{d_\lambda}{2N}1_\lambda, f_h = \frac{1}{2N}$ | $\mathbf{S}_{2N}$ | all permutations |
| Integrable? | $C_{\mathbf{2}_1}^{12} = C_{\mathbf{2}_1}^{11\star}e^{i\frac{\pi}{3}}$ | $\mathbf{D}_3$ | $N = 3$ only |

Table 6: Notable cases of the flux ladder (27) for $\mathbf{G} = \mathbf{D}_N$.

In the *Gauge theory* case, the symmetry enlarges further to $\mathbf{D}_N \times \mathbf{D}_N$, represented by products of $\overrightarrow{X}^{(\text{global})}$ and $\overleftarrow{X}^{(\text{global})}$. The two-body term is $\text{Tr}[\widehat{\mathcal{Z}}_\lambda^{(x)\dagger}.\widehat{\mathcal{Z}}_\lambda^{(x+1)}]$ (akin to the the two-body term in the principal chiral model), while $f_h$ depends only on the conjugacy class of $h$. An additional "inversion" symmetry $\widehat{\mathcal{Z}} \to \widehat{\mathcal{Z}}^\dagger$ and $\overrightarrow{X} \to \overleftarrow{X}$ is also present. This case is a finite-group version of Kogut-Susskind gauge theory on an exotic Hawaiian earring geometry [86].

The dihedral structure of the local Hilbert space is not necessary to describe the *Self-dual* case, which is equivalent to the self-dual point of the $\mathbf{Z}_{2N}$ clock model (see Sec. 3.6). We suspect the same is true for the *Gauge theory* case, at least for $\mathbf{D}_3$, as we have numerically verified that the two-body term can be expressed in terms of $\mathbf{Z}_6$-clock-model $Z$ operators (generated by $\widehat{Z} \otimes \widehat{\sigma}_z$ on each site).

In the *Integrable?* case for $\mathbf{D}_{N=3}$, the spectrum of the two-body term is symmetric around a particular value (which can be shifted to zero by adding a constant). For the clock model, such a special point is relevant in studying the stability of parafermionic zero modes [87,88], and signals that there may be integrable points [47]. We leave this to future work.

# C   Explicit $\mathbf{D}_N \hookrightarrow \mathbf{SO}(3)$ embedding procedure

Recall that the dihedral group $\mathbf{D}_N = \mathbf{Z}_N \rtimes \mathbf{Z}_2$, where $\mathbf{Z}_N = \langle r \rangle$ is the rotation subgroup, and $\mathbf{Z}_2 = \langle p \rangle$ is the reflection subgroup. In order to take continuum limits, we would like to embed this group into a larger group space such that any pair of elements $h, k \in \mathbf{D}_N$ can be connected to each other via a continuous path. A space satisfying this criteria is $\mathbf{SO}(3)$, the group of 3D proper rotations.

## C.1   Dihedral to $\mathbf{SO}(3)$ flux ladder

The structure of the $\mathbf{SO}(3)$ group-space operators is much like that of $\mathbf{D}_N$: the space consists of element-labeled basis vectors, $\{|R\rangle, R \in \mathbf{SO}(3)\}$, while left $(\overrightarrow{X}_R)$ and right $(\overleftarrow{X}_R)$ rotations act by permuting basis elements according to the group multiplication rules (13). We can directly embed our left and right dihedral multipliers for $h \in \mathbf{D}_N$ as $\overrightarrow{X}_h \to \overrightarrow{X}_{R=h}$ and $\overleftarrow{X}_h \to \overleftarrow{X}_{R=h}$, where in the latter formulas $h \in \mathbf{D}_N$ is treated as an element of a dihedral subgroup of $\mathbf{SO}(3)$. The notation for $X$-operators is thus the same for both groups; we will make it clear from context which group we mean. We also pick a particular dihedral subgroup for ease of presentation; its rotation and reflection generators are, respectively,

$$r = \tfrac{2\pi}{N}\text{-rotation around z-axis}; \qquad p = \pi\text{-rotation around y-axis}. \qquad (120)$$

As with the dihedral group, diagonal operators on $\mathbf{SO}(3)$ consist of irrep-matrix elements

of the group. The irreps are indexed by the total angular momentum $\ell \geq 0$, and each irrep matrix $U_\ell$ is of dimension $2\ell + 1$. The corresponding operators $\widehat{U}_\ell$ satisfy the $\mathbf{SO}(3)$ analogue of Eq. (14),

$$\widehat{U}_\ell = \int_{\mathbf{SO}(3)} \mathrm{d}R \, U_\ell(R) \, |R\rangle \langle R| \, , \qquad (121)$$

where $\mathrm{d}R$ is the Haar measure satisfying $\int_{\mathbf{SO}(3)} \mathrm{d}R = 8\pi^2$. Since $U_{\ell>0}$ is a matrix, the above acts on both the internal irrep space and the $\mathbf{SO}(3)$ site space [see Eq. (14)].

Our embedding of dihedral operators $\widehat{Z}$ into $\mathbf{SO}(3)$ operators $\widehat{U}$ relies on knowledge of how the $\mathbf{SO}(3)$ irreps decompose or *branch* into $\mathbf{D}_N$ irreps. The defining irrep branches as $\ell = 1 \downarrow \mathbf{1}' \oplus \mathbf{2}_1$, yielding the sign and defining two-dimensional irreps of $\mathbf{D}_N$. In terms of operators, this means that $\widehat{U}_{\ell=1}$, when acting on basis vectors in $\mathbf{D}_N$, branches into a block matrix of the $\mathbf{1}'$- and $\mathbf{2}_1$-irrep dihedral $Z$-operators,

$$\widehat{U}_1 |R\rangle = [\widehat{Z}_{\mathbf{1}'} \oplus \widehat{Z}_{\mathbf{2}_1}] |R\rangle \qquad \text{for } R \in \mathbf{D}_N \, . \qquad (122)$$

To showcase the branching property, let us use the standard angular momentum basis [Ref. [122], Eq. (3.24)] for $U_\ell$ and recall our choice of dihedral group (120). Then,

$$U_1(r) = \begin{bmatrix} e^{i\frac{2\pi}{N}} & 0 & 0 \\ 0 & 1 & 0 \\ 0 & 0 & e^{-i\frac{2\pi}{N}} \end{bmatrix} \quad \text{and} \quad U_1(p) = \begin{bmatrix} 0 & 0 & 1 \\ 0 & -1 & 0 \\ 1 & 0 & 0 \end{bmatrix} . \qquad (123)$$

The middle entry corresponds to the $\mathbf{1}'$ irrep of $\mathbf{D}_N$, where $r \to 1$ and $p \to -1$, while the two-by-two matrix made up of the four corners is precisely the $\mathbf{2}_1$ irrep (16). In the chosen basis, $U_1$ is only a permutation away from the two-block decomposition in Eq. (122).

More generally, $U_\ell(r)$ for $\ell \leq \lfloor N/2 \rfloor$ has diagonal entries $e^{\pm i\frac{2\pi}{N}\ell}$ in its corners, while $U_\ell(p)$ has contains 1's in its upper-right and lower-left corners. These four corners thus form the $\lambda = \mathbf{2}_\ell$ irrep of $\mathbf{D}_N$. On the operator level, this means that $\widehat{Z}_{\mathbf{2}_n}$ descends from $\widehat{U}_{\ell=n}$ for all $n$. For $N$ even, the extra 1D irreps $\mathbf{1}^\pm$ can be similarly obtained.

Embedding the general $\mathbf{D}_N$ flux ladder yields an instance of the $\mathbf{SO}(3)$ flux ladder,

$$\widehat{H}_{\mathbf{D}_N}^{\mathrm{edge}} \to -\sum_{\mathsf{x}} \sum_{\lambda \in \mathrm{irr}(\mathbf{D}_N)} \mathrm{Tr}\big[ \mathsf{C}_{\ell_\lambda}^{\mathrm{edge}} . \widehat{U}_{\ell_\lambda}^{\dagger(\mathsf{x})} . \widehat{U}_{\ell_\lambda}^{(\mathsf{x}+1)} \big] + \sum_{h \in \mathbf{D}_N} \mathsf{f}_h^{\mathrm{edge}} \overleftarrow{X}_{R=h}^{(\mathsf{x})} . \qquad (124)$$

All $\overrightarrow{X}_{R=h}$ and $\overleftarrow{X}_{R=h}$ are now rotations acting on an $\mathbf{SO}(3)$ space at each site $\mathsf{x}$, where the dihedral elements $h$ are represented as orthogonal $\mathbf{SO}(3)$ matrices $R$ following Eq. (120), and $\mathsf{f}_h^{\mathrm{edge}} = \mathsf{f}_h$. Each $\mathbf{D}_N$ irrep $\lambda$ in the two-body term has a corresponding $\mathbf{SO}(3)$ irrep $\ell_\lambda$ which branches into $\lambda$ when restricted to $\mathbf{D}_N$. For example, we have seen above that the $\ell = 1$ irrep houses two dihedral irreps ($\ell_{\mathbf{1}'} = \ell_{\mathbf{2}_1} = 1$). The corresponding $\mathbf{SO}(3)$ parameter matrix for $\ell = 1$, written in the standard angular momentum basis, is then

$$\mathsf{C}_{\ell_{\mathbf{1}'}=1}^{\mathrm{edge}} + \mathsf{C}_{\ell_{\mathbf{2}}=1}^{\mathrm{edge}} = \begin{bmatrix} [\mathsf{C}_{\lambda=\mathbf{2}_1}]_{11} & 0 & [\mathsf{C}_{\lambda=\mathbf{2}_1}]_{12} \\ 0 & \mathsf{C}_{\lambda=\mathbf{1}'} & 0 \\ [\mathsf{C}_{\lambda=\mathbf{2}_1}]_{21} & 0 & [\mathsf{C}_{\lambda=\mathbf{2}_1}]_{22} \end{bmatrix} , \qquad (125)$$

where $[\mathsf{C}_{\mathbf{2}_1}]_{mn}$ are elements of the parameter matrix $\mathsf{C}_{\lambda=\mathbf{2}_1}$ of $\widehat{H}_{\mathbf{D}_N}^{\mathrm{edge}}$ (27). The remaining $\mathsf{C}_\lambda$ can be embedded similarly into angular momenta $\ell_\lambda \leq N$.

In order to make sure the $\mathbf{SO}(3)$ model (124) remembers its dihedral origins, we need to provide a potential $\widehat{V}_{\mathbf{D}_N}^{\mathrm{smooth}}$ that pins each site onto the dihedral subspace. This potential can be obtained by projecting $\widehat{U}_{\ell=N}$ onto the trivial dihedral irrep for which $\mathsf{z}$-axis rotations by $\phi$ are represented by $e^{-i\phi N}$,

$$\widehat{V}_{\mathbf{D}_N}^{\mathrm{smooth}} = \mathrm{Tr}\big[ \Pi_{\mathbf{1}}^{\mathbf{D}_N} . \widehat{U}_N \big] = \int_{\mathbf{SO}(3)} \mathrm{d}R \, V_{\mathbf{D}_N}^{\mathrm{smooth}}(R) \, |R\rangle \langle R| \, . \qquad (126)$$

To show that $V_{\mathbf{D}_N}^{\text{smooth}}(R)$ is maximized only at the $2N$ coordinates of our embedded dihedral group, we parametrize $R$ using Euler angles $(\alpha, \beta, \gamma)$ in the $ZYZ$ convention, with $\alpha, \gamma \in [0, 2\pi)$ and $\beta \in [0, \pi]$. The potential function is

$$V_{\mathbf{D}_N}^{\text{smooth}}(\alpha, \beta, \gamma) = \cos\left[N(\alpha + \gamma)\right] \cos^{2N} \frac{\beta}{2} + \cos\left[N(\alpha - \gamma)\right] \sin^{2N} \frac{\beta}{2}. \tag{127}$$

This "stabilizer" [49] is maximized at $\beta \in \{0, \pi\}$ and $\alpha \pm \gamma$ being a multiple of $2\pi/N$ — precisely the $2N$ dihedral coordinates from Eq. (120).

## C.2 $\mathbf{SO}(3)$ flux ladder to principal chiral model

The key feature of $\mathbf{SO}(3)$ that yields sensible continuum limits is that all rotations can be written as exponentials of elements of the angular momentum Lie algebra. For the $X$-operators,

$$\overrightarrow{X}_R = e^{-i\phi_R^{\mathtt{a}} \overrightarrow{L}^{\mathtt{a}}} \qquad \text{and} \qquad \overleftarrow{X}_R = e^{-i\phi_R^{\mathtt{a}} \overleftarrow{L}^{\mathtt{a}}}, \tag{128}$$

where $\phi_R^{\mathtt{a}}$ with $\mathtt{a} \in \{\mathtt{x}, \mathtt{y}, \mathtt{z}\}$ are the *axis-angle coordinates* of $R$ (all repeated `typewriter`-font indices are summed). The direction of the vector $\phi_R$ is the axis around which the rotation matrix $R$ rotates, while the norm $\phi_R^{\mathtt{a}} \phi_R^{\mathtt{a}} \leq \pi$ is the angle of rotation. For our chosen dihedral embedding (120), $\phi_{R=r} = (0, 0, \frac{2\pi}{N})$ and $\phi_{R=p} = (0, \pi, 0)$. Commuting with each other, these respective triples obey the standard [122] commutation relations amongst themselves: $[\overrightarrow{L}^{\mathtt{a}}, \overrightarrow{L}^{\mathtt{b}}] = i\epsilon_{\mathtt{abc}} \overrightarrow{L}^{\mathtt{c}}$, $[\overleftarrow{L}^{\mathtt{a}}, \overleftarrow{L}^{\mathtt{b}}] = i\epsilon_{\mathtt{abc}} \overleftarrow{L}^{\mathtt{c}}$, and $[\overrightarrow{L}^{\mathtt{a}}, \overleftarrow{L}^{\mathtt{b}}] = 0$ (with $\epsilon$ the anti-symmetric tensor).

The $\mathbf{SO}(3)$ irrep matrices $U_\ell$ can also be written in terms axis-angle coordinates $\phi^{\mathtt{a}}$ and their own $2\ell + 1$-dimensional generators $L_\ell^{\mathtt{a}}$,

$$U_\ell(R) = e^{-i\phi_R^{\mathtt{a}} L_\ell^{\mathtt{a}}}, \qquad \text{where} \qquad [L_\ell^{\mathtt{a}}, L_\ell^{\mathtt{b}}] = i\epsilon_{\mathtt{abc}} L_\ell^{\mathtt{c}} \qquad \text{and} \qquad \text{Tr}[L_\ell^{\mathtt{a}}.L_\ell^{\mathtt{b}}] = \delta_{\mathtt{ab}} c_\ell. \tag{129}$$

We continue to use the standard angular momentum basis for these generators, with the normalization constant $c_\ell = \frac{1}{3}\ell(\ell + 1)(2\ell + 1)$.

Performing all expansions, assuming that $\mathsf{C}^{\text{edge}}$ are Hermitian and $\mathsf{f}^{\text{edge}}$ are real, and adding a pinning term yield a generalized $\mathbf{SO}(3)$ principal chiral model

$$\widehat{H}_{\mathbf{D}_N}^{\text{edge}} \to \int d\mathsf{x} \, \mathsf{C}_{\mathtt{ab}} \overleftarrow{\partial \phi}^{\mathtt{a}}(\mathsf{x}) \overleftarrow{\partial \phi}^{\mathtt{b}}(\mathsf{x}) + \mathsf{f}_{\mathtt{ab}} \overleftarrow{L}^{\mathtt{a}}(\mathsf{x}) \overleftarrow{L}^{\mathtt{b}}(\mathsf{x}) - \mathsf{M} \text{Tr}\left[\Pi_{\mathbf{1}}^{\mathbf{D}_N}.\widehat{U}_N(\mathsf{x})\right]. \tag{130}$$

The constituent fields are the $\mathbf{SO}(3)$-valued position $\widehat{U}$ (121), position current $\overleftarrow{\partial \phi} \propto \widehat{U}^\dagger \partial \widehat{U}$ (77), and "right" momentum current $\overleftarrow{L}$ (128). The last term (126) pins the $\mathbf{SO}(3)$ space to $\mathbf{D}_N$ in the $\mathsf{M} \to \infty$ limit. The parameters $\{\mathsf{C}_\ell^{\text{edge}}, \mathsf{f}_h^{\text{edge}}\}$ of the flux ladder (124) are absorbed into matrices $\mathsf{C}_{\mathtt{ab}}$ and $\mathsf{f}_{\mathtt{ab}}$, shown explicitly for this *General-edge* case in Table 4.

In generalizing $\mathbf{SO}(3)$ to other $\mathbf{\Gamma}$, the $\epsilon_{\mathtt{abc}}$ in the defining (75) commutation relations should be replaced by the structure constants $f_{\mathtt{abc}}$ defining $\mathbf{\Gamma}$'s Lie algebra. Expanding a uniform sum of $\overleftarrow{X}$'s in the principal chiral model (124) is generally ambiguous because the coordinates $\phi_g^{\mathtt{a}}$ specifying elements $g \in \mathbf{G}$ inside the manifold $\mathbf{\Gamma}$ are not unique. A unique expansion can be obtained by averaging the $X$-term over $\mathbf{G}$ before expansion,

$$\sum_{g \in \mathbf{G}} e^{-i\phi_g^{\mathtt{a}} \overleftarrow{L}^{\mathtt{a}}} = \frac{1}{2} \sum_{g \in \mathbf{G}} e^{-i\phi_g^{\mathtt{a}} \overleftarrow{L}^{\mathtt{a}}} + \frac{1}{2} \sum_{g \in \mathbf{G}} e^{-i\phi_{hgh^{-1}}^{\mathtt{a}} \overleftarrow{L}^{\mathtt{a}}} = \frac{1}{|\mathbf{G}|} \sum_{g,h \in \mathbf{G}} e^{-i(R_h \phi_g)^{\mathtt{a}} \overleftarrow{L}^{\mathtt{a}}} \to \mathsf{f}_{\mathtt{ab}} \overleftarrow{L}^{\mathtt{a}} \overleftarrow{L}^{\mathtt{b}}, \tag{131}$$

where $\overleftarrow{L}$ are the generators of $\mathbf{\Gamma}$'s Lie algebra, we have used the adjoint representation (81) in writing $\phi_{hgh^{-1}}^{\mathtt{a}} = R_h^{\mathtt{ab}} \phi_g^{\mathtt{b}} = (R_h \phi_g)^{\mathtt{a}}$, and

$$\mathsf{f}_{\mathtt{ab}} = \frac{1}{2|\mathbf{G}|} \sum_{g,h \in \mathbf{G}} (R_h \phi_g)^{\mathtt{a}} (R_h \phi_g)^{\mathtt{b}}. \tag{132}$$

This averaging twirls the sum over the possible coordinate realizations, guaranteeing that $f_{ab} \overleftarrow{L}^a \overleftarrow{L}^b$ is invariant under $\overleftarrow{X}_a^{(\text{global})}$ for any $a \in \mathbf{G}$. This was not necessary for $\mathbf{\Gamma} = \mathbf{SO}(3)$ due to the uniqueness of the angle-axis representation when $\phi_g^a \phi_g^a$ is minimized.

# D  Continuum description of the quantum-double bulk

We apply the embedding and pinning procedure from Sec. 5 to the surface-code (3) and quantum-double (18) bulk Hamiltonians. We recover, respectively, an abelian gauge theory and a non-Abelian Yang-Mills gauge theory [52, 82, 100, 101]. Both are known to contain non-Abelian defects analogous to the anyons of the respective lattice models [102–106].

## D.1  Surface code to abelian gauge theory

Here, we apply the procedure from Sec. 5 to convert the qunit surface code (3) to a $\mathbf{Z}_N$-pinned abelian gauge field theory. We first perform the well-known [52,82] relabeling required for the 2D continuum limit. This relabeling produces a pair of fields, each corresponding to one of the two dimensions of the lattice. Operators on the top and bottom of a plaquette are identified as the field $\widehat{\phi}_1$, evaluated at two points that are shifted by the unit vector $2_x$, $\widehat{\phi}^{\square} \to \widehat{\phi}_1(x)$ and $\widehat{\phi}^{\square} \to \widehat{\phi}_1(x + 2_x)$. Similarly, the left and right sides of a plaquette are identified with the other field $\widehat{\phi}_2$, yielding $\widehat{\phi}^{\square} \to \widehat{\phi}_2(x)$ and $\widehat{\phi}^{\square} \to \widehat{\phi}_2(x + 1_x)$. A similar identification is done for the star terms.

After the above relabeling, the continuum limit from lattice gauge theory proceeds as

$$\widehat{\phi}^{\square} + \widehat{\phi}^{\square} - \widehat{\phi}^{\square} - \widehat{\phi}^{\square} \to \widehat{F}(x) = \partial_1 \widehat{\phi}_2(x) - \partial_2 \widehat{\phi}_1(x) \tag{133a}$$

$$\widehat{L}^{\dagger} + \widehat{L}^{\dagger} - \widehat{L}^{\dagger} - \widehat{L}^{\dagger} \to \widehat{G}(x) = \partial_1 \widehat{L}_1(x) + \partial_2 \widehat{L}_2(x) . \tag{133b}$$

The $\widehat{F}$ operator is known as the curvature of gauge potential $\widehat{\phi}$, and $\widehat{G}$ is a generator of local gauge transformations. Assuming all fields vary smoothly and slowly and expanding both plaquettes and stars to second order yields the gauge field theory

$$\widehat{H}_{\mathbf{Z}_N}^{\text{bulk}} \to \int d^2x \; \mathsf{C}\,\widehat{F}(x)\widehat{F}(x) + \mathsf{f}\,\widehat{G}(x)\widehat{G}(x) - \mathsf{M}\widehat{V}_{\mathbf{Z}_N}^{\text{smooth}}(x) , \tag{134}$$

with parameters $\mathsf{C} = \frac{1}{2}\sum_{\lambda=1}^{N-1}\lambda^2$ and $\mathsf{f} = \frac{1}{2}\sum_{h=1}^{N-1}\left(\frac{2\pi}{N}h\right)^2$, and with $\widehat{V}_{\mathbf{Z}_N}^{\text{smooth}}(x)$ consisting of one cosine for each $\widehat{\phi}_1$ and $\widehat{\phi}_2$. This 2D bulk field theory is similar to the 1D edge field theory (62) in that it is quadratic in two "dual" fields. However, there are also key differences in the field algebras: (I) the 1D fields $\{\partial\widehat{\phi}, \widehat{L}\}$ do not quite commute (93), while the commutator of the 2D fields $\{\widehat{F}, \widehat{G}\}$ is zero; and (II) the 1D symmetry generator $\widehat{L}^{(\text{global})} = \int dx\, \widehat{L}(x)$ shifts $\widehat{\phi}$, while the 2D gauge-transformation generator $\widehat{G}^{(\text{global})} = \int dx\, \widehat{G}(x)$ commutes with each $\widehat{\phi}_i$. These imply all terms in this 2D theory are $\mathbf{U}(1)$ symmetric. The $\mathbf{Z}_N$ symmetry comes about when $\mathsf{M} \to \infty$, in which case $\widehat{\phi}_i$ are pinned to multiplies of $\frac{2\pi}{N}$.

Does the above continuum limit preserve key features of the surface-code lattice model? It turns out that topological defects of the pure gauge theory ($\widehat{F}^2$ and $\widehat{G}^2$ terms above) are in fact classified by the same data as the anyonic excitations of the surface code, *when* the $\mathbf{U}(1)$ gauge symmetry is broken to a $\mathbf{Z}_N$ subgroup. A mechanism ( [106], Sec. 1.3) for identifying such excitations proceeds by (A) adding a matter or Higgs field $\widehat{\Phi}$ (which transforms as $\widehat{\Phi} \to e^{iN}\widehat{\Phi}$ under the gauge symmetry), (B) using $\widehat{\Phi}$ to spontaneously break said symmetry to $\mathbf{Z}_N$ via the Higgs mechanism (see [122], Sec. 18.4), and (C) constructing long-range interacting electric

| | $\widehat{\phi}_j^b(y)$ $\widehat{F}^b(y)$ | | $\widehat{L}_{j+1}^b(y)$ | $\widehat{G}^b(y)$ |
|---|---|---|---|---|
| $\widehat{\phi}_i^a(x)$ | 0 | 0 | $i\delta_{ab}\delta_{i,j+1}\delta(x-y)$ | $i\epsilon_{abc}\widehat{\phi}_i^c(x)\delta(x-y) - i\delta_{ab}\partial_i\delta(x-y)$ |
| $\widehat{F}^a(x)$ | $\star$ | 0 | $(-1)^j\left[i\epsilon_{abc}\widehat{\phi}_j^c(x)\delta(x-y) - i\delta_{ab}\partial_j\delta(x-y)\right]$ | $i\epsilon_{abc}\widehat{F}^c(x)\delta(x-y)$ |
| $\widehat{L}_i^a(x)$ | $\star$ | $\star$ | 0 | $i\epsilon_{abc}\widehat{L}_i^c(x)\delta(x-y)$ |
| $\widehat{G}^a(x)$ | $\star$ | $\star$ | $\star$ | $i\epsilon_{abc}\widehat{G}^c(x)\delta(x-y)$ |

Table 7: Table of commutators for the fields of the bulk gauge theories from Sec. D. Each entry lists the commutator $[A, B]$ of an operator $A$ from the first column and $B$ from the top row.

charges and magnetic vortices (topological defects) out of combinations of the gauge and Goldstone fields. In our theory (134), the symmetry is broken using the gauge potential *itself*: the cosine potentials (as M $\to \infty$) ensure that $\widehat{\phi}_i$ are pinned to $\mathbf{Z}_N$, which in turn should constrain the $\widehat{F}$ and $\widehat{G}$ operators (133) in similar fashion.

## D.2 Quantum double to non-Abelian gauge theory

Now let us sketch the embedding of the dihedral quantum-double bulk (18) into a gauge theory. Embedding each dihedral site into $\mathbf{SO}(3)$ space proceeds analogously to the edge case from Sec. 5, yielding the $\mathbf{SO}(3)$ generalized lattice gauge theory

$$\widehat{H}_{\mathbf{D}_N}^{\text{bulk}} \to -\sum_{\square}\sum_{\lambda\in\text{irr}(\mathbf{D}_N)} \text{TR}\left[C_{\ell_\lambda}^{\text{bulk}}.\widehat{U}_{\ell_\lambda}^{\square}.\widehat{U}_{\ell_\lambda}^{\square}.\widehat{U}_{\ell_\lambda}^{\dagger\square}.\widehat{U}_{\ell_\lambda}^{\dagger\square}\right] - \frac{1}{2N}\sum_{+}\sum_{h\in\mathbf{D}_N} \overrightarrow{X}_{R=h}^{+}\overrightarrow{X}_{R=h}^{+}\overleftarrow{X}_{R=h}^{+}\overleftarrow{X}_{R=h}^{+}. \quad (135)$$

The irrep matrices store the dimension $d_\lambda$ of the dihedral irreps $\lambda$ which result from the branching of an angular momentum irrep $\ell$ (see Appx. B). For example, in the angular momentum basis,

$$C_{\ell_{1'}=1}^{\text{bulk}} + C_{\ell_2=1}^{\text{bulk}} = d_{\mathbf{1}}\Pi_{\mathbf{1}}^{\mathbf{D}_N} + d_{\mathbf{2}_1}\Pi_{\mathbf{2}_1}^{\mathbf{D}_N} = \begin{bmatrix} 2 & 0 & 0 \\ 0 & 1 & 0 \\ 0 & 0 & 2 \end{bmatrix}. \quad (136)$$

We now perform the same relabeling as before [52, 82], e.g., $\widehat{U}_\ell^{\square} \to e^{-i\widehat{\phi}_1^a(x)L_\ell^a}$ for the $Z$-type field and $\overleftarrow{X}_R^{+} \to e^{-i\phi_R^a\overleftarrow{L}_2^a(x-2_x)}$ for the $X$-type field. The key difference from the abelian case is the presence of two momentum generators, $\overrightarrow{L}$ and $\overleftarrow{L}$. Since they are related via the equivalence (100), one can obtain the continuum version of that equivalence by recalling that matrix elements of the generators $l^a$ of the adjoint rotation $R$ correspond to the antisymmetric tensor $\epsilon_{abc}$ [ [122], Eq. (4.3)],

$$\overleftarrow{L}_j^a = -\widehat{R^{-1}}_{ab}\overrightarrow{L}_j^b \approx -\overrightarrow{L}_j^a - i\widehat{\phi}^c[l^c]_{ab}\overrightarrow{L}_j^b = -\overrightarrow{L}_j^a - i\epsilon_{abc}\widehat{\phi}^b\overrightarrow{L}_j^c. \quad (137)$$

Taking the continuum limit, renaming $\overrightarrow{L} \to \widehat{L}$, and plugging this into the star term, the plaquette and star terms become exponentials of the following two respective fields,

$$\widehat{F}^a = \partial_1\widehat{\phi}_2^a - \partial_2\widehat{\phi}_1^a - i\epsilon_{abc}\widehat{\phi}_1^b\widehat{\phi}_2^c \quad (138a)$$

$$\widehat{G}^a = \partial_1\widehat{L}_1^a + \partial_2\widehat{L}_2^a - i\epsilon_{abc}\left(\widehat{\phi}_1^b\widehat{L}_1^c + \widehat{\phi}_2^b\widehat{L}_2^c\right). \quad (138b)$$

The field $\widehat{F}$ is the curvature of the $\mathbf{SO}(3)$ gauge potential, while $\widehat{G}$ generate $\mathbf{SO}(3)$ gauge transformations. The commutation relations between the site degrees of freedom $\{\widehat{\phi}_j, \widehat{L}_j\}$ and these

gauge fields are calculated in Table 7. As with the 1D case in Sec. 6.4, derivatives of $\delta$ arise, but remarkably cancel to yield the correct behavior of $\widehat{F}$ and $\widehat{G}$ under gauge transformations.

Expanding the star and plaquette terms yields the $\mathbf{D}_N$-pinned Yang-Mills theory

$$\widehat{H}_{\mathbf{D}_N}^{\text{bulk}} \to \int \mathrm{d}^2 \mathsf{x} \ \mathsf{C}_{\mathsf{ab}} \widehat{F}^{\mathsf{a}}(\mathsf{x}) \widehat{F}^{\mathsf{b}}(\mathsf{x}) + \mathsf{f}_{\mathsf{ab}} \widehat{G}^{\mathsf{a}}(\mathsf{x}) \widehat{G}^{\mathsf{b}}(\mathsf{x}) - \mathrm{M}\widehat{V}_{\mathbf{D}_N}^{\text{smooth}}(\mathsf{x}) \ , \tag{139}$$

with parameters $\{\mathsf{C}_{\mathsf{ab}}, \mathsf{f}_{\mathsf{ab}}\}$ identical to those in the second line of Table 4, but with edge $\to$ bulk. The potential $\widehat{V}_{\mathbf{D}_N}^{\text{smooth}}$ (126) pins both $\widehat{\phi}_1^{\mathsf{a}}$ and $\widehat{\phi}_2^{\mathsf{a}}$. The quadratic terms of this theory, under the Higgs mechanism, are known [106] to admit topological defects with properties identical to the quantum double anyons of the lattice model (18).

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
