# Peer review of "Spin chains, defects, and quantum wires for the quantum-double edge"

_SciPost Physics_

## Round 1 · Referee Report · Michele Burrello (Referee 1) · 2022-1-19

Strengths

1) This work makes significant advances on the study of fundamental models related to topological order 2) It develops a powerful theoretical field description for topological objects hard to tackle, which pose the basis for many potential developments in this and neighboring fields. 3) It exposes very advanced and technical aspects in a clear and detailed way

Report

In the manuscript "Spin chains, defects, and quantum wires for the quantum-double edge" the authors thoroughly investigate the edge theory of quantum double models associated with discrete and non-Abelian groups. Two main results are presented.
First, it is shown through a known coarse-gaining procedure that the edge theory of the 2D quantum double models match a flux ladder model, thus a non-Abelian generalization of quantum clock models.
Second, the authors apply a field theoretical approach to describe the low energy behavior of this model in a continuum limit; this is obtained by embedding the discrete group into a Lie group and applying non-Abelian bosonization techniques.

I find these results very interesting, especially for the community working on systems with topological order. The analysis here presented, for example, opens several stimulating questions about the generalization of parafermionic zero-energy modes to systems with non-Abelian groups and, in particular, on the possibility of describing them through field theory.

Furthermore, I emphasize that these results might be important also for the growing community of scientists working on the quantum simulation of lattice gauge theories: indeed I believe that the field theoretical construction here presented may open the path for more detailed analyses of several models of interest stemming from systems with non-Abelian symmetries.

Therefore, I definitely think that the material here presented is suitable for publication in SCIPOST.

Concerning the exposition of the work, the manuscript is very advanced and technical but the authors did, in general, a good job in trying to present their result in a clear, self-contained and quite didactic way. I believe, however, that a few minor improvements could be considered to further clarify some of the technical aspects of the work.

I list in the following ("Requested changes" section) several points that the authors are invited to consider. Some of them are just very minor details. Some other are slightly broader issues.

I think that, after a minor revision aimed at improving further the readability of this work, the manuscript will be ready for the readership of SciPost.

Requested changes

1) The molding procedure from the 2D model to the bicycle wheel is only briefly discussed in Sec. 3.1 and 3.4, and the readers are referred to Refs. 75 and 76 for details. In order to make the work more self-contained, though, I suggest to extend this part and discuss in more detail how this coarse graining is performed, based on the operations mentioned in Sec. 3.4. Essentially, also at the level of geometry, I would like to see more explicitly how the bicycle wheel is derived and which degrees of freedom are traced out [it seems to me that Refs. 75 and 76 present only the general operations]. This could be discussed in an appendix.

2) When introducing the Hamiltonian 4, the authors write "The most general translationally invariant edge Hamiltonian... consists..." and then they specify that all its terms commute with the bulk operators. To be more rigorous, it would be better to specify first the commutation requirement. Namely, "The most general translationally invariant edge Hamiltonian that commutes with the bulk operators...". Then it may be worth stressing explicitly that the authors are considering a geometry with "smooth" boundaries [in the meaning of Ref. 10]: I guess that a similar dual picture would be valid for "rough" boundaries by considering 3-body plaquette operators and X single site boundary operators.

3) Concerning Eq. 5, it could be useful for the reader to specify that control qunits are taken from selected neighboring sites (as shown in Fig. 4).

4) In Eq. 33a, the matrices $\Pi^K_1$ are introduced a bit too abruptly (although their meaning is explained a few paragraph below). Maybe the authors could slightly extend the comment about their definition after Eq. 33 to introduce their role. Furthermore, since they depend on the irrep $\lambda$ in the summation, I also suggest to add an index $\lambda$.

5) In Sec. 4.2, the authors present the JW operators. I would find interesting to add a few further short comments to argument that indeed they create the expected anyonic excitations. In particular, I think it may be worth stressing the following two points: (i) analogously to the ribbon operators, the JW operators in Eq. 51 do not commute with the two-body $ZZ^\dagger$ terms describing the flux at the end of the string (namely $Tr[Z^{\dagger (x-1)}Z^{x}$] for the first of the JW operators in Eq. 51, and the following two-body term for the second); this implies that they are modifying the last "flux" in the chain. (ii) Their transformation rules under the global symmetry, which the authors briefly mention, match what we see in Eq. 52. Namely, the authors could briefly show that, under a generic global transformation h, the JW flux g is conjugated, while, additionally, the JW operator transforms based on the $\lambda$ irrep. I think this would clarify further that indeed these operators create anyonic excitations with both flux and charge, and it would make the fusion in Eq. 53 more explicit. Actually, I realize that here there could be a further ambiguity to clarify: namely, depending on the chosen convention, one could consider the global symmetry acting or not on the zero-th site [in my argument before I was not transforming $Z^{(0)}$, to show the behavior on the edge only].

6) An observation related to the previous point, which could be useful for some readers, is that the expression 50, if I understand correctly, is not generalized to the non-Abelian case by simply replacing the Abelian with non-Abelian X and Z because the resulting expression would not commute with the inner $\Pi$ operators in eq. 28, in contrast with the flattened ribbons (as shown in page 17). Essentially, it seems to me that the authors are imposing a very strict requirement on the construction of the JW operator: it must preserve all the "fluxes" in the inner part of the chain and not only their conjugacy classes. Maybe the authors could add a brief comment about their construction in Eq. 51 to make this more explicit. My observation stems from the fact that often it is the conjugacy classes of the fluxes which are considered to be the flux excitations of quantum double models [as written in Appendix A2]; in that respect, I would think that the simple non-Abelian definition of the JW operator given by the rhs of Eq. 50 would suffice, because it creates new fluxes in the end points and it "only" conjugates the fluxes in the inner part of the chain. In Sec. 4.2, the authors apply the above stricter requirement and, if they wish, they could emphasize more the reasons for this choice. A consequence of this stricter requirement (commuting with all inner fluxes) is the non-locality of Eq. 54: in the simpler construction taken from the rhs of Eq. 50, the trace in Eq.54 would indeed result in a local operator depending on $X^{(x)}$ only.

7) Another very minor detail the authors could specify to avoid any confusion is that the $\dagger$ taken in the second Eq. 52 refers to both the operators F and the matrices $Z_\lambda$ in Eq. 51.

8) There is a detail concerning Eq. 57 which I ask the authors to check. From their definition in Eq. 56 and the following line, I obtain:

$$ \Gamma^{\dagger (2x-1)} \Gamma^{(2x)} = Z^{\dagger (x)} X^{\dagger (x-1)}... X^{\dagger (1)} X^{(1)} ... X^{(x)} Z^{(x)} = Z^{\dagger (x)} X^{(x)} Z^{(x)} = Z^{\dagger (x)} . Z(g^{-1}) . Z^{(x)} X^{(x)} $$
Here I did not write all the g and lambda indices, nor the right arrows. In the last step I applied Eq. 15a. When I take the trace of the previous term, I do not get $d_\lambda X^{(x)}$, but, instead, $\chi_\lambda(g^{-1}) X^{(x)}$ (where $\chi$ is the character of the irrep $\lambda$ of the previous Z operators), am I correct? This implies that the second term in Eq. 57b is not the same as 57a. I think that the two decoupled JW operators inside the trace must be swapped.

9) Concerning what the authors write after Eq. 58, the fact that the Hamiltonian 57 is in the same form studied in Ref. 71 is not coincidental: Eq. 56 corresponds to the dyonic modes in Eq. 45 of Ref. 71 with trivial auxiliary representation (A=1) [apart for a dagger in the Z operator], and also in Ref. 71 one of the assumptions was the invariance under the global transformations matching X_k (with the right arrow, which corresponds to left gauge transformations). In this respect, as shown by the authors, it is necessary for the second term in Eq. 57 to depend only on the conjugacy classes of g. After the Hamiltonians are expressed in terms of the JW operators, only a few minor differences appear due to their slightly different definitions (when one considers A=1 in Ref. 71, the coupling constants are suitably rewritten and the minor issue at point 8 is considered).

10) Regarding the realization of the flux ladder models in fermionic systems, the authors write that "The motivation is to foster realization of quantum-double non-Abelian defects beyond Majorana and parafermion zero modes in real materials" and they mention "the long-term goal of realizing continuum analogues of quantum-double non-Abelian defects (beyond Majorana and parafermion zero modes) in electronic systems." These statements expose in a generic way long term goals which I can share. However, one must avoid a possible source of confusion: in one-dimensional fermionic systems with local interactions (to be more specific, one dimensional systems which can be described by matrix product states + the standard JW transformation, thus encompassing most of the physical gapped many-body states of matter) there are known no-go theorems that imply that the emerging localized and topologically protected edge modes can be only of the Majorana kind. In particular I am referring to the works Phys. Rev. B 83, 075102 (2011); Phys. Rev. B 83, 075103 (2011); Phys. Rev. B 95, 075108 (2017); and several other extensions also to gapless systems. Therefore I believe it is worth stressing further that, in order to obtain topologically protected and exotic non-Abelian defects in fermionic systems, it will be necessary to consider 2D systems (as in the case of parafermions cited by the authors), in such a way that the theory presented in section 6 must refer to the edges of 2D systems (if the purpose is indeed to get topologically protected modes). My impression is that, in purely 1D fermionic systems, it is still possible to obtain localized zero-energy defect modes, but the degeneracy they cause is not a topological degeneracy but rather a spontaneous G-symmetry breaking. I think that this point does not diminish the importance and the beauty of this work, but must be emphasized to avoid any ambiguity.

11) In the wire description presented in Sec. 6.1 and 6.2, and in many of the works on parafermions, the authors discuss how a combination of backscattering and superconducting interactions can yield parafermions zero modes. If I look at the previous non-Abelian Hamiltonian 78, only one interaction is present, which, If I understand correctly, is a function of the position fields $\phi$ only. Therefore I would interpret it as a generalization of the backscattering interaction. Is it possible to explicitly include an L (or $\theta$) dependent interaction also in the formalism of Eq. 78? Is there a sort of dual interaction with respect to M in Eq. 78?

12) I appreciate the effort of introducing in a clear way non-Abelian bosonization and the mapping from flux ladder operators to the field construction in Sec. 5.3 and 6.4. I am not an expert on these non-Abelian extensions, and I observe that the authors define the operators ($dU/dx . U^\dag$) but never write explicitly U as an exponential of $\phi$. Also the commutation relations presented in table 5 refer to the currents $d \phi / dx$, but not to the field $\phi$. On a practical point of view, though, to derive the commutation relations 99, which I find important since they match Eq. 15, I think one needs an explicit definition of U and an explicit commutation of $\phi$ and L. Can the authors add these details, maybe in appendix C? Essentially, for a person who is not expert in non-Abelian bosonization like me, it is a bit hard to derive Eq. 99, and I would appreciate seeing it in the appendices. I am also wondering whether the commutators in table 5 for the 1D theory, which involve the derivatives of the $\delta$, become similar to the ones in table 7 for the 2D theory when considering the commutators of the fields $\phi$ with L. This would generalize the familiar Abelian relation in Eq. 83, making my understanding better.

13) A very minor detail: in Eq. 104 the notation $\ell=1$ might be a bit inconvenient since $\ell=1$ may create confusion with the trivial representation.

14) Concerning the Abelian bosonization, it may be worth mentioning that a quite extensive bosonization analysis of $\mathbb{Z}_N$ lattice gauge theories on a ladder geometry has been presented in Phys. Rev. Research 3, 013133 (2021). In this work the phase diagram of these discrete gauge theories has been investigated also based to the a renormalization group approach applied to a construction which generalizes the one in Eq. 62 and Sec. 6.1 (to be precise with fields $\theta$ and $\phi$ exchanged). In particular, recipes analogous to Eq. 73 and Eqs. 76-77 have been extensively applied for the description of the $\mathbb{Z}_N$ degrees of freedom.

15) In Eq. 131, I must admit that I do not understand very much the first equality (what is h? Is there a missing sum? it seems that the first equality is true for any h in the group, but I do not see its purpose). I agree in any case with the equality between the first and the last term (which displays a sum over all h).

---

## Editorial Decision

awaiting_resubmission